# Ageing-associated changes in transcriptional elongation influence longevity

Cédric Debès[1,8], Antonios Papadakis[1,8], Sebastian Grönke[2,8], Özlem Karalay[2,8], Luke S. Tain[2], Athanasia Mizi[3], Shuhei Nakamura[2], Oliver Hahn[1,2], Carina Weigelt[2], Natasa Josipovic[3,4], Anne Zirkel[4], Isabell Brusius[1], Konstantinos Sofiadis[3,4], Mantha Lamprousi[3], Yu-Xuan Lu[2], Wenming Huang[2], Reza Esmaillie[1,4,5], Torsten Kubacki[5], Martin R. Späth[1,5], Bernhard Schermer[1,5], Thomas Benzing[1,4,5], Roman-Ulrich Müller[1,5], Adam Antebi[1,2 ✉], Linda Partridge[1,2,6 ✉], Argyris Papantonis[3,4 ✉] & Andreas Beyer[1,4,7 ✉]

Physiological homeostasis becomes compromised during ageing, as a result of impairment of cellular processes, including transcription and RNA splicing[1–4]. However, the molecular mechanisms leading to the loss of transcriptional fidelity are so far elusive, as are ways of preventing it. Here we profiled and analysed genome-wide, ageing-related changes in transcriptional processes across different organisms: nematodes, fruitflies, mice, rats and humans. The average transcriptional elongation speed (RNA polymerase II speed) increased with age in all five species. Along with these changes in elongation speed, we observed changes in splicing, including a reduction of unspliced transcripts and the formation of more circular RNAs. Two lifespan-extending interventions, dietary restriction and lowered insulin–IGF signalling, both reversed most of these ageing-related changes. Genetic variants in RNA polymerase II that reduced its speed in worms[5] and flies[6] increased their lifespan. Similarly, reducing the speed of RNA polymerase II by overexpressing histone components, to counter age-associated changes in nucleosome positioning, also extended lifespan in flies and the division potential of human cells. Our findings uncover fundamental molecular mechanisms underlying animal ageing and lifespan-extending interventions, and point to possible preventive measures.

Ageing impairs a wide range of cellular processes, many of which affect the quality and concentration of proteins. Among these, transcription is particularly important, because it is a main regulator of protein levels[7–9]. Transcriptional elongation is critical for proper mRNA synthesis, owing to the co-transcriptional nature of pre-mRNA processing steps such as splicing, editing and 3′ end formation[2,10]. Indeed, dysregulation of transcriptional elongation results in the formation of erroneous transcripts and can lead to numerous diseases[1,3]. During ageing, animal transcriptomes undergo extensive remodelling, with large-scale changes in the expression of transcripts involved in signalling, DNA damage responses, protein homeostasis, immune responses and stem cell plasticity[11]. Furthermore, some studies uncovered an age-related increase in variability and errors in gene expression[12–14]. Such previous work has provided insights into how the transcriptome adapts to, and is affected by, ageing-associated stress. However, it is not known if, or to what extent, the transcription process itself affects or is affected by ageing.

In this study, we used high-throughput transcriptome profiling to investigate how the kinetics of transcription are affected by ageing, how such changes affect mRNA biosynthesis and to elucidate the role of these changes in age-related loss of function at the organismal level. We show an increase in RNA polymerase II (Pol II) elongation speed with age across five metazoan species, a speed reduction under lifespan-extending conditions and a causal contribution of Pol II elongation speed to lifespan. We thus reveal an association of fine-tuning Pol II speed with genome-wide changes in transcript structure and chromatin organization.

## Pol II elongation speed increases with age

The translocation speed of elongating Pol II can be measured using RNA sequencing (RNA-seq) coverage in introns. This is because Pol II speed and co-transcriptional splicing are reflected in the characteristic saw-tooth pattern of read coverage, observable in total RNA-seq or nascent RNA-seq measurements[15,16]. Read coverage generally decreases 5′ to 3′ along an intron, and the magnitude of this decrease depends on Pol II speed: the faster the elongation, the shallower the slope[17–19]. High speeds of Pol II result in fewer nascent transcripts interrupted within introns at the moment when the cells are lysed. Thus, by quantifying the

[1]Cluster of Excellence on Cellular Stress Responses in Aging-associated Diseases (CECAD), University of Cologne, Cologne, Germany. [2]Max Planck Institute for Biology of Ageing, Cologne, Germany. [3]Institute of Pathology, University Medical Centre Göttingen, Göttingen, Germany. [4]Center for Molecular Medicine Cologne (CMMC), University of Cologne, Faculty of Medicine and University Hospital Cologne, Cologne, Germany. [5]Department II of Internal Medicine, University of Cologne, Faculty of Medicine and University Hospital Cologne, Cologne, Germany. [6]Department of Genetics, Evolution and Environment, Institute of Healthy Ageing, UCL, London, UK. [7]Institute for Genetics, Faculty of Mathematics and Natural Sciences, University of Cologne, Cologne, Germany. [8]These authors contributed equally: Cédric Debès, Antonios Papadakis, Sebastian Grönke, Özlem Karalay. ✉e-mail: aantebi@age.mpg.de; partridge@age.mpg.de; argyris.papantonis@med.uni-goettingen.de; andreas.beyer@uni-koeln.de

gradient of read coverage along an intron, it is possible to determine the elongation speeds of Pol II at individual introns (Fig. 1a,b). Note that this measure is only weakly associated with the expression level of the transcript[20] (Supplementary Table 2). To monitor how the kinetics of transcription changes during ageing, we quantified the distribution of intronic reads resulting from RNA-seq in five animal species—the worm *Caenorhabditis elegans*, the fruitfly *Drosophila melanogaster*, the mouse *Mus musculus*, the rat *Rattus norvegicus*[21] and the human *Homo sapiens*—at different adult ages (see Methods; Supplementary Table 1), and using diverse mammalian tissues (the brain, liver, kidney and whole blood), fly brains and whole worms. Human samples originated from whole blood (healthy donors, 21–70 years of age) and from two primary human cell lines (fetal lung fibroblasts (IMR90) and human umbilical vein endothelial cells (HUVECs)) driven into replicative senescence. After filtering, we obtained between 518 and 6,969 introns that passed quality criteria for reliable Pol II speed quantification (see Methods). These different numbers of usable introns mostly result from interspecies variation in intron size and number, and to some extent from variation in sequencing depth. To rate the robustness of Pol II speed changes across biological replicates, we clustered samples based on their 'speed signatures', that is, on the detected elongation speeds across all introns that could be commonly quantified across each set of experiments. We observed largely consistent co-clustering of samples from the same age across species, whereas young and old samples mostly separated from each other (Extended Data Fig. 1). This suggests that age-related speed changes were consistent across biological replicates and reliably quantifiable in independent measurements. We observed an increase in average Pol II elongation speed with age in all five species and all tissue types examined (Fig. 1c and Extended Data Fig. 2a). Changes in Pol II speed did not correlate with either the length of the intron or with its position within the gene (Supplementary Table 2). The observed increase in Pol II elongation speed was even more pronounced after selecting introns with consistent speed changes across all replicates (that is, always up or down with age; Extended Data Fig. 3). This result is non-trivial because our analysis also revealed introns with a consistent reduction in Pol II speed. To confirm our findings with an orthogonal assay, we monitored transcription kinetics in IMR90 cells using 4-thiouridine (4sU) labelling of nascent RNA. After inhibiting transcription with 5,6-dichloro-1-β-D-ribofuranosyl benzimidazole (DRB), we conducted a pulse-chase-like experiment quantifying 4sU-labelled transcripts at four time points after transcription release (that is, at 0, 15, 30 and 45 min). This enabled us to quantify Pol II progression into gene bodies (see Methods for details) and confirmed our results based on intronic slopes using proliferating (young) and senescent (old) IMR90 cells. Measurements of Pol II speed from the 4sU-based assay showed significant correlation with those from the slope-based assay (Fig. 1d), with Pol II speed increasing on average in both approaches (Fig. 1e and Extended Data Fig. 4). Note that, although many individual genes showed a decrease in elongation speed with ageing in both assays, the majority exhibited increased speed.

To assess whether known lifespan-extending interventions—inhibition of insulin–IGF signalling and dietary restriction—affected Pol II speeds, we sequenced RNA from insulin–IGF signalling mutants, using *daf-2*-mutant worms at day 14 and fly brains from *dilp2-3,5* mutants at day 30 and day 50, as well as the hypothalamus from aged wild-type and *Irs1*-null mice. We also sequenced RNA from the kidney and liver of dietary-restricted and ad libitum-fed mice. In all comparisons, except *Irs1*-null mice and the livers from 26-month-old dietary-restricted mice, lifespan-extending interventions resulted in a significant reduction of Pol II speed. Pol II elongation speeds thus increased with age across a wide range of animal species and tissues, and this increase was, in most cases, reverted under lifespan-extending conditions (Fig. 1).

Although Pol II speed changed consistently with age across replicates (Extended Data Fig. 1), we did not observe specific classes of genes to be affected across models. To determine whether genes with

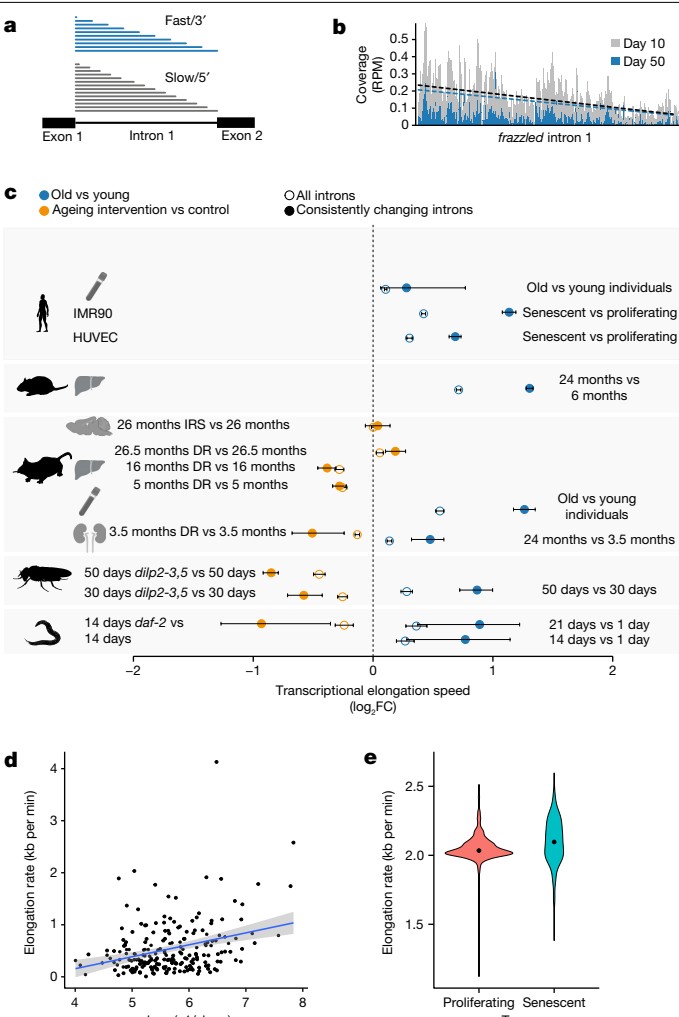

**Fig. 1 | Pol II elongation speed increases with age and is slowed down by reduced insulin signalling and dietary restriction in multiple species.** **a**, Schematic representation of read coverage along introns in total RNA-seq. Intronic reads represent transcriptional production at a given point in time. A shallower slope of the read distribution is a consequence of increased Pol II elongation speed. **b**, Exemplary read distribution in intron 1 of *frazzled*, with coverage in reads per million (RPM) for *D. melanogaster* at age day 10 and day 50. Black dashed line, slope at day 10; blue dashed line, slope at day 50. **c**, Log$_2$ fold change (FC) of average Pol II elongation speeds in the worm (whole body), fruitfly (brains), mouse (the kidney, liver, hypothalamus and blood), rat (liver), human blood, and HUVECs and IMR90 cells. Error bars show median variation ± 95% CI (two-sided paired Wilcoxon test). Empty circles indicate results using all introns passing the initial filter criteria, whereas solid circles show results for introns with consistent effects across replicates. The number of introns considered (*n*) ranged from 518 to 6,969 (see Supplementary Table 3 for details). DR, dietary restriction; IRS, inhibition of insulin–IGF signalling. Dashed line at 0 indicates no change as a visual aid. **d**, Estimate of transcriptional elongation speed from 4sUDRB-seq in IMR90 cells versus intronic slopes for 217 genes for which elongation speed could be estimated using both assays. Each dot represents one gene (Pearson correlation = 0.313, *P* = 2.5 × 10$^{-6}$). The grey band shows the 95% CI for predictions from the linear model of elongation rate-log$_{10}$(−1/slope). **e**, Distribution of elongation speeds in IMR90 cells based on 4sUDRB-seq. The black dot indicates the average speed. The difference between speeds is statistically significant (two-sided paired Wilcoxon test, *P* = 2.13 × 10$^{-10}$). The same genes (464 genes) were used for both conditions (see Methods for details). In panel **c**, the silhouettes of the organs were created using BioRender (https://biorender.com), and the silhouettes of species are from PhyloPic (https://phylopic.org).

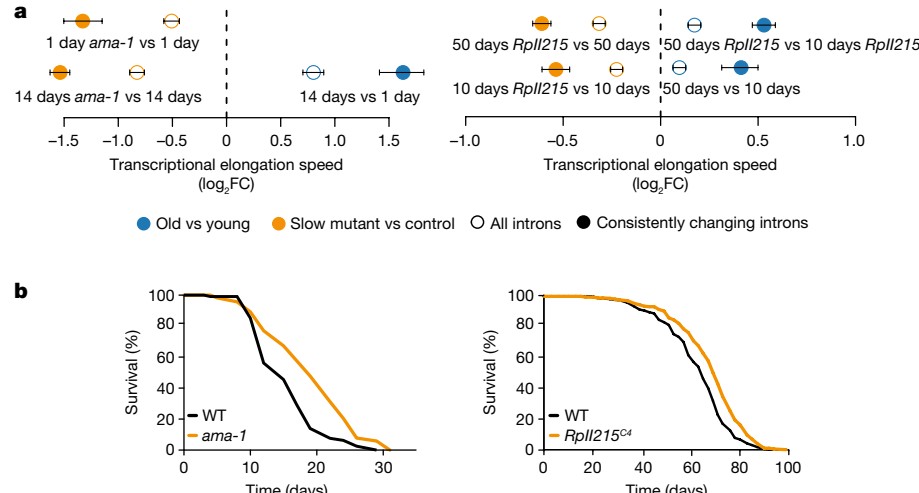

**Fig. 2 | Molecular and lifespan effects of reduced Pol II elongation speed in *C. elegans* and *D. melanogaster*. a**, Differences of average Pol II elongation speeds between Pol II-mutant and wild-type (WT) worms (left; 509 introns) and flies (right; 1,354 introns). Error bars show median variation ± 95% CI. All average changes of Pol II elongation speeds are significantly different from zero (P < 0.001; two-sided paired Wilcoxon test). Empty circles indicate results using all introns passing the initial filter criteria, whereas solid circles show results for introns with consistent effects across replicates. Dashed line at 0 indicates no change as a visual aid. **b**, Survival curves of worms with the *ama-1*(m322) mutation (left; replicate 1) and flies with the *RpII215^C4* mutation (right; averaged survival curve). n = 4 replicates for worms and 3 replicates for flies. Animals with slow Pol II have a significantly increased lifespan (+20% and +10% median lifespan increase for *C. elegans* (n = 120; P < 0.001, log-rank test) and *D. melanogaster* (n = 220; P < 0.001, log-rank test), respectively).

particular functions were more strongly affected by age-related Pol II speed changes, we performed gene set enrichment analysis on the 200 genes with the highest increase in Pol II speed during ageing in worms, fly brains, mouse kidneys and livers, and rat livers. Only very generic functional classes, such as metabolic activity, were consistently enriched across three or more species (Extended Data Fig. 5). Thus, no specific cellular process appeared to be consistently affected across species and tissues. Next, we examined age-associated gene expression changes of transcriptional elongation regulators. We observed that some regulators (for example, *PAF1* and *THOC1*) were consistently downregulated across species during ageing (Extended Data Fig. 6), which was also confirmed using gene set enrichment analysis (Extended Data Fig. 7). These expression changes potentially represent a compensatory cellular response to a detrimental increase in transcriptional elongation speeds.

## Reducing Pol II elongation speed extends lifespan

To determine whether changes in Pol II speed are causally involved in the ageing process, we used genetically modified worm and fly strains carrying point mutations in a main Pol II subunit that reduce its elongation speed (the *ama-1*(m322) mutant in *C. elegans*[5] and the *RpII215^C4* mutant in *D. melanogaster*)[6]. We sequenced total RNA from wild-type and 'slow' Pol II-mutant worms (whole animal at day 14) or fly heads (at day 10 and day 50). Measurements of elongation speeds confirmed the expected reduction of average Pol II speeds in both *C. elegans ama-1*(m322) and *D. melanogaster RpII215^C4* (Fig. 2a). To assess whether Pol II speed and its associated maintenance of transcriptional fidelity also affected ageing of the whole organism, we measured survival of these animals. Slowing down Pol II increased lifespan in both worms and fruitflies (median lifespan increase of approximately 20% in *C. elegans* and approximately 10% in *D. melanogaster*; Fig. 2b and Extended Data Fig. 8a). CRISPR–Cas9-engineered reversal of the Pol II mutations in worms restored lifespan essentially to wild-type levels (Extended Data Fig. 8b). Furthermore, mutant worms displayed higher pharyngeal pumping rates at older age than wild-type worms, suggesting that health span was also extended by slowing down Pol II elongation speed (Extended Data Fig. 8c).

## Changes of transcript structure and sequence

Optimal elongation rates are required for fidelity of alternative splicing for some exons[22,23]. Slow elongation favours weak splice sites that lead to exon inclusion, whereas these exons are skipped if elongation is faster[10,24,25]. Faster elongation rates can also promote intron retention leading to the degradation of transcripts via nonsense-mediated decay[26] and possibly contributing to disease phenotypes[27]. Therefore, we next quantified changes in splicing. The first measure that we used was splicing efficiency, which is the fraction of spliced reads from all reads aligning to a given splice site[28]. In most datasets, from total and nascent RNA-seq, we observed an increase of the spliced exon junctions relative to unspliced junctions during ageing, and a decrease of the percent spliced junctions under lifespan-extending conditions (Fig. 3a). Consistent with earlier work[29], we observed more spliced transcripts under conditions of increased Pol II speed, that is, greater splicing efficiency. For co-transcriptional splicing to occur, Pol II first needs to transcribe all parts relevant to the splicing reaction (that is, 5' donor, branch point and 3' acceptor), which are located at the opposite ends of an intron[30,31]. Our data suggest that accelerated transcription shortens the interval in which splicing choices are made, thus shortening the time between nascent RNA synthesis and intron removal.

Accelerated transcription and splicing carries the risk of increasing the frequency of erroneous splicing events, which has been associated with advanced age and shortened lifespan[4,32–34]. It is non-trivial to deduce whether a specific splice isoform is the product of erroneous splicing or created in response to a specific signal. Simply checking whether an observed isoform is annotated in some database can be problematic for multiple reasons. For instance, most databases have been created on the basis of data from young animals or embryonic tissue. Thus, a detected isoform that only may be functionally relevant in old animals will not be reported in such databases. Moreover, an annotated isoform might be the result of erroneous splicing if its expression is normally suppressed at a particular age or cellular context. We therefore based our analysis on the notion that extremely rare isoforms (rare with respect to all other isoforms of the same gene in the same sample) are more likely erroneous than frequent isoforms[35,36]. We used Leafcutter[37],

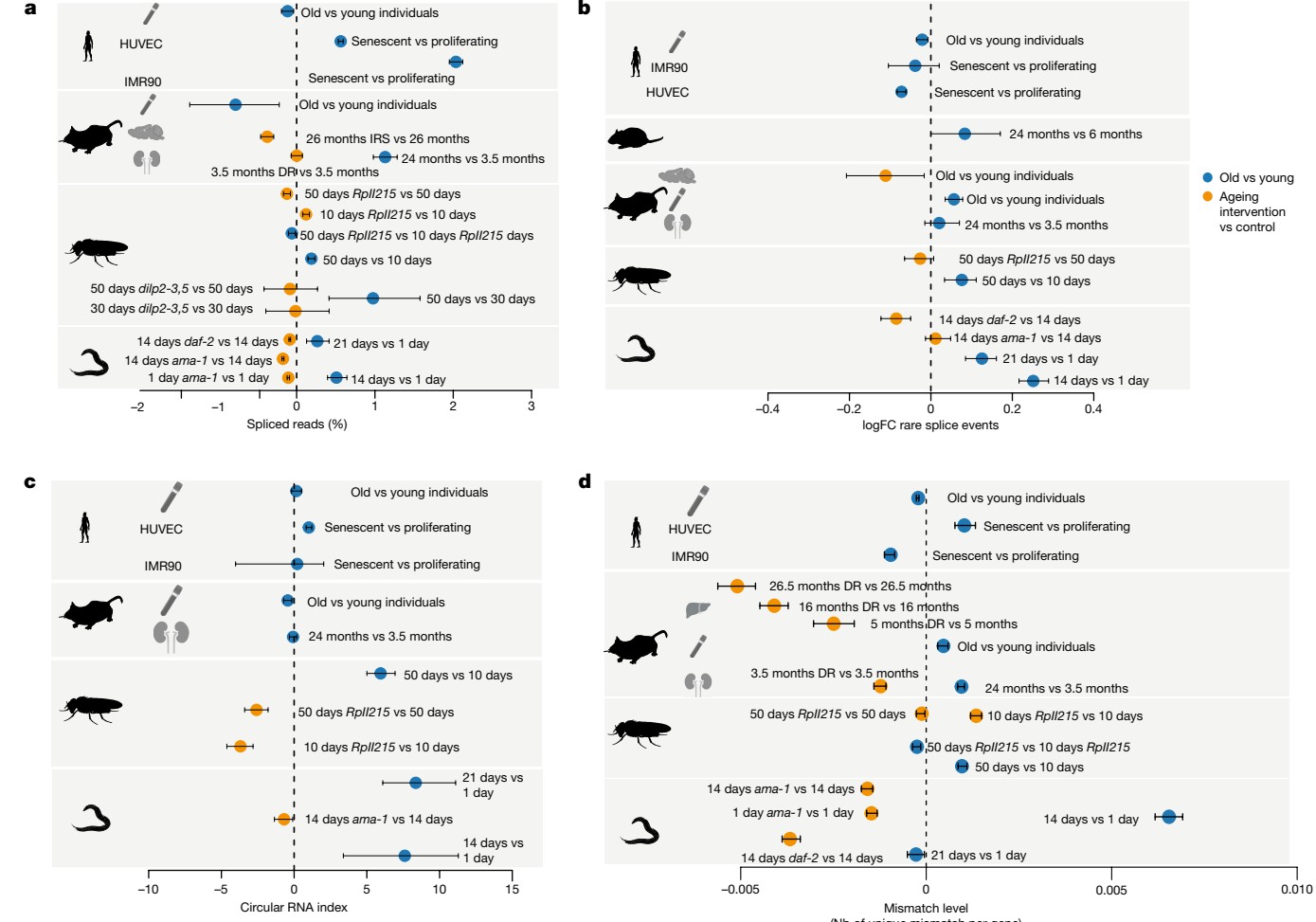

**Fig. 3 | Changes in transcript structure upon ageing (old versus young) and after lifespan-extending interventions. a**, Average changes of the fraction of spliced transcripts. The number of genes considered (*n*) ranged from 120 to 15,328. **b**, Average per cent changes of rare splice events (0.7% or less of total gene expression). The number of genes considered (*n*) ranged from 376 to 8,486. **c**, CircRNA index (back-spliced reads divided by the sum of linear and back-spliced reads) for worms, fly heads, mouse and rat livers, and human cell lines. The number of back-spliced junctions considered (*n*) ranged from 1,121 to 41,004. **d**, Changes in the average mismatch level. The number of genes considered (*n*) ranged from 1,950 to 8,620. Error bars show median variation ± 95% CI. See Supplementary Table 3 for details on the number of genes and back-spliced junctions used per comparison. In all panels, the dashed line at 0 indicates no change as a visual aid. In all panels, the silhouettes of the organs were created using BioRender (https://biorender.com), and the silhouettes of species are from PhyloPic (https://phylopic.org).

which performs de novo quantification of exon–exon junctions based on split-mapped RNA-seq reads. Owing to its ability to identify alternatively excised intron clusters, Leafcutter is particularly suitable to study rare exon–exon junctions[38]. We defined rare splicing events as exon–exon junctions supported by 0.7% or less of the total number of reads in a given intron cluster, and the gene-specific fraction of rare clusters was computed as the number of rare exon–exon junctions divided by the total number of detected exon–exon junctions in that gene. We observed that such rare exon–exon junctions often resulted from exon skipping or from the usage of cryptic splice sites (Extended Data Fig. 9). The average fraction of rare splicing events increased during ageing in the fly and worm, and this effect was reverted under most lifespan-extending conditions (Fig. 3b). However, we did not observe a consistent age-associated increase of the fraction of rare splice variants across all species, which may at least in part be due to the more complex organization of splice regulation in mammalian cells.

Another potential indicator of transcriptional noise is the increased formation of circular RNAs (circRNAs)[39,40], that is, of back-spliced transcripts with covalently linked 3′ and 5′ ends[41]. Increased speed of Pol II has previously been associated with increased abundance of circRNA[42].

Thus, we quantified the fraction of circRNAs as the number of back-spliced junctions normalized by the sum of back-spliced fragments and linearly spliced fragments. We observed either increased or unchanged average circRNA fractions during ageing, whereas reducing the speed of Pol II also reduced the formation of circRNAs (Fig. 3c). This suggests that faster Pol II elongation correlates with a general increase in the level of circRNAs. Nevertheless, our data do not provide evidence that increased levels of circRNA directly result from increased Pol II speed, despite it being a consequence of the overall reduced quality in RNA production.

Increased speeds of Pol II can lead to more transcriptional errors because the proofreading capacity of Pol II is challenged[12]. To assess the potential effect of accelerated elongation on transcript quality beyond splicing, we measured the number of mismatches in aligned reads for each gene. For this, we normalized mismatch occurrence to individual gene expression levels and excluded mismatches that were probably due to genomic variation or other artefacts (see Methods for details). We observed that the average fraction of mismatches increased with age, but decreased under most lifespan-extending treatments (Fig. 3d). Consistent with previous findings[12], slow Pol II mutants

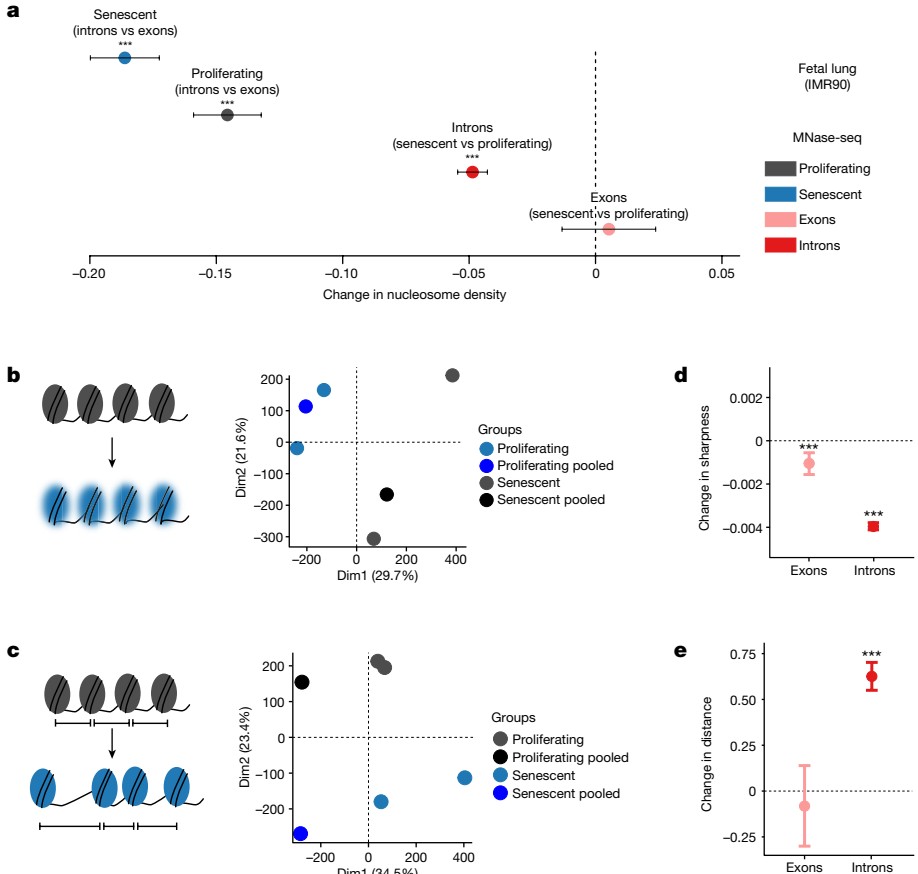

**Fig. 4 | Profiling of nucleosome positioning in human cell models. a**, Average differences in nucleosome density between exons (*n* = 37,625) and introns (*n* = 193,912), and between proliferating and senescent cells. Error bars show median variation ± 95% CI. **b,c**, Principal component analysis plots of nucleosome sharpness (**b**) and distances between nucleosome summits (**c**) in introns for individual samples and pooled data. **d**, Changes in nucleosome sharpness between senescent and proliferating cells in exons (*n* = 37,277) and introns (*n* = 193,131). **e**, Changes in distance between nucleosome summits between senescent and proliferating cells in exons (*n* = 36,956) and introns (*n* = 192,194). For **a,d,e**, error bars show median variation ± 95% CI. Statistical significance of difference in pseudo-median distribution is indicated by asterisks (\*\*\**P* < 0.001, two-sided paired Wilcoxon rank test). Dashed line at 0 indicates no change as a visual aid.

exhibited reduced numbers of mismatches compared with wild-type control levels in three out of four comparisons.

## Changes in chromatin structure associated with Pol II elongation

Subsequently, we explored alterations in chromatin structure as a possible cause of the age-associated changes in Pol II speeds. Nucleosome positioning along DNA is known to affect both Pol II elongation and splicing[19,43–45]. Furthermore, aged eukaryotic cells display reduced nucleosomal density in chromatin and 'fuzzier' core nucleosome positioning[46,47]. Thus, age-associated changes in chromatin structure could contribute to the changes in Pol II speed and splicing efficiency that we observed. To test this, we performed micrococcal nuclease (MNase) digestion of chromatin from early (proliferating) and late-passage (senescent) human IMR90 cells, followed by approximately 400 million paired-end read sequencing of mononucleosomal DNA (MNase-seq). Following mapping, we examined nucleosome occupancy. In senescent cells, introns were less densely populated with nucleosomes than proliferating cells[48] (Fig. 4a). In addition, we quantified peak 'sharpness', reflecting the precision of nucleosome positioning in a given MNase-seq dataset (see Methods), as well as the distances between consecutive nucleosomal summits as a measure of the spacing regularity[48,49]. Principal component analysis of the resulting signatures indicated consistent changes of nucleosome sharpness and distances upon entry into senescence as the samples clearly separated by condition (Fig. 4b,c).

Both measures were significantly, but moderately, altered in senescent cells (Fig. 4d,e): average sharpness was slightly decreased (along both exons and introns) and average internucleosomal distances slightly increased in introns. In conclusion, the transition from a proliferating cell state to replicative senescence was associated with small but significant changes in chromatin structure, involving nucleosome density and positioning changes that were previously shown to affect Pol II elongation[43,47,50].

## Overexpression of histone proteins decreases Pol II elongation speed and extends fly lifespan

The organization of nucleosomes is severely influenced by histone availability[45,46]. For example, histone H3 depletion reduces nucleosomal density and renders chromatin more accessible to MNase digestion[51]. Such global loss of histones constitutes a hallmark of ageing and senescence[52]. Consistent with this, our senescent IMR90 cells and HUVECs carry significantly reduced protein levels of histone H3 (Fig. 5a). Conversely, elevated levels of histones promote lifespan extension in yeast[46], *C. elegans*[53] and *D. melanogaster*[54]. To assess whether Pol II elongation speed and senescence entry in human cells are causally affected by changes in nucleosomal density, we generated IMR90 cell populations homogeneously overexpressing GFP-tagged H3 or H4 in an inducible manner (Fig. 5b and Extended Data Fig. 10a,b). Overexpression of either histone resulted in significant reduction of Pol II speed, confirming the causal connection between chromatin

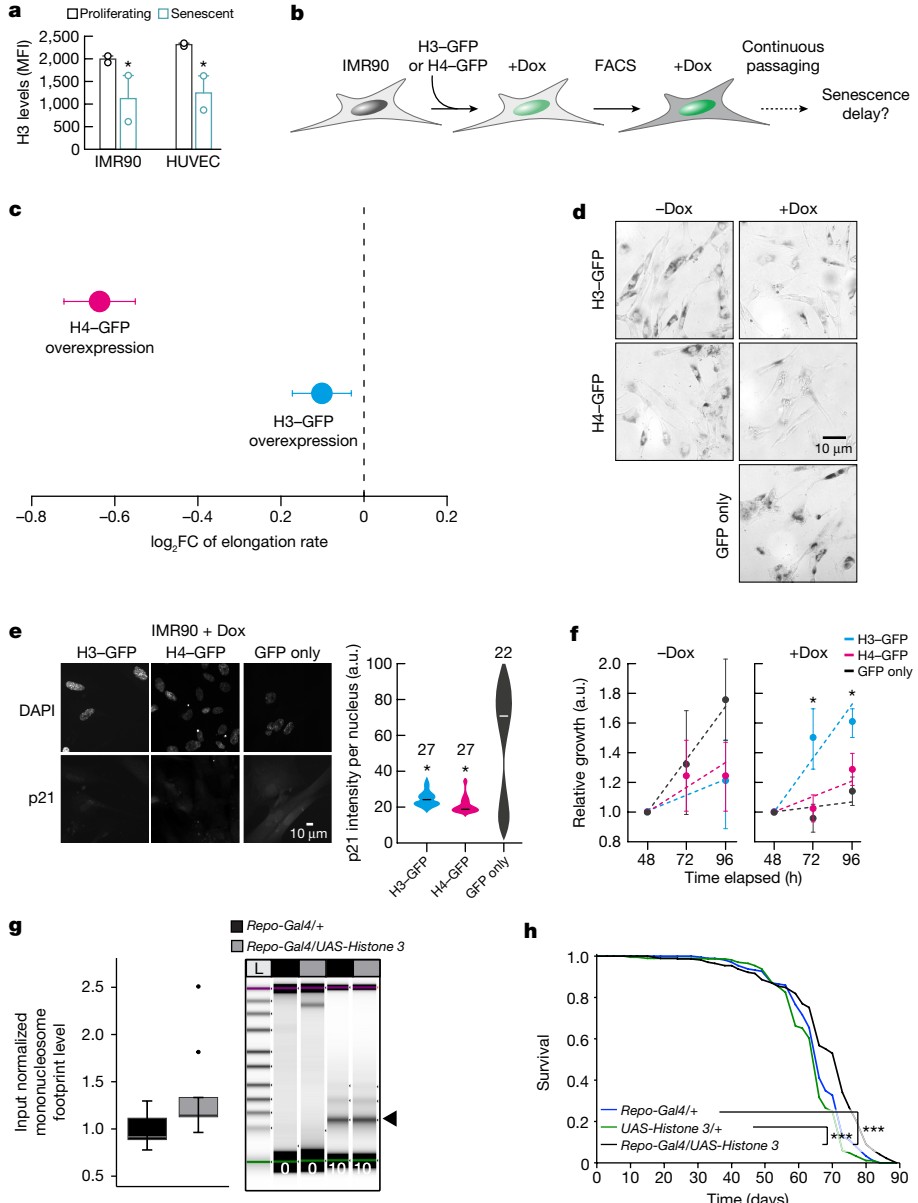

**Fig. 5 | Histone overexpression slows down entry into senescence and decreases Pol II speed. a**, Levels of H3 protein in proliferating and senescent HUVECs and IMR90 cells. Each data point comes from an independent biological replicate (different HUVEC donor or IMR90 isolate) and is the mean of three technical triplicates. H3 expression decreases with senescence (*$P < 0.05$, two-tailed Student's $t$-test). MFI, mean fluorescence intensity. **b**, Schematic representation of the experiment. FACS, fluorescence-activated cell sorting. **c**, Differences of average Pol II elongation speeds between histone overexpression mutants and wild-type IMR90 cells (derived from 1,212 introns). Error bars show median variation ± 95% CI. All average changes of Pol II elongation speeds are significantly different from zero ($P < 0.01$; paired Wilcoxon rank test). Dashed line at 0 indicates no change as a visual aid. **d**, Typical images from β-galactosidase staining of H3–GFP, H4–GFP and control IMR90 cells, in the presence and absence of doxycycline (Dox). **e**, Typical immunofluorescence images of H3–GFP, H4–GFP and control IMR90 cells (left) show reduced levels of p21 in histone overexpression nuclei. Violin plots (right) quantify this

reduction (*$P < 0.05$, two-tailed Student's $t$-test). The number ($n$) of cells analysed per condition is indicated. **f**, MTT proliferation assay. For each H3–GFP or H4–GFP overexpression, mean ± s.d. were derived from two independent clones over three replicates for each clone. (*$P < 0.05$, two-tailed Student's $t$-test). **g**, Quantification of input normalized mononucleosome footprints (black arrowhead) between the heads of aged (60 days) flies overexpressing *Histone 3* in glial cells (*Repo-Gal4/UAS-Histone 3*) and control fly heads (*Repo-Gal4/+*). Significance was determined by paired two-tailed $t$-test (ten biologically independent replicates per fly group; *$P = 0.0417$). The upper and lower 'hinges' of the boxplot correspond to the first and third quartiles of the measurements. Digests were halted after 10 min and visualized by Tapestation (Agilent) ($n > 5$). The lines in each lane represent the internal upper (purple) and lower (green) markers, for sizing and alignment. L, DNA ladder. **h**, Lifespan analysis of *Repo-Gal4/UAS-Histone 3* flies and control flies (*Repo-Gal4/+* and *UAS-Histone 3/+*) ($n = 200$, ***$P < 0.001$).

structure and transcriptional elongation (Fig. 5c). Reduction of Pol II speed was accompanied by markedly reduced senescence-associated β-galactosidase staining in H3-overexpressing or H4-overexpressing cells compared with both control (GFP only) and uninduced cells (Fig. 5d). Moreover, both H3-overexpressing and H4-overexpressing

cells did not display induction of p21 or depletion of HMGB1, which are both hallmarks of senescence entry, compared with control IMR90 cells (Fig. 5e and Extended Data Fig. 10c). Finally, MTT assays showed that viability and proliferation were improved in H3-overexpressing and, to a lesser extent, in H4-overexpressing cells compared with control cells

(Fig. 5f). Together, these results suggest that H3 or H4 overexpression decelerates Pol II and compensates for the ageing-induced core histone loss[44,50] to restrict senescence entry.

The average speed reduction following H4 overexpression was significantly larger than that obtained upon H3 overexpression, yet H4-overexpressing near-senescent IMR90 cells only marginally outperformed control cells in MTT assays (Fig. 5f). This raises the possibility of excessive reduction in Pol II speed negatively affecting aspects of cell function[55]. To address the role of nucleosome density in organismal lifespan, we used UAS-Histone 3 (ref. 54) to overexpress *His3*, specifically in *Drosophila* glial cells using Repo-Gal4. H3 overexpression led to significantly increased numbers of mononucleosomes in aged (60 days of age) compared with wild-type fly heads (Fig. 5g), thus possibly compensating for age-associated loss of histone proteins. Furthermore, H3 overexpression in glial cells increased fruitfly lifespan (Fig. 5h). These in vivo results are consistent with our in vitro data from IMR90 cells, demonstrating that H3 overexpression partially reverts the ageing effects on chromatin density and promotes longevity in flies. As this was linked to a reversal in Pol II elongation speed, our findings, together with earlier findings in yeast[46,51], *C. elegans*[53] and *D. melanogaster*[54], demonstrate how the structure of the chromatin fibre probably modulates Pol II elongation speed and lifespan.

## Discussion

We found a consistent increase in the average intronic Pol II elongation speed with age across four animal models, two human cell lines and human blood, and could revert this trend by using lifespan-extending treatments. We also documented ageing-related changes in splicing and transcript quality, such as the elevated formation of circRNAs and increased numbers of mismatches with genome sequences, which probably contribute to age-associated phenotypes. Furthermore, we observed a consistent increase in the ratios of spliced to unspliced transcripts (splicing efficiency) with age across species (Fig. 3a), which has been reported to be a result of increased elongation speed[29]. However, we cannot exclude the possibility that this increase resulted from changes in RNA half-lives. Although average speed changes were predominantly significant, they remained small in absolute terms. This is expected, as drastic, genome-wide changes of RNA biosynthesis would quickly be detrimental for cellular functions and would probably lead to early death. Instead, what we monitored here is a gradual reduction of cellular fitness characteristic for normal ageing. Critically, we were able to increase lifespan in two species by decelerating Pol II. Thus, despite being small in magnitude and stochastically emerging in tissues or cell populations, these effects are clearly relevant for organismal lifespan.

Genes exhibiting accelerated Pol II elongation were not enriched for specific cellular processes, indicating that speed increase is probably not a deterministically cell-regulated response, but rather a spontaneous age-associated defect. Yet, the genes affected were not completely random, as we observed consistent changes across replicates for a subset of introns. Thus, there must be location-specific factors influencing which genomic regions are more prone to increases in Pol II speed and which are not. This observation is consistent with earlier findings and our data, indicating that chromatin structure may causally contribute to age-associated Pol II speed increase. Although we still lack a complete understanding of the molecular events driving Pol II speed increase, our findings indicate that ageing-associated changes in chromatin structure have an important role.

Our work establishes Pol II elongation speed as an important contributor to molecular and physiological traits with implications beyond ageing. Misregulation of transcriptional elongation reduces cellular and organismal fitness and may therefore contribute to disease phenotypes[53,56,57]. Together, the data presented here reveal a molecular mechanism contributing to ageing and serve as a means for assessing the fidelity of the cellular machinery during ageing and disease.

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

## Methods

### Worm strains and demography assays

Nematodes were cultured using standard techniques at 20 °C on nematode growth media (NGM) agar plates and were fed with *Escherichia coli* strain OP50. The DR786 strain carrying the *ama-1*(m322) IV mutation in the large subunit of Pol II (RBP1), which confers α-amanitin resistance, was obtained from the *Caenorhabditis* Genetics Center[58,59]. The DR786 strain was then outcrossed into wild-type N2 strain four times and the mutation was confirmed by sequencing. The 5′–3′ AGAAGGTCACACAATCGGAATC primer was used for sequencing. For each genotype, a minimum of 120 age-matched day 1 young adults were scored every other day for survival and transferred to new plates to avoid starvation and carry-over progeny. Lifespan analyses using the *C. elegans* Lifespan Machine were conducted as previously described[60]. In brief, wild-type N2 and mutant worms were synchronized by egg-prep (hypochlorite treatment) and grown on NGM agar plates seeded with OP50 at 20 °C. Upon reaching L4 stage, these worms were transferred onto plates containing 0.1 g ml$^{-1}$ 5-fluoro-2′-deoxyuridine (FUDR) and placed into the modified flatbed scanners (35 worms per plate). The scan interval was 30 min. Objects falsely identified as worms were censored. Time of death was automatically determined by the *C. elegans* Lifespan Machine[60]. Demography experiments were repeated multiple times. For all experiments, genotypes were blinded. Statistical analyses were performed using the Mantel–Cox log-rank method.

### Measurements of pharyngeal pumping rates in worms

Synchronized wild-type and *ama-1*(m322) animals were placed on regular NGM plates seeded with OP50 bacteria on day 1 and day 8 of adulthood, and the pharyngeal pumping rate was assessed by observing the number of pharyngeal contractions during a 10-s interval using a dissecting microscope and Leica Application Suite X imaging software. The pharyngeal pumping rate was then adjusted for the number of pharyngeal pumping per minute. Animals that displayed bursting, internal hatching and death were excluded from the experiments. Experiments were repeated three independent times in a blinded manner, scoring a minimum of 15 randomly selected animals per genotype and time point for each experiment. One-way ANOVA with Tukey's multiple comparison test was used for statistical significance testing, with ****$P < 0.0001$, and error bars representing standard deviation.

### Fly strains and fly maintenance

The *RpII215$^{C4}$* fly strain (RRID:BDSC_3663), which carries a single point mutation (R741H) in the gene encoding the *Drosophila* Pol II 215 kDa subunit (RBP1), was received from the Bloomington *Drosophila* Stock Center. Flies carrying the *RpII215$^{C4}$* allele[61] are homozygous viable but show a reduced transcription elongation rate[6]. *RpII215$^{C4}$* mutants were backcrossed for six generations into the outbred white Dahomey (wDah) wild-type strain. A PCR screening strategy was used to follow the *RpII215$^{C4}$* allele during backcrossing. Therefore, genomic DNA from individual flies was used as a template for a PCR with primers SOL1064 (CCGGATCACTGCTGCATATTTGTT) and SOL1047 (CCGCGCGACTCAGGACCAT). The 582-bp PCR product was restricted with BspHI, which specifically cuts only in the *RpII215$^{C4}$* allele, resulting in two bands of 281 bp and 300 bp. At least 20 individual positive female flies were used for each backcrossing round. Long-lived insulin-mutant flies, which lack three of the seven *Drosophila* insulin-like peptides, *dilp2-3,5* mutants (RRID:BDSC_30889)[62], were also backcrossed into the wDah strain, which was used as the wild-type control in all fly experiments. Flies were maintained and experiments were conducted on 1,0 SY-A medium at 25 °C and 65% humidity on a 12-h light–12-h dark cycle[62].

### Fly lifespan assays

For lifespan assays, fly eggs of homozygous parental flies were collected during a 12-h time window and the same volume of embryos was transferred to each rearing bottle, ensuring standard larval density. Flies that eclosed during a 12-h time window were transferred to fresh bottles and were allowed to mate for 48 h. Subsequently, flies were sorted under brief $CO_2$ anaesthesia and transferred to vials. Flies were maintained at a density of 15 flies per vial and were transferred to fresh vials every 2–3 days and the number of dead flies was counted. Lifespan data were recorded using Excel and were subjected to survival analysis (log-rank test) and presented as survival curves.

### Mouse maintenance and dietary restriction protocol

The dietary restriction (DR) study was performed in accordance with the recommendations and guideline of the Federation of the European Laboratory Animal Science Association (FELASA), with all protocols approved by the Landesamt für Natur, Umwelt und Verbraucherschutz, Nordrhein-Westfalen, Germany. Details of the mouse liver DR protocol have been previously published[63]. For the mouse kidney, male C57BL/6 mice were housed under identical specific pathogen-free conditions in group cages (five or fewer animals per cage) at a relative humidity of 50–60% and a 12-h light and 12-h dark rhythm. For DR versus control, 8-week-old mice were used. DR was applied for 4 weeks. Control mice received food and water ad libitum. Mice were killed at 12 weeks. For comparison of young versus aged mice, 14-week-old and 96-week-old mice were used. Food was obtained from ssniff (Art. V1534-703) and Special Diet Services. The average amount of food consumed by a mouse was determined by daily weighing for a period of 2 weeks and was on average 4.3 g daily. DR was applied for 4 weeks by feeding 70% of the measured ad libitum amount of food. Water was provided ad libitum. Mice were weighed weekly to monitor weight loss. Neither increased mortality nor morbidity was observed during DR.

### RNA extraction

Wild-type N2 strain, α-amanitin-resistant *ama-1*(m322) mutants and long-lived insulin–IGF signalling mutants, *daf-2*(e1370), were sent for RNA-seq. For each genotype, more than 300 aged-matched adult worms at desired time points were collected in TRIzol (Thermo Fisher Scientific) in three biological replicates. Total RNA was extracted using the RNAeasy Mini kit (Qiagen). The RNA-seq data for brains of 30-day-old and 50-day-old *dilp2-3,5* and wDah control flies have been previously published[64]. Ten-day-old and 50-day-old *RpII215$^{C4}$* mutants and wDah control flies were snap frozen and fly heads were isolated by vortexing and sieving on dry ice. Total RNA from three biological samples per treatment group was prepared using TRIzol reagent according to the manufacturer's instructions, followed by DNase treatment with the TURBO DNA-free Kit (Thermo Fisher Scientific). Mouse liver samples were isolated from 5-month-old, 16-month-old and 27-month-old ad libitum and DR animals, which corresponded to 2, 13 and 24 months of DR treatment, respectively. RNA was isolated by TRIzol and was treated with DNase. The RNA-seq data for 5-month-old and 27-month-old liver DR samples have been previously published[63], whereas the data for 16 months are first published here. The RNeasy Mini Kit and TRIzol were used to isolate RNA from snap-frozen kidneys as per manufacturer's instructions. Hypothalamus tissue of long-lived insulin receptor substrate 1 (*Irs1$^{-/-}$*) knockout mice[65] and C57BL/6 black control animals was dissected manually at the age of 27 months. RNA was isolated by TRIzol with subsequent DNase treatment. For blood samples, globin RNA was removed using the GLOBINclear Kit mouse/rat/human for globin mRNA depletion.

### Human whole-blood sample acquisition and RNA extraction

Human samples were obtained as part of a clinical study on ageing-associated molecular changes (German Clinical Trials Register: DRKS00014637) at University Hospital Cologne. The study cohort consisted of healthy participants between 21 and 70 years of age. Whole-blood samples were obtained using the PAXgene Blood RNA system (Becton Dickinson GmbH) directly after informed consent.

After storage at −80 °C for at least 24 h, RNA extraction was performed by usage of the PAXgene Blood RNA Kit (Qiagen) according to the manufacturer's protocol. The study was operated in accordance with the Declaration of Helsinki and the good clinical practice guidelines by the International Conference on Harmonization. All patients provided informed consent and approval of each study protocol was obtained from the local institutional review board (Ethics committee of the University of Cologne (17-362, 2018-01-17)).

## Human cell culture
HUVECs from pooled donors (Lonza) and human fetal lung (IMR90) cells (from two donors) were grown to 80–90% confluence in endothelial basal medium 2-MV with supplements (EBM; Lonza) and 5% FBS and MEM (Sigma-Aldrich) with 20% FBS (Gibco) and 1% non-essential amino acids (Sigma-Aldrich) for HUVECs and IMR90 cells, respectively.

## Total RNA and nascent RNA-seq
From 1 μg input of total RNA, ribosomal RNA was removed using the Ribo-Zero Human/Mouse/Rat kit (Illumina). Sequencing libraries were generated according to the TruSeq stranded total RNA (Illumina) protocol. To generate the final complementary DNA (cDNA) library, products were purified and amplified by PCR for 15 cycles. After validation and quantification of the library on an Agilent 2100 Bioanalyzer, equimolar amounts of libraries were pooled. Pools of five libraries were sequenced per lane on an Illumina HiSeq 4000 sequencer. For a description of all the RNA-seq datasets used in this study, see Supplementary Table 1. The same protocol was used to sequence cDNA libraries from human cell 'factory' RNA, which was isolated as previously described[66].

## RNA-seq alignments and gene expression analysis
Raw reads were trimmed with trimmomatic[67] version 0.33 using parameters 'ILLUMINACLIP:./Trimmomatic-0.33/adapters/TruSeq3-PE.fa:2:30:10 LEADING:3 TRAILING:3 SLIDINGWINDOW:4:15 MINLEN:45' for paired-end datasets and 'ILLUMINACLIP:./Trimmomatic-0.33/adapters/TruSeq3-SE.fa:2:30:10 LEADING:3 TRAILING:3 SLIDINGWINDOW:4:15 MINLEN:45' for single-end datasets. Alignment was performed with STAR version 2.5.1b[68] using the following parameters: '–outFilterType BySJout –outWigNorm None' on the genome version mm10, rn5, hg38, dm6 and ce5 for *M. musculus*, *R. norvegicus*, *H. sapiens*, *D. melanogaster* and *C. elegans*, respectively. We estimated transcript counts using Kallisto version 0.42.5 for each sample. To determine differentially expressed genes, we used DESeq2 version 1.8.2 (ref. 69) with RUVr normalization version 1.6.2 (ref. 70). For the differential analysis of transcriptional elongation regulators, we downloaded the list of positive and negative regulators from the GSEA/MSigDB[71]. Gene ontology (GO) term enrichment analysis of differentially expressed genes or genes with increased Pol II elongation speed was carried out using TopGO version 2.20.0. For GO enrichment analysis of differentially expressed genes, we identified 4,784 genes as evolutionarily conserved from each species of our study to humans: genes were either direct orthologues (one2one) or fusion genes (one2many) of *H. sapiens* were retrieved from ENSEMBL database using biomaRt 2.24.1 (ref. 72). Using our 4,784 genes evolutionary conserved, we further divided into consistently upregulated or downregulated genes across species during ageing or ageing intervention (as the target set for GO enrichment: 92 genes (ageing upregulated) and 71 genes (ageing downregulated); and 164 genes (ageing intervention upregulated) and 473 genes (ageing intervention downregulated) as background set of 4,784 orthologue genes between *R. norvegicus*, *M. musculus*, *D. melanogaster*, *C. elegans* and *H. sapiens*). For GO enrichment analysis of genes harbouring increasing Pol II speed, we used as the target set the top 200 or 300 genes with an increase in Pol II speed change for each species. Quantification of transcript abundance for *ITPR1* and *AGO3* was obtained by using StringTie[73]. For circRNAs, we aligned the reads using STAR version 2.5.1b[68] with the following parameters:

'–chimSegmentMin 15 –outSJfilterOverhangMin 15 15 15 15 –alignSJoverhangMin 15 –alignSJDBoverhangMin 15 –seedSearchStartLmax 30 –outFilterMultimapNmax 20 –outFilterScoreMin 1 –outFilterMatchNmin 1 –outFilterMismatchNmax 2 –chimScoreMin 15 –chimScoreSeparation 10 –chimJunctionOverhangMin 15'. We then extracted back-spliced reads from the STAR chimeric output file and normalized the number of back-spliced reads by the sum of back-spliced (BS$_i$) and spliced reads from linear transcripts (S1$_i$, S2$_i$) for an exon *i* (8):

$$\text{CircRatio}_i = \frac{\text{BS}_i}{\text{BS}_i + \frac{S1_i + S2_i}{2}} \times 100$$

Here, S1$_i$ refers to the number of linearly spliced reads at the 5′ end of the exon and S2$_i$ refers to the respective number of reads at the 3′ end of the exon. Thus, this score quantifies the percent of transcripts from this locus that resulted in circRNA. Finally, we quantified the significance of the average change in circRNA formation between two conditions using the Wilcoxon rank test.

## Definition of intronic regions
All annotation files for this analysis were downloaded from the Ensembl website[74] using genome version ce10 for *C. elegans*, mm10 for *M. musculus*, hg38 for *H. sapiens*, rn5 for *R. norvegicus* and dm6 for *D. melanogaster*. The following filtering steps were applied on the intronic ENSEMBL annotation files. First, we removed overlapping regions between introns and exons to avoid confounding signals due to variation in splicing or transcription initiation and termination. Overlapping introns were merged to remove duplicated regions from the analysis. In the next step, we used STAR[68] to detect splice junctions and compared them with the intronic regions. Introns with at least five split reads bridging the intron (that is, mapping to the flanking exons) per condition were kept for subsequent analyses. As a result, we ensured a minimum expression level of the spliced transcript. When splice junctions were detected within introns, we further subdivided those introns accordingly. Introns with splice junction straddling were discarded. The above-mentioned steps were performed using Bedtools version 2.22.1 using subtract and merge commands. After these filtering steps, the number of usable introns per sample varied between a few hundred (n = 546, *C. elegans*, total RNA) to over 10,000 (n = 13,790, *H. sapiens*, nascent RNA-seq). These large differences resulted from different sequencing depths, sequencing quality (number of usable reads) and from the complexity of the genome (numbers and sizes of introns, number of alternative isoforms, among others). To avoid artefacts due to the different numbers of introns used per sample, we always contrasted the same sets of introns for each comparison of different conditions (for example, old versus young; treatment versus control). Note that certain comparisons were not possible for all species owing to variations in the experimental design. For instance, for mouse kidney, only a single time point after lifespan intervention (DR, age 3 months) was available, which prevented a comparison of old versus young DR mice, but allowed comparison with ad libitum-fed mice at the young age.

## Transcriptional elongation speed based on intronic read distribution
To calculate Pol II speeds, we used RNA-seq data obtained from total RNA[75] and nascent RNA[65,76] enrichment. In contrast to the widely used polyA enrichment method[77], which primarily captures mature, spliced mRNAs and is therefore not suitable to estimate Pol II speeds based on intronic reads, these methods yield sufficient intronic coverage to quantify elongation rates. To analyse the distribution of intronic reads between conditions, we devised a score for each intron. We fitted the read gradient (slope) along each of the selected introns (5′→3′; see above for the filtering criteria). Note that the intron gradient is not influenced by exonucleolytic degradation of excised intron lariats[20,78]

and that this measure is only weakly associated with the expression level of the transcript[20] (Supplementary Table 2).

To transform slopes to Pol II elongation speed, we used the following formalism. We assume an intron of length $L$ and we assume that at steady state, a constant number of polymerases is initiating and the same number of polymerases is terminating at the end of the intron; that is, we assume that premature termination inside the intron can be ignored. Polymerases are progressing at a common speed of $k$ (base pairs per minute). The average time that it takes a polymerase to traverse the whole intron is hence:

$$\Delta t = \frac{L}{k}$$

Transcription is initiated at a rate of $n$ polymerases per unit time (1/minute). Hence, the number of polymerases $N$ initiating during $\Delta t$ is:

$$N = \Delta t \times n$$

The slope $s$ is the number of transcripts after the distance $L$ minus the number of transcripts at the beginning divided by the length of the intron:

$$s = \frac{0 - N}{L} = \frac{-\Delta t \times n}{L} = \frac{\frac{-L}{k} \cdot n}{L} = \frac{-n}{k}$$

and thus, the speed $k$ can be computed from the slope as:

$$k = \frac{-n}{s}$$

Hence, slope and speed are inversely related and the speed depends also on the initiation rate (that is, the expression rate). However, we observed empirically only a small dependency between expression and slope[20] (Supplementary Table 2).

To validate our estimates of Pol II speeds, we compared our data with experimental values estimated via GRO-seq[19] and tiling microarray data[15]. There was a significant correlation (GRO-seq: $R = 0.38$, $P = 4 \times 10^{-5}$, compared with time point 25–50 min (see Jonkers et al.[19]); tilling array: $R = 0.99$, $P \leq 2.6 \times 10^{-16}$ (data not shown)) between our data and experimentally measured transcriptional elongation values. We noted that our Pol II speed estimates for different introns of the same gene were more similar than Pol II speed estimates for random pairs of introns, implying that gene-specific factors or local chromatin structure influence Pol II speed (Extended Data Fig. 2b).

### 4sUDRB labelling, TUC conversion and elongation rate calculation

The estimation of transcription elongation speed using RNA labelling was based on the measurement of nucleotides added per time unit in a newly synthesized nascent transcript. First, transcription was reversibly inhibited by DRB to achieve accumulation of Pol II at the transcription start sites and synchronized transcriptional elongation initiation upon DRB removal. Simultaneously with the DRB removal, cells were pulsed for different time points with the uridine analogue 4sU to enrich for newly synthesized transcripts. Last, total RNA was isolated per each time point and the Pol II speed was determined by calculating the 4sU nucleotides added to the nascent transcript per time point. To estimate Pol II speed change in ageing cells, human fetal lung fibroblasts (IMR90) in proliferating and in senescent state were treated using this experimental procedure.

To select the time points to be used in the experiment, validate the DRB treatment and removal and check the enrichment efficiency of 4sU, a control experiment was set according to a previous protocol[79]. Two million proliferating cells (passage 14) were treated with 100 µM DRB (D1916, Merck) in their medium for 3 h at 37 °C and, upon DRB removal, they were pulsed with 1 mM 4sU (T4509, Sigma-Aldrich) for

0, 5, 15, 30, 45, 60, 90 and 120 min. Immediately after the completion of each time point, cells were lysed in TRIzol (15596018, Thermo Fisher) and RNA was isolated with the Direct-Zol RNA mini-prep kit (R2052, Zymo Research). To validate DRB treatment, quantitative PCR with reverse transcription was performed in cDNA from all time points using the primers designed by Fuchs et al.[79] in proximal and distant introns of the *OPA1* gene. Furthermore, to estimate 4sU enrichment, the RNA collected in each time point was biotinylated using the EZ-Link biotin HPDP kit (21341, Thermo Fisher) and biotinylated RNA was enriched with streptavidin-coated beads (DYNAL Dynabeads M-280 streptavidin; 11205D, Thermo Fisher). Evaluation of quantitative PCR with reverse transcription was performed also with the primers suggested by Fuchs et al.[79] against TTC-17 nascent and mature mRNA and 18S rRNA.

For the actual experiment, we performed the thiouridine-to-cytidine conversion sequencing (TUC-seq) protocol developed by Lusser et al.[80] to detect the 4sU-labelled transcripts in different time points. In this method, the thiol group of 4sU is quantitatively converted to cytidine via oxidation by $OsO_4$ in aqueous $NH_4Cl$ solution. The $OsO_4$-treated RNA samples are submitted to RNA-seq to quantify labelled and non-labelled transcripts and define the number of reads containing uridine-to-cytidine conversions. To this aspect, 9 million proliferating (passage 9) cells and 9 million cells that had entered senescence (passage 35) were treated with 100 µM DRB for 3 h at 37 °C. Immediately after DRB removal, cells were pulsed with 1 mM 4sU for 0, 5, 15, 30 and 45 min. RNA was isolated manually according to the TRIzol protocol and treated with 40 units of DNAse I (E1010, Zymo Research) for 20 min at room temperature. RNA was purified with the RNA Clean & Concentrator-25 kit (R1018, Zymo Research) and quantified using a NanoDrop spectrophotometer. For the TUC conversion, 10 µg of 4sU-labelled RNA was treated with 1.43 mM $OsO_4$ (251755, Merck) in 180 mM $NH_4Cl$ (09718, Merck) solution pH 8.88 for 3 h at 40 °C as described by Lusser et al.[80]. Subsequent sample concentration and purification were also done according to this protocol. 4sU-labelled and $OsO_4$-treated RNA samples derived from proliferating and senescent IMR90 cells in all five time points were subjected to RNA-seq. As a negative control for the TUC conversion, we used a mixture 1:1 of 4sU-labelled but not $OsO_4$-treated samples from the time points 30 min and 45 min. The RNA-seq was performed in two biological replicates per condition.

Detection of labelled transcripts was performed based on the Lusser protocol, modified for Illumina RNA-seq:
(1) FASTQ files were aligned to the genome using STAR to produce BAM files.
(2) Sam2tsv[81] was then used to identify single-nucleotide mismatches.
(3) A custom R script was used to count the number of A-G or T-C mismatches per read.

Only read pairs with at least three A-G or T-C mismatches were assumed to be 4sU-labelled and thus retained for subsequent analyses. Because the 5-min samples contained a very low number of reads with conversions, they were discarded from the rest of the analysis. We used two approaches for estimating the elongation rate per gene from the 4sU-labelling data. For the first approach, we tracked the progress of Pol II complexes constructing single-gene coverage profiles using 4sU-labelled reads. Progression was determined by picking the 99th percentile of gene body coverage in each sample to determine the front of elongating RNA polymerases. (We did not use the last converted read to determine the front because this measure would be too sensitive to noise in the data.) Elongation rates were calculated by fitting a linear model on the front positions of Pol II in 0, 15, 30 and 45 min in the first 100 kb of each gene. To determine elongation rates with greater accuracy, we filtered out genes with a length of less than 100 kb, as short genes can be fully transcribed in less than 45 min or even 30 min. This first approach of estimating Pol II speeds is characterized by high accuracy, but is limited to genes longer than 100 kb. The data in Fig. 1e are based on this approach.

The direct comparison of the 4sU data to the approach using read-coverage slopes in introns required a large set of genes for which Pol II speed could be measured using both assays. To maximize this gene set, we devised a second alternative approach for deriving speed from 4sU-labelling data that is applicable to shorter genes. For this second approach, we measured the front position of the polymerase in the same way as before (using the 99th percentile) but across the whole gene. For genes 30–100 kb long, we calculated the elongation rate from the difference in the front positions of the polymerase at 15 min and 30 min and divided this distance by 15 min to obtain speed measures per minute. For genes more than 100 kb long, we calculated the elongation rate from the difference in the positions of the polymerase at 30 min and 45 min divided by 15 min. This second speed measure is less accurate than the first measure, because it uses only two time points per gene; however, it enables estimating speed for genes shorter than 100 kb. The data in Fig. 1d are based on this second measure. Note that both measures confirmed the increase in average Pol II elongation speed from proliferating to senescent IMR90 cells.

### [35]S-methionine or [35]S-cysteine incorporation to measure translation rates in *Drosophila*

Ex vivo incorporation of radio-labelled amino acids in fly heads was performed as previously described[82]. In brief, 25 heads of each young (10 days) and old (50 days) wDah control and *RpII215^C4*-mutant animals were dissected in replicates of five and collected in DMEM (#41965-047, Gibco) without supplements, at room temperature. For labelling, DMEM was replaced with methionine and cysteine-free DMEM (#21-013-24, Gibco), supplemented with [35]S-labelled methionine and cysteine (#NEG772, Perkin-Elmer). Samples were incubated for 60 min at room temperature on a shaking platform, then washed with ice-cold PBS and lysed in RIPA buffer (150 mM sodium chloride, 1.0% NP-40, 0.5% sodium deoxycholate, 0.1% SDS and 50 mM Tris, pH 8.0) using a pestle gun (VWR). Lysates were centrifuged at 13,000 rpm at 4 °C for 10 min and protein was precipitated by adding 1 volume of 20% TCA, incubating for 15 min on ice and centrifuging at 13,000 rpm at 4 °C for 15 min. The pellet was washed twice in acetone and resuspended in 200 µl of 4 M guanine-HCl. Of the sample, 100 µl was added to 10 ml scintillation fluid (Ultima Gold, Perkin-Elmer) and counted for 5 min per sample in a scintillation counter (Perkin-Elmer). Protein determination was done in duplicates (25 µl each) per sample using the Pierce BCA assay kit (Thermo Fisher Scientific). Scintillation counts were normalized to total protein content.

### Mismatch detection

Mismatch detection was performed using the tool rnaseqmut (https://github.com/davidliwei/rnaseqmut), which detects mutations from the NM tag of BAM files. To avoid detection of RNA editing or DNA damage-based events, we only considered genomic positions with only one mismatch detected (that is, occurring in only one single read). Reads with indels were excluded and only mismatches with a distance of more than four from the beginning and the end of the read were considered. A coverage-level filter was applied so that only bases covered by at least 100 reads were kept. A substantial number of mismatches may result from technical sequencing errors. However, as young and old samples were always handled together in the same batch, we can exclude that consistent differences in the number of mismatches are due to technical biases. The fraction of RNA editing events is generally relatively low and not expected to globally increase with age[83].

### MNase-seq sample preparation

Mononucleosomal DNA from proliferating and senescent IMR90 cells (from two donors) were prepared and sequenced on an Illumina HiSeq4000 platform as previously described[84]. For fly heads, a MNase digestion assay was performed using the EZ nucleosomal DNA prep kit, as per the manufacturer's guidelines (Zymo Research). In brief,

25 snap-frozen heads were lysed in nuclei prep buffer and incubated on ice for 5 min. Cuticle fragments were then removed via centrifugation (at 50*g* for 30 s). Nuclei were pelleted (for 500*g* at 5 min) and washed twice in digestion buffer and resuspended in 100 µl of digestion buffer. Nucleosome footprints were then digested using 0.05 U of MNase (Zymo Research). Samples were taken at 0, 2, 3 and 5 min or 10 min for prolonged digestion, and immediately stopped in MN stop buffer (Zymo Research). Samples were isolated using Zymo Spin IIC columns. Nucleosome footprints (1:10 dilution) were visualized by Tapestation using High-sensitivity D1000 ScreenTape (Agilent).

### MNase-seq analysis

We used nucleR[85] with default parameters to calculate peak sharpness as a combination of peak width and peak height. Peak width was quantified as the standard deviation around the peak centre, and peak height was quantified as the number of reads covering each peak[85] and the distance between peak summits. Intron and exon annotations were downloaded from UCSC table utilities[74] and filtered as described in definition of intronic regions. Nucleosome density (Fig. 5a) is defined as the number of nucleosome peaks found within an exon or an intron divided by the length of the exon or intron.

### Western blotting

Western blots were carried out on protein extracts of individual dissected tissues. Proteins were quantified using BCA (Pierce). Equal amounts were loaded on Any-KD pre-stained SDS–PAGE gels (Bio-Rad) and blotted according to standard protocols. Antibody dilutions varied depending on the antibody and are listed here: histone H3 (1:1,000) and HP1 (DSHB) (1:500). Appropriate secondary antibodies conjugated to horseradish peroxidase were used at a dilution of 1:10,000.

### Inducible histone overexpression

Doxycycline-inducible expression of histones H3 and H4 in proliferating human fetal lung fibroblasts (IMR90) was achieved using the PiggyBac transposition system[86]. The open reading frames of H3 and H4 were cloned in the doxycycline-inducible expression vector KA0717 (KA0717_pPB-hCMV*1-cHA-IRESVenus was a gift from H. Schöler, Addgene plasmid #124168; http://n2t.net/addgene:124168; RRID:Addgene_124168) fused at their 3' end in-frame to the sequence of the YFP mVenus[87]. After sequencing validation, each construct was co-transfected in IMR90 cells with the transactivator plasmid KA0637 (KA0637_pPBCAG-rtTAM2-IN was a gift from H. Schöler, Addgene plasmid #124166; http://n2t.net/addgene:124166; RRID:Addgene_124166) and the Super piggyBac transposase expression vector (PB200PA-1, SBI System Biosciences) using the Lipofectamine LTX reagent with PLUS reagent (15338100, Thermo Fisher Scientific) according to the manufacturer's instruction. In total, 2.5 µg of the vectors KA0717, KA0637 and PB200PA-1 were used for each transfection in a 10:3:1 ratio. Stable transgene-positive cells were selected using 250 µg ml⁻¹ G418 (resistance gene carried in KA0637) for 7 days. Emerging cells were induced for 24 h with 2.5 µg ml⁻¹ doxycycline and then subjected to FACS to select the cells expressing mVenus (BD FACSAria II, BD Biosciences). H3 and H4 overexpression was verified by western blot with anti-H3 and anti-H4 antibodies (ab1791 and ab10158, Abcam, respectively). All further assays were repeated in proliferating cells and cells at the senescence entry state. Senescence state was monitored by β-galactosidase staining[88] in different passages (Senescence β-Galactosidase Staining Kit, 9860, Cell Signaling Technology). Immunofluorescence stainings to detect HMGB1, p21 and HMGB2 (ab18256, ab184640 and ab67282, Abcam, respectively) were performed as previously described[89] and images were acquired in a widefield Leica DMi8 S with an HCX PL APO ×63/1.40 (Oil) objective. For MTT assays[90], 6,000 cells of each condition were seeded per well in a 96-well plate in four replicates, incubated for 4 h at 37 °C after the addition of 1 mM MTT (3-(4,5-dimethylthiazol-2-yl)-2,5-diphenyltetrazolium bromide; M6494, Thermo Fisher), treated with

DMSO for 10 min at 37 °C, and finally their absorbance was measured at 540 nM using an Infinite 200 PRO plate reader (Tecan). For RNA-seq, 1 million cells of each condition were lysed in TRIzol (15596018, Thermo Fisher) and RNA was isolated with the Direct-Zol RNA mini-prep kit (R2052, Zymo Research). Elongation rates for wild-type and mutant samples were calculated as described in the section 'Transcriptional elongation speed based on intronic read distribution'.

## Eukaryotic cell lines

HUVECs from individual healthy donors were purchased by Lonza; human primary lung fibroblasts (IMR90) from two different isolates were obtained via the Coriell repository. All of these lines were biannually checked for mycoplasma contamination and tested negative.

## Animal strains used and animal ethics

***M. musculus.*** Female F1 hybrid mice (C3B6F1) were generated in-house by crossing C3H/HeOuJ females with C57BL/6 NCrl males (strain codes 626 and 027, respectively, Charles River Laboratories). The DR study involving live mice was performed in accordance with the recommendations and guideline of the Federation of the European Laboratory Animal Science Association (EU directive 86/609/EEC), with all protocols approved by the Landesamt für Natur, Umwelt und Verbraucherschutz, Nordrhein-Westfalen (LANUV), Germany (reference numbers: 8.87-50.10.37.09.176, 84-02.04.2015.A437 and 84-02.04.2013.A158) and the Netherlands (IACUC in Bilthoven, NIH/NIA 1PO1 AG 17242).

***D. melanogaster.*** Wild-type, *RpII215* (RRID:BDSC_3663) mutant flies and *Repo-Gal4* flies were obtained from the Bloomington *Drosophila* Stock Center (NIH P40OD018537). The *RpII215* allele and *Repo-Gal4* (ref. 91) were backcrossed for six generations into the outbred wDah wild-type background generating the wDah, *RpII215* stock, which was used for experiments. wDah, *dilp2-3,5* flies (RRID:BDSC_30889) and *UAS-Histone 3* (*UAS-H3*) were previously generated in the laboratory[62] and backcrossed for six generations into the outbred wDah wild-type background. Female flies were used for all experiments.

*C. elegans* strains used: AA4274 *ama-1*(m322), *ama-1*(syb2315), CB1370 *daf-2*(e1370), N2 wild type.

## Human research participants and human ethics

Participants were searched using bulletins in which healthy individuals interested in taking part in a study examining ageing-related changes were asked to contact Department 2 of Internal Medicine (University of Cologne) by telephone. A trained employee ruled out relevant pre-existing diseases using a structured questionnaire. The test participants were then invited for an appointment at the University Hospital Cologne to obtain the blood samples used for sequencing in the study at hand. Ethics were reviewed by the Institutional Review Board, Medical Faculty, University of Cologne. Permission was granted on 17 January 2018 (proposal ID: 17-362). The study was registered to the 'Deutsches Register Klinischer Studien (DRKS)' (DRKS00014637).

## Reporting summary

Further information on research design is available in the Nature Portfolio Reporting Summary linked to this article.

## Data availability

The RNA-seq data used in this study are available at the GEO database under accession numbers GSE102537 and GSE92486.

## Code availability

Standard procedures and published codes were used for data analysis (see Methods). The custom code created for processing the data is available at https://github.com/beyergroup/ElongationRate. For the

figures, the silhouettes of species were downloaded from PhyloPic (https://phylopic.org) and the silhouettes of organs were created using BioRender (https://biorender.com). Figures were generated using Adobe Illustrator and Inkscape.

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

**Acknowledgements** We acknowledge the Bloomington *Drosophila* Stock Center for fly stocks; the Max Planck-Genome-Centre Cologne and the Cologne Center for Genomics for library preparation and sequencing; M. Dollé for providing mice used in the ageing kidney experiment; and H. Schöler for sharing cell lines. We acknowledge funding from the Max

Planck Society to L.P. and A.A., Bundesministerium für Bildung und Forschung grant SyBACol to T.B., A.A., A.B. and L.P. (grant no. 0315893A-B) and the European Research Council under the European Union's Seventh Framework Programme (FP7/2007-2013)/European Research Council to L.P. (grant no. 268739) and A.A. (grant no. 834259). We acknowledge support by the Deutsche Forschungsgemeinschaft (DFG) under Germany's Excellence Strategy—CECAD, EXC 2030—390661388 to A.A., A.B., L.P., R.-U.M. and T.B. A. Papantonis received funding from the DFG via SPP1935 (project 313408820) and basic module grants (290613333 and 285697699). R.-U.M. received funding from the Ministry of Science North Rhine-Westphalia (Nachwuchsgruppen.NRW) and the DFG (MU 3629/6-1 and MU 3629/2-1).

**Author contributions** A.A., L.P., A. Papantonis and A.B. conceived and organized the study based on the original design by A.B. C.D., A. Papadakis, A.A., L.P., A. Papantonis and A.B. wrote the manuscript, with input from all authors. A.B., A. Papantonis, A.A., L.P., R.-U.M., T.B. and B.S. supervised the work, contributed to the data interpretation and provided financial resources. O.K., S.N., W.H. and R.E. designed and performed the worm experiments, and collected and analysed the data. S.G. and L.S.T. designed and performed the fly experiments, and collected and analysed the data. S.G. designed and performed the mouse experiments, and collected and analysed the data. O.H. provided and analysed the mouse sequencing data. C.W. provided the fly sequencing data. A.M., N.J., A.Z., K.S. and M.L. performed experiments, and provided and analysed data from human cells. Y.-X.L. provided reagents. T.K. provided mouse tissue. M.R.S. set up the clinical study to provide blood samples of human individuals. C.D. and A. Papadakis analysed the RNA-seq and MNase-seq data, and performed gene set enrichment analysis. A. Papadakis developed the methods for quantifying Pol II elongation speed based on RNA-labelling data. I.B. contributed to the analysis of splicing variation.

**Funding** Open access funding provided by Universität zu Köln.

**Additional information**
**Correspondence and requests for materials** should be addressed to Adam Antebi, Linda Partridge, Argyris Papantonis or Andreas Beyer.
Competing interests The authors declare no competing interests.

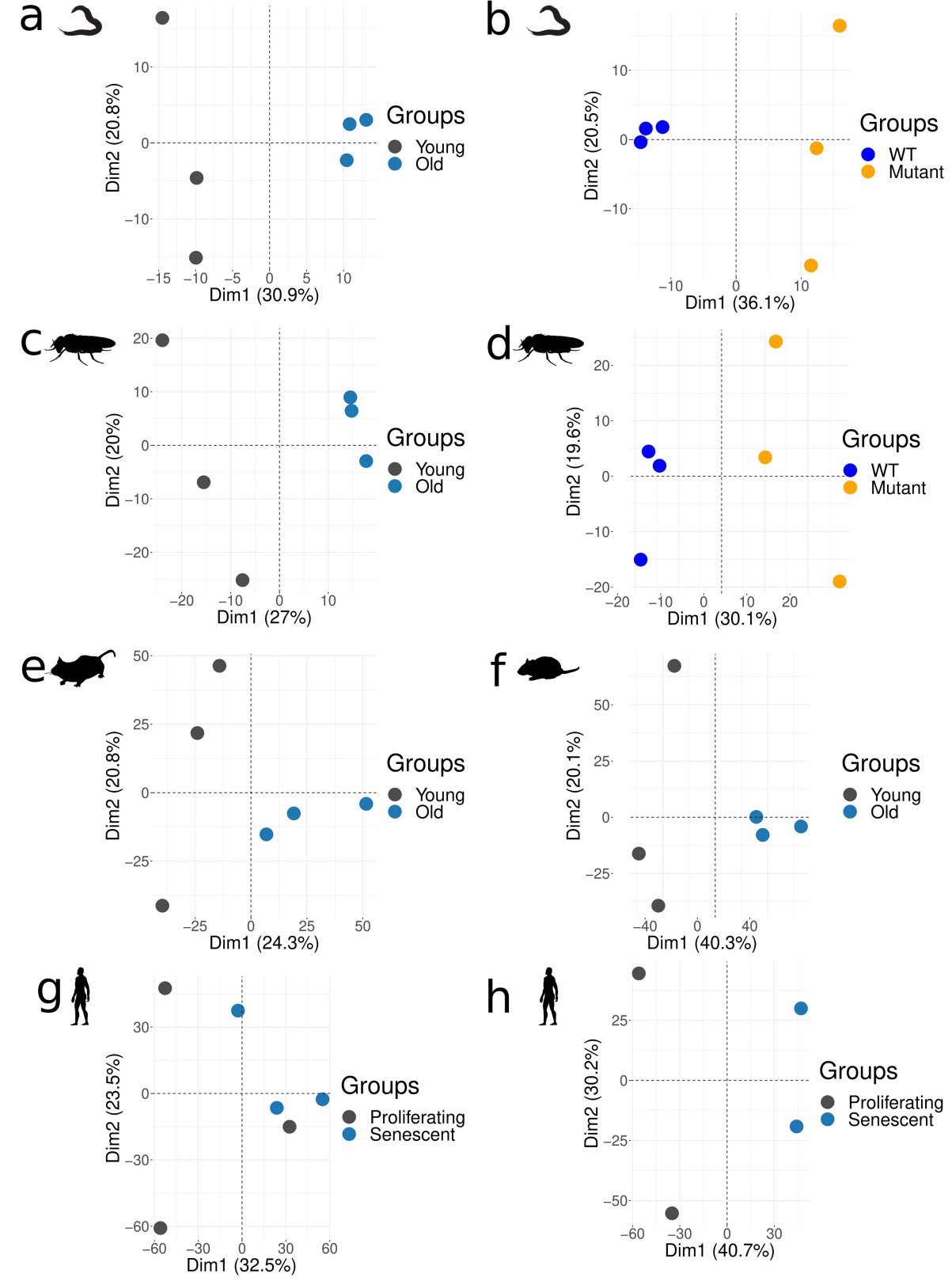

**Extended Data Fig. 1 | PCAs of slopes of intronic read distribution.** Principal component analysis (PCA) of the slopes of *C. elegans* ((**a**) wt 21 d vs 1 d; (**b**) 14 *ama-1(m322)* d vs wt 14 d), *D. melanogaster* ((**c**) wt heads 50 d vs 10 d, (**d**) RpII215[4] heads 50 d vs wt 50 d), *M. musculus* ((**e**) kidney: 24 mo vs 3 mo), *R. norvegicus* ((**f**) liver: 24 mo vs 6 mo), *H. sapiens* ((**g**) HUVEC and (**h**) IMR90: Senescent vs Proliferating). In all panels, the silhouettes of species are from PhyloPic (https://phylopic.org).

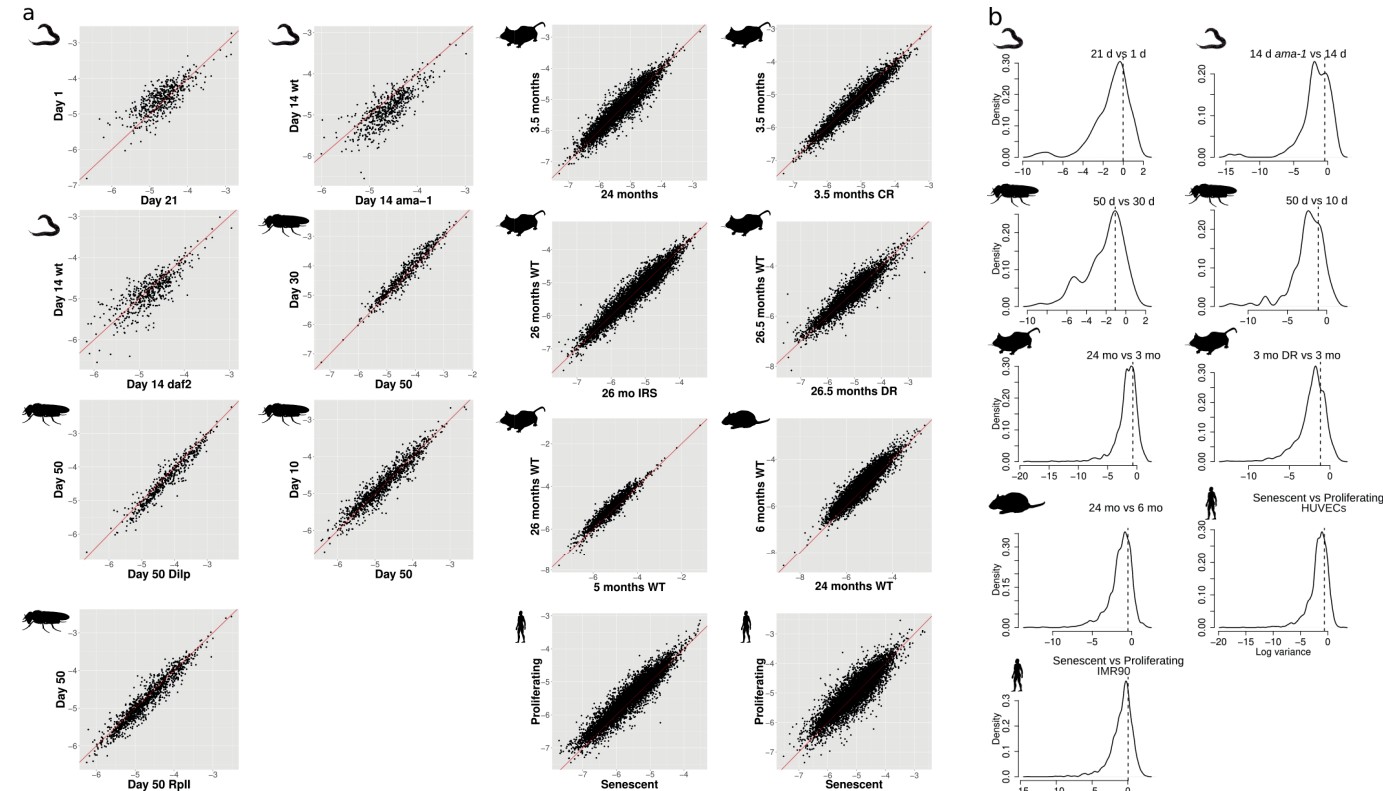

**Extended Data Fig. 2 | Consistency of Pol-II elongation speed estimates between samples and within genes.** (**a**) Scatterplots of intronic slope (−log10) for each condition and species (*C. elegans, D. melanogaster, M. musculus, R. norvegicus, H. sapiens*). (**b**) Variation of intronic slope (−log10) changes for different introns of the same gene. Distribution of variances of Pol-II speed estimates (slope per intron) for introns within the same gene. Average variance of speed estimates across all introns (i.e. between genes; global average) is shown as a dashed vertical line for *C. elegans* (21 d vs 1 d; 14 *daf-2* d vs 14 d),

*D. melanogaster* (heads 50 d vs 30 d; 50 d vs 10 d*), M. musculus* (kidney: 24 mo vs 3 mo; 3 DR mo vs 3 mo), *R. norvegicus* (liver: 24 mo vs 6 mo), *H. sapiens* (Umbilical vein endothelial (HUVECs); fibroblast fetal lung (IMR90): Senescent vs Proliferating). The vast majority of intra-gene variances are below the average inter-gene variance, suggesting that introns of the same gene have coupled Pol-II elongation speeds. In all panels, the silhouettes of species are from PhyloPic (https://phylopic.org).

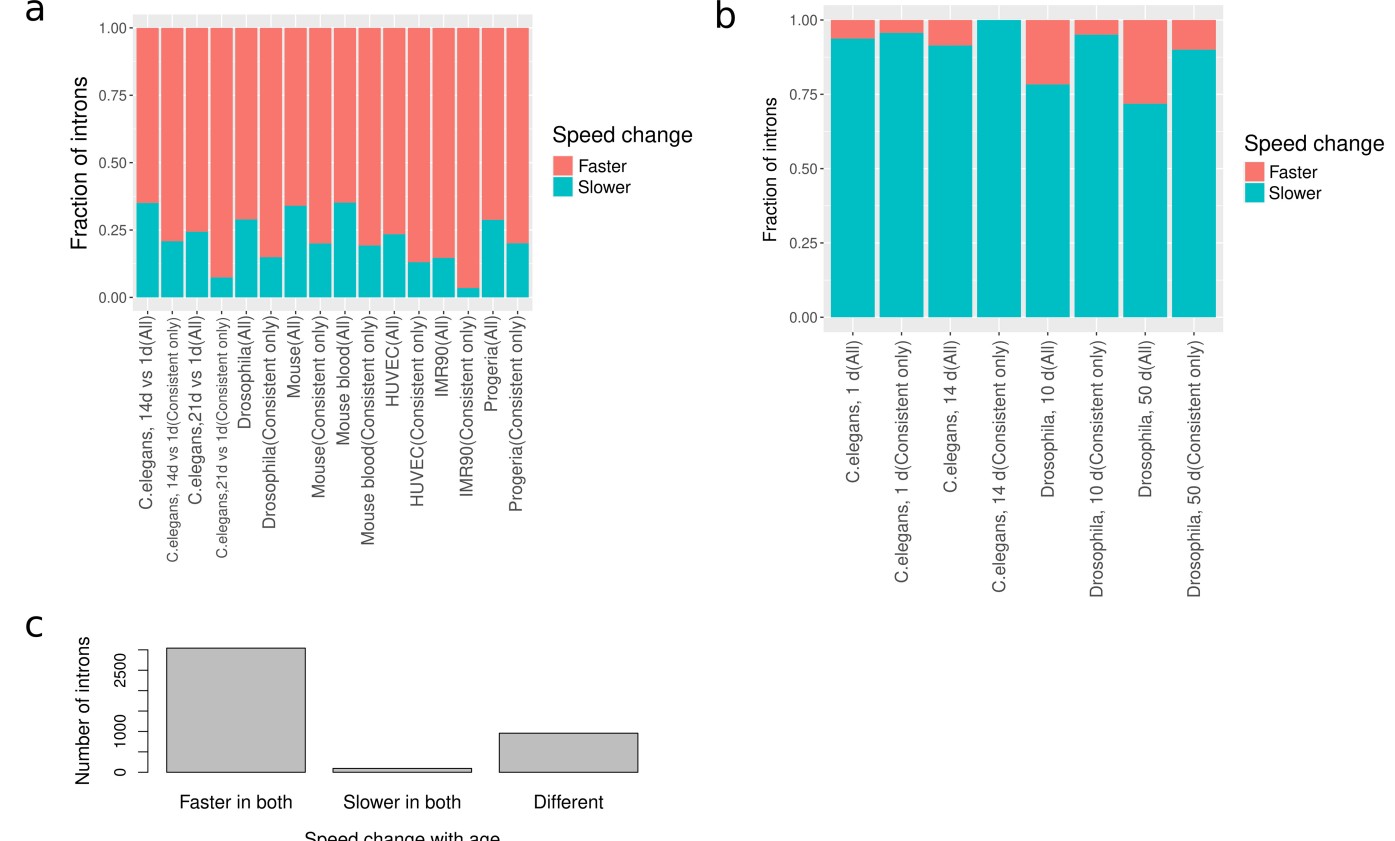

**Extended Data Fig. 3 | Consistency of RNA Pol-II speed changes.** (**a**) Change of elongation rate with aging or senescence in introns of *C. elegans*, *D. melanogaster*, *M. musculus* and *H. sapiens*, before and after filtering for introns that consistently change in speed in all replicates. (**b**) Change of elongation rate with mutations that slow down the speed of RNA-Pol-II in introns of *C. elegans* and *D. melanogaster* before and after filtering for introns that consistently change in speed in all replicates. (**c**) Comparison of the change of elongation rate with aging between IMR90 and HUVEC using the same introns.

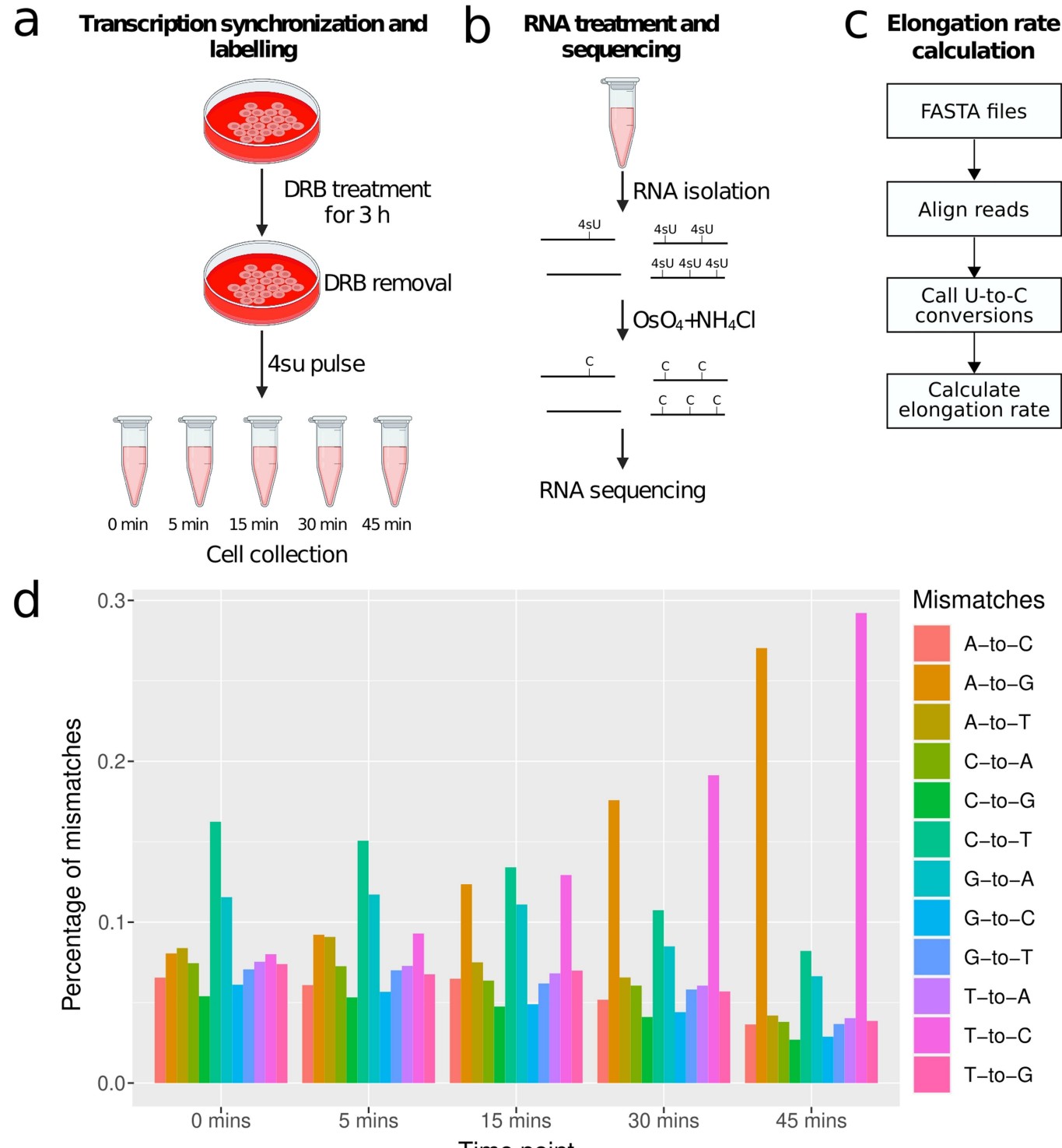

**Extended Data Fig. 4 | 4SU-DRB labelling and TUC conversion to calculate RNA-Pol-II elongation rate.** (**a**–**c**) Schematic representation of the 4SU-DRB labelling (**a**), TUC conversion (**b**) and elongation rate calculation (**c**). Cell culture and eppendorf icons created with BioRender.com. (d) Percentage of mismatches in every time point of the experiment (0 mins, 15 mins, 30 mins, 45 mins) in one of the proliferating replicates. There is a noticeable increase in A-to-G and T-to-C mismatches in the last two time points.

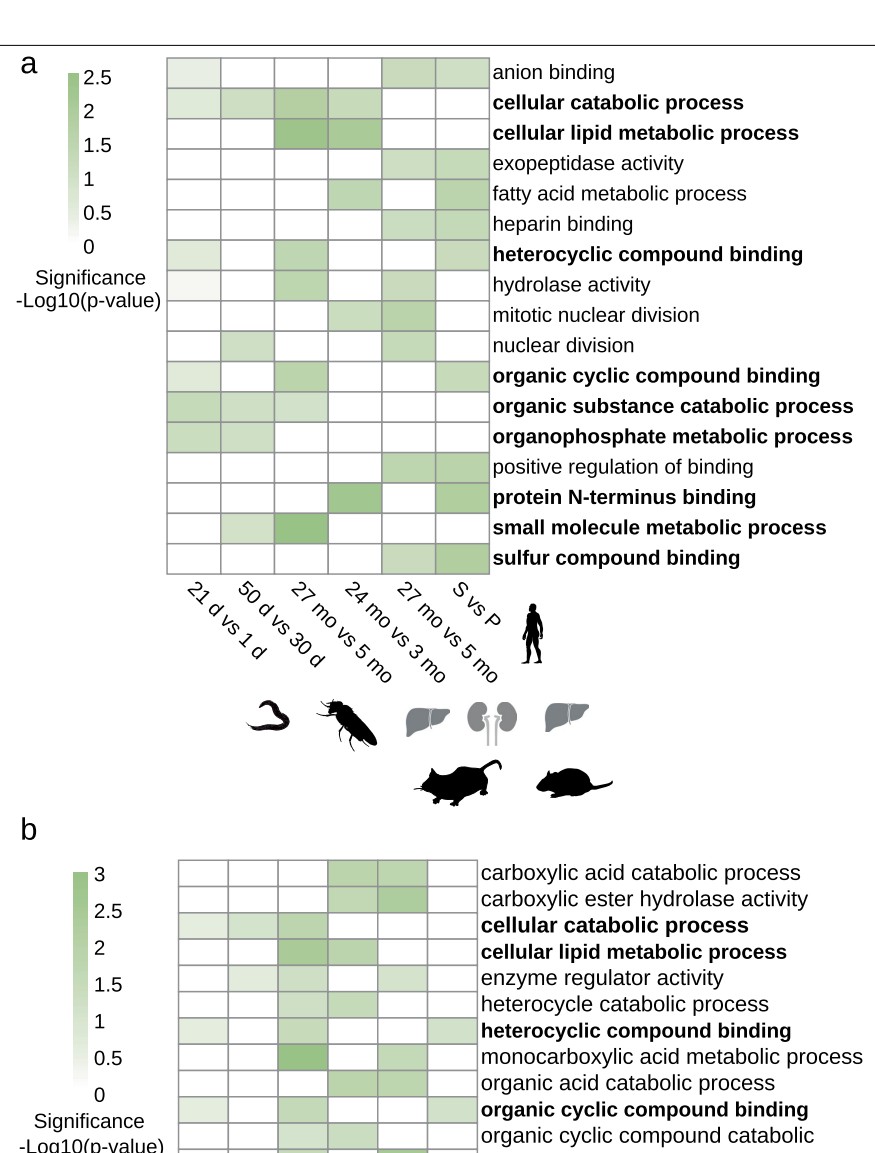

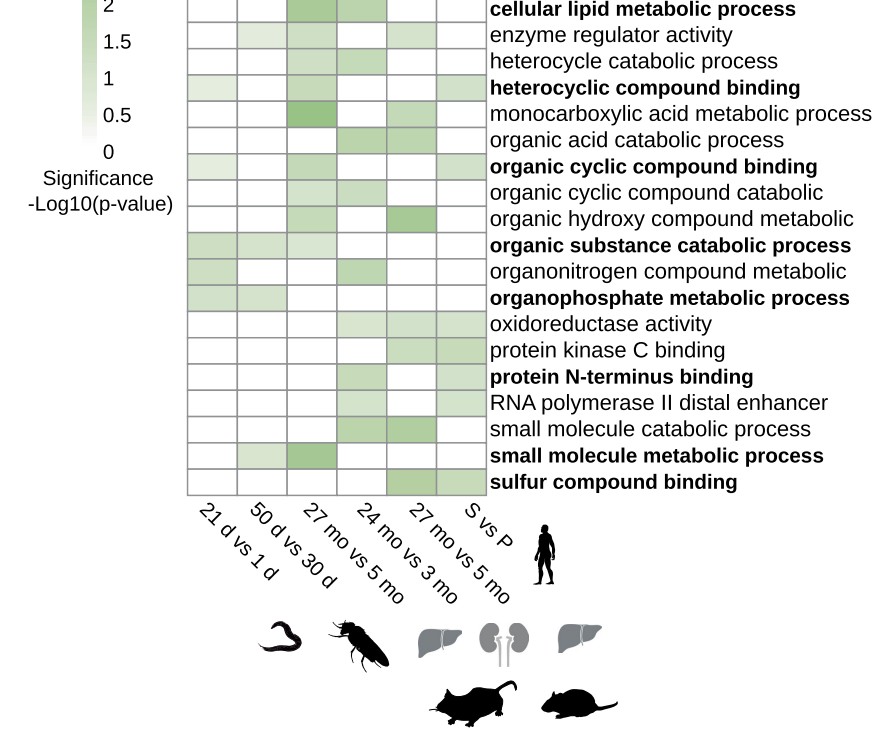

**Extended Data Fig. 5 | Genes with increase in Pol-II speed are associated with metabolism and catabolism related pathways.** GO enrichment analysis of genes with increased Pol-II speed across species: *C. elegans* (21 d vs 1 d), *D. melanogaster* (heads: 50 d vs 30 d), *M. musculus* (kidney: 24 mo vs 3 mo), *R. norvegicus* (liver: 24 mo vs 6 mo), *H. sapiens* (IMR90: Senescent vs Proliferating). GO enrichment of (**a**), top 200 (**b**), top 300 genes with an increase in Pol-II speed change for each species (common terms between the two sets in bold). Color scale indicates the significance of the enrichment (all GO terms enriched with p-values below 0.05, with at least 10 significant genes for each GO categories, Fisher elim test). In all panels, the silhouettes of the organs were created using BioRender (https://biorender.com), and the silhouettes of species are from PhyloPic (https://phylopic.org).

# REGULATION OF DNA TEMPLATED TRANSCRIPTION ELONGATION

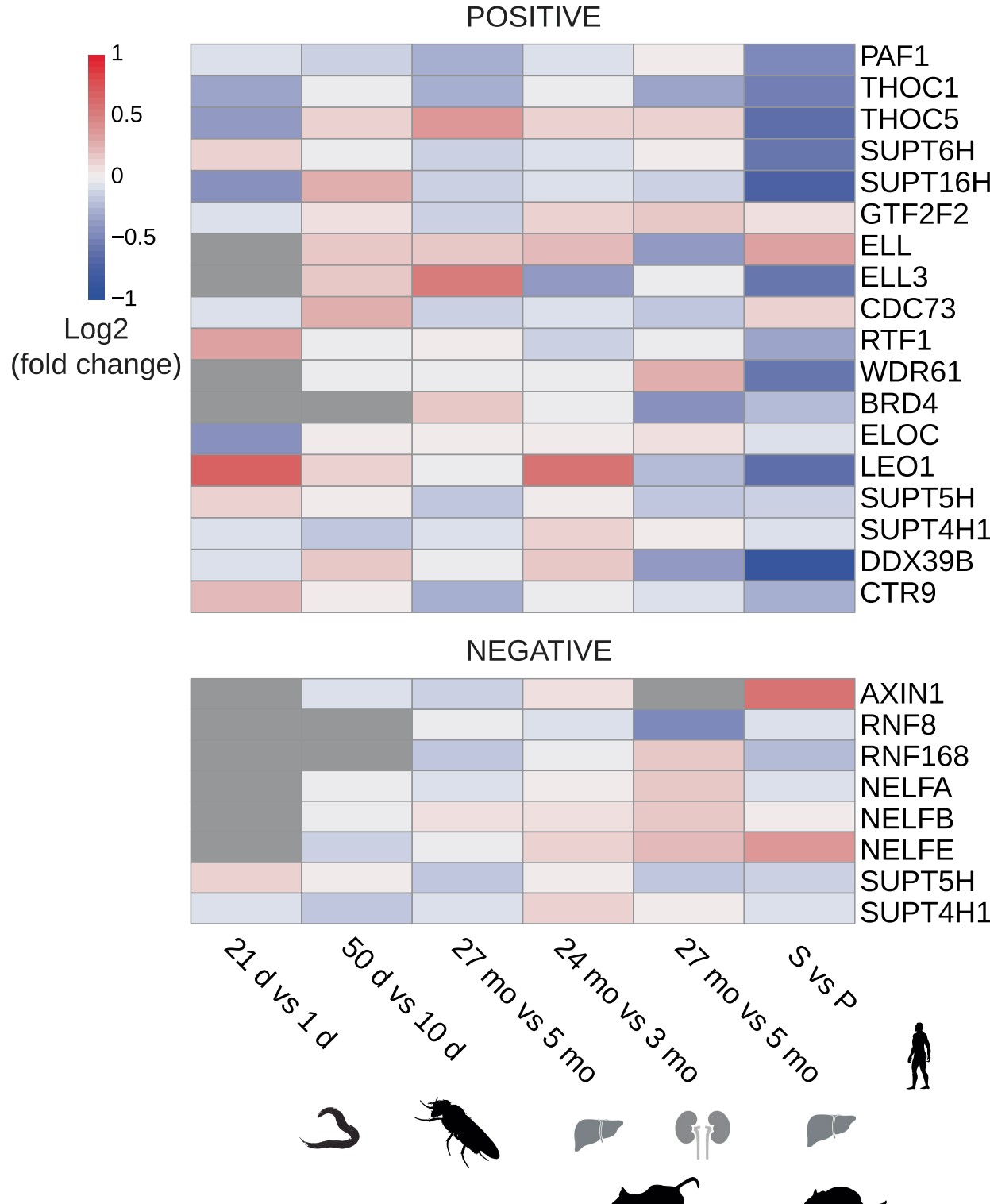

**Extended Data Fig. 6 | Heatmap of differential expression (log₂ fold change) of MSigDB (61) annotated genes for 'regulation of DNA templated transcriptional elongation'.** Top: activators of transcriptional elongation (POSITIVE); Bottom: repressors of transcriptional elongation (NEGATIVE). Data shown for WT aging time courses: worm (21 d vs 1 d), fly heads (50 d vs 10 d), mouse liver (27 mo vs 5 mo), mouse kidneys (24 mo vs 3 mo) and human fibroblast cell line (IMR90: Senescent vs proliferating). The silhouettes of the organs were created using BioRender (https://biorender.com), and the silhouettes of species are from PhyloPic (https://phylopic.org).

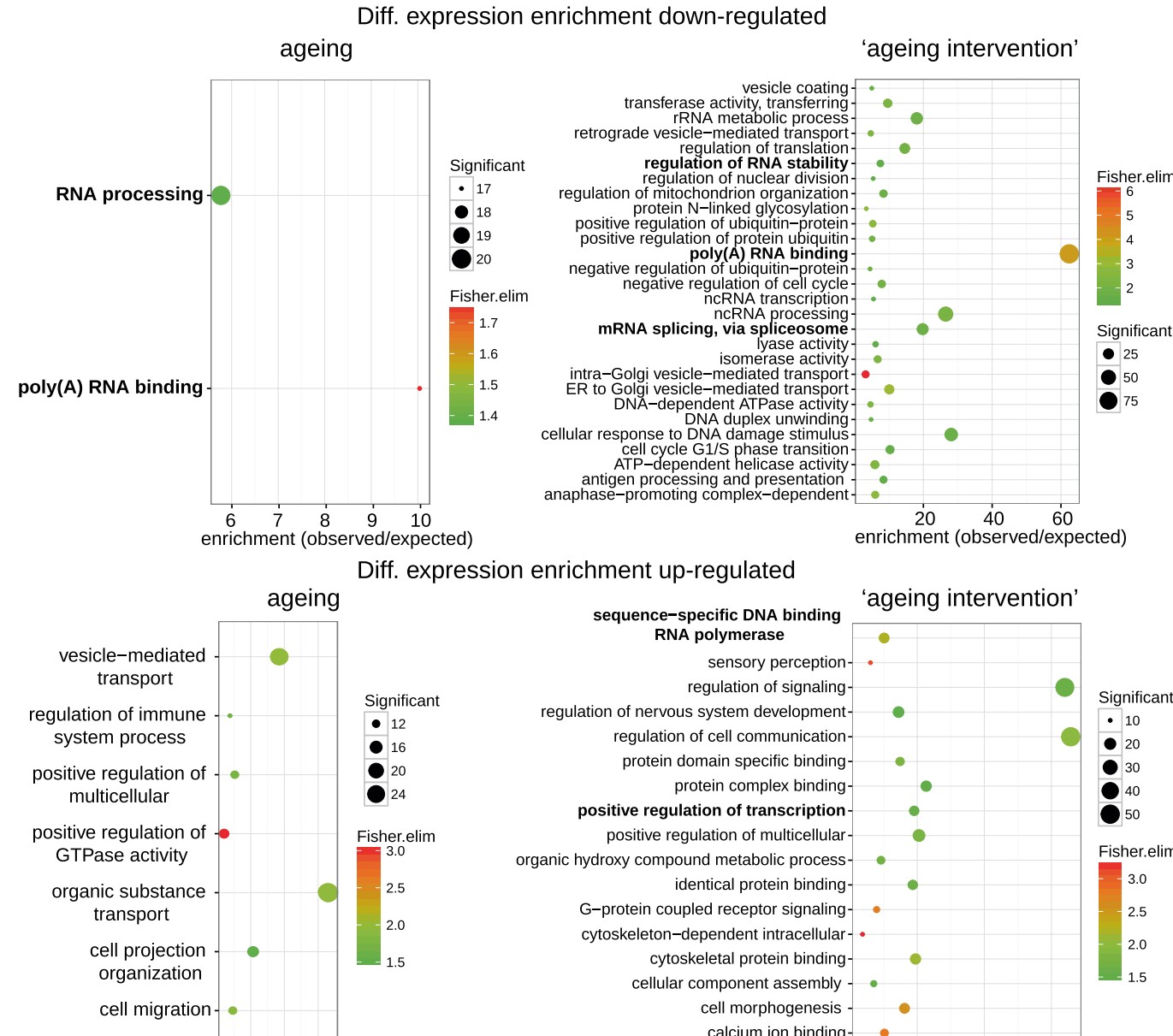

**Extended Data Fig. 7 | Functional enrichment for across-species differential expression analysis.** GO enrichment for consistently down-regulated (top) or up-regulated (bottom) genes across species during aging (left) or 'aging intervention' (right) (aging up-regulated: 92 genes; aging down-regulated: 71 genes; 'aging intervention' up-regulated: 164 genes; 'aging intervention' down-regulated: 473 genes; as background for the enrichment analysis a set of 4784 orthologue genes between *H. sapiens*, *R. norvegicus*, *M. musculus*, *D. melanogaster*, *C. elegans* was used. All p-values *P < 0.05, significant genes > 10, fisher elim test). GO terms related to transcription and splicing are indicated in bold.

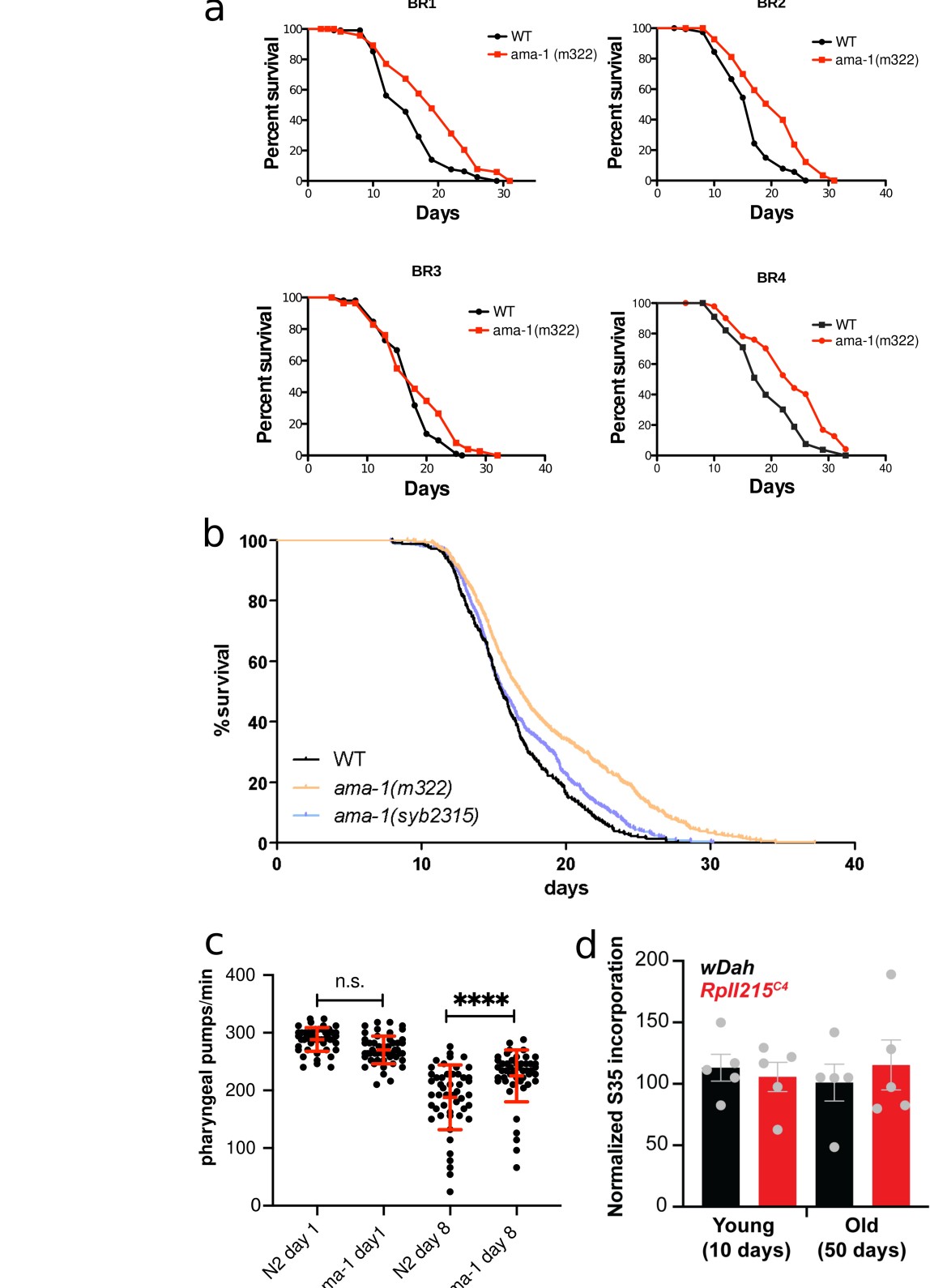

**Extended Data Fig. 8** | See next page for caption.

**Extended Data Fig. 8 | Physiological consequences of reducing Pol-II elongation speed in animal models.** (**a**) Survival of wild-type and *ama-1(m322)* mutant worms conferring a slow Pol-II elongation rate (4 replicates, BR1:1.267, P < 0,0001; BR2:1.23, P < 0.0001; BR3:1, P = 0.0342; BR4:1.263, P < 0.0001, log-rank test, Mantel-cox). (**b**) *C. elegans* lifespan analysis after CRISPR/Cas9 mediated reversion of the slow RNAPII mutation. Survival curves of the strain harbouring the slow RNAPII mutation (ama-1 m322) and wild-type controls compared to worms after CRISPR/Cas9 engineered reversion of the slow mutation back to the wild type allele (ama-1 syb2315). Animals with slow Pol-II have a significantly increased lifespan. CRISPR/Cas9 engineered reversion restored lifespan essentially back to wild-type levels. (3 replicates; n > 300 per strain). (**c**) Pumping rates of wild type N2 (day 1: 47 worms, day 8: 49 worms) and ama-1 mutant (day 1: 52 worms, day 8: 48 worms) worms were measured on day 1 and day 8. Pumping rates were not significantly different on day 1, but ama-1 worms showed higher pumping rates compared to wild types on day 8, suggesting that the mutant worms are healthier at old age. The error bars represent standard deviation. (**d**) Ex-vivo S35 incorporation assay shows no significant difference in translation rates in female fly heads between wDah control and RpII215C4 mutants both at young (10days) and old age (50 days). N = 5 biological replicates with 25 heads per replicate. The error bars show the standard error of the mean.

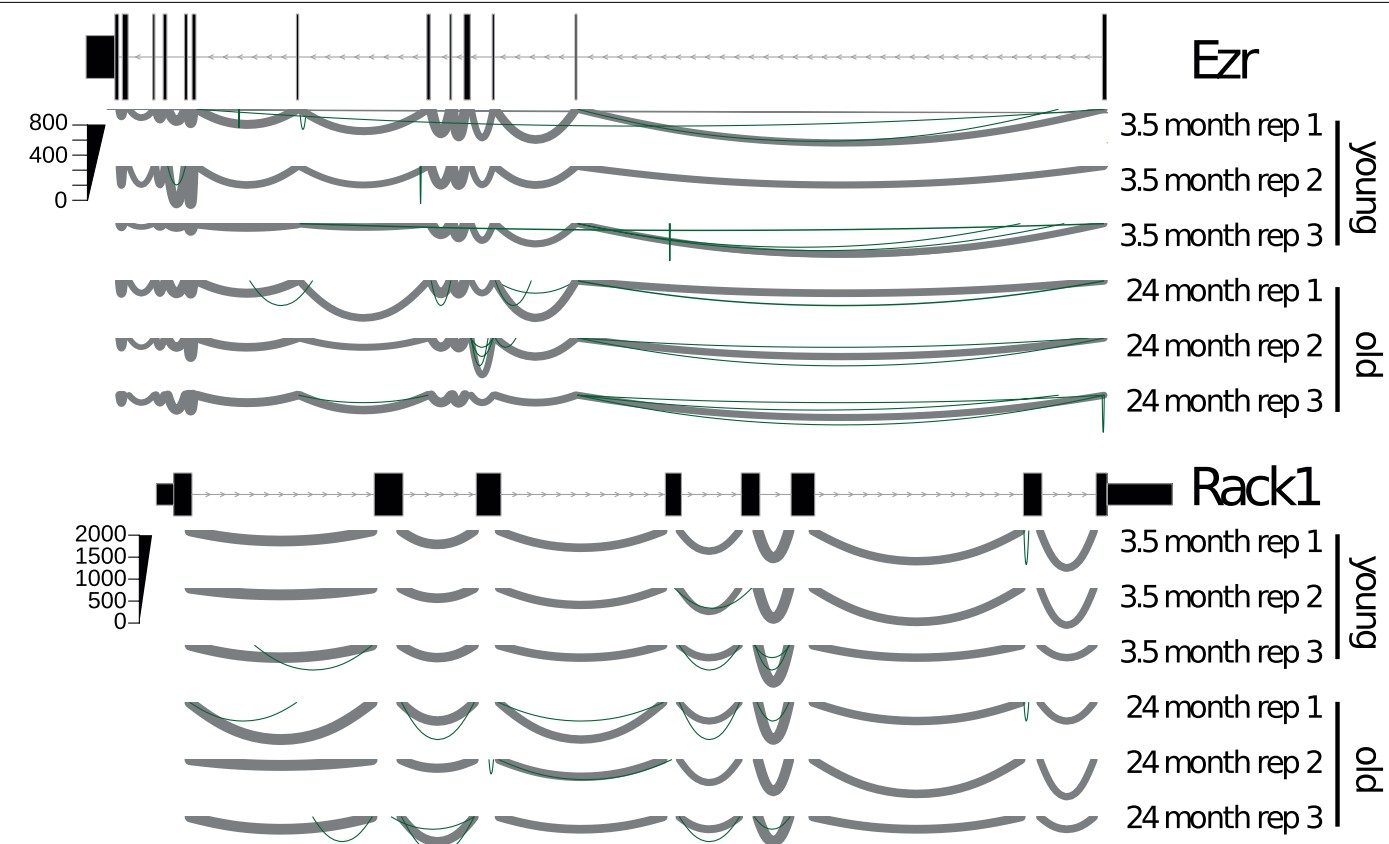

**Extended Data Fig. 9 | Examples of rare splice site changes.** for gene Ezr and Rack1 with 3 replicates young (3.5 month) and old (26 month). Line thickness encodes the number of reads supporting this junction. Rare splice sites are shown in green.

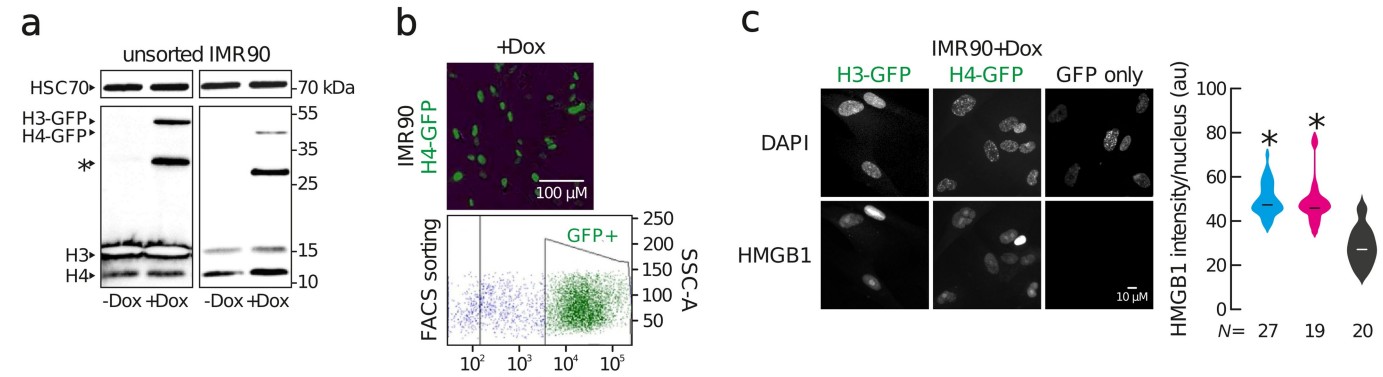

**Extended Data Fig. 10 | H3-GFP and H4-GFP overexpression in IMR90 cells.**
(**a**) Western blot experiments confirm the overexpression of the H3-GFP and H4-GFP proteins. (**b**) Visual confirmation of the Dox induction of H3/H4 expression and FACS sorting of GFP-positive cells. Both the blots and FACS data shown are representative of data obtained from independent experiments produced using the two different IMR90 isolates. (**c**) Typical immunofluorescence images of H3-GFP, H4-GFP and control IMR90 cells (left) show increased DAPI levels in histone overexpression nuclei. Violin plots (right) quantify this reduction (two-tailed t-test). N specifies the number of cells analyzed per condition.

# Reporting Summary

Nature Research wishes to improve the reproducibility of the work that we publish. This form provides structure for consistency and transparency in reporting. For further information on Nature Research policies, see Authors & Referees and the Editorial Policy Checklist.

## Statistics

For all statistical analyses, confirm that the following items are present in the figure legend, table legend, main text, or Methods section.

| n/a | Confirmed | |
|---|---|---|
| ☐ | ☒ | The exact sample size (*n*) for each experimental group/condition, given as a discrete number and unit of measurement |
| ☐ | ☒ | A statement on whether measurements were taken from distinct samples or whether the same sample was measured repeatedly |
| ☐ | ☒ | The statistical test(s) used AND whether they are one- or two-sided<br>*Only common tests should be described solely by name; describe more complex techniques in the Methods section.* |
| ☒ | ☐ | A description of all covariates tested |
| ☐ | ☒ | A description of any assumptions or corrections, such as tests of normality and adjustment for multiple comparisons |
| ☐ | ☒ | A full description of the statistical parameters including central tendency (e.g. means) or other basic estimates (e.g. regression coefficient) AND variation (e.g. standard deviation) or associated estimates of uncertainty (e.g. confidence intervals) |
| ☐ | ☒ | For null hypothesis testing, the test statistic (e.g. *F*, *t*, *r*) with confidence intervals, effect sizes, degrees of freedom and *P* value noted<br>*Give P values as exact values whenever suitable.* |
| ☒ | ☐ | For Bayesian analysis, information on the choice of priors and Markov chain Monte Carlo settings |
| ☒ | ☐ | For hierarchical and complex designs, identification of the appropriate level for tests and full reporting of outcomes |
| ☐ | ☒ | Estimates of effect sizes (e.g. Cohen's *d*, Pearson's *r*), indicating how they were calculated |

*Our web collection on statistics for biologists contains articles on many of the points above.*

## Software and code

Policy information about availability of computer code

| Data collection | Code details and version are provided in the Methods section. Only open source code was used and references are provided in the Methods section. |
|---|---|
| Data analysis | Details about the data analysis are provided in the Methods section. Only open source code was used and references are provided in the Methods section.<br>Newly developed code can be downloaded from here:<br>https://github.com/beyergroup/ElongationRate |

For manuscripts utilizing custom algorithms or software that are central to the research but not yet described in published literature, software must be made available to editors/reviewers. We strongly encourage code deposition in a community repository (e.g. GitHub). See the Nature Research guidelines for submitting code & software for further information.

## Data

Policy information about availability of data

All manuscripts must include a data availability statement. This statement should provide the following information, where applicable:
- Accession codes, unique identifiers, or web links for publicly available datasets
- A list of figures that have associated raw data
- A description of any restrictions on data availability

Data were uploaded to GEO. Reference is provided in the manuscript.

# Field-specific reporting

Please select the one below that is the best fit for your research. If you are not sure, read the appropriate sections before making your selection.

☒ Life sciences        ☐ Behavioural & social sciences        ☐ Ecological, evolutionary & environmental sciences

For a reference copy of the document with all sections, see nature.com/documents/nr-reporting-summary-flat.pdf

# Life sciences study design

All studies must disclose on these points even when the disclosure is negative.

| | |
|---|---|
| Sample size | All measurements were done in triplicates, unless stated otherwise (see Extended Data Table 2). Results were summarized across large numbers of genes/introns (see figures for details). Thus, even duplicate measurements were sufficient to gain sufficient statistical power. (P-values are provided in the manuscript.) |
| Data exclusions | No data was excluded. All data was uploaded to GEO. |
| Replication | All measurements were done in biological replicates (at least twice, mostly three times). The main conclusions were highly consistent across species and tissues, which indicates high reproducibility. The findings regarding splicing changes were reproducible across most, but not all species. We attribute this to the large variation in splicing complexity across species. (The number of alternative isoforms in mammals is vastly larger than in fly and worm.) |
| Randomization | There was no grouping, i.e. no randomization necessary. In case of the human blood samples we had the same number of male and female subjects included. |
| Blinding | Since no grouping was done, there was no need for blinding. |

# Reporting for specific materials, systems and methods

We require information from authors about some types of materials, experimental systems and methods used in many studies. Here, indicate whether each material, system or method listed is relevant to your study. If you are not sure if a list item applies to your research, read the appropriate section before selecting a response.

## Materials & experimental systems

| n/a | Involved in the study |
|---|---|
| ☒ | ☐ Antibodies |
| ☐ | ☒ Eukaryotic cell lines |
| ☒ | ☐ Palaeontology |
| ☐ | ☒ Animals and other organisms |
| ☐ | ☒ Human research participants |
| ☒ | ☐ Clinical data |

## Methods

| n/a | Involved in the study |
|---|---|
| ☒ | ☐ ChIP-seq |
| ☒ | ☐ Flow cytometry |
| ☒ | ☐ MRI-based neuroimaging |

## Eukaryotic cell lines

Policy information about cell lines

| | |
|---|---|
| Cell line source(s) | Human umbilical vein endothelial cells (HUVECs) from individual healthy donors were purchased from Lonza Inc.; human primary lung fibroblasts (IMR90) from two different isolates were obtained via the Coriell repository. |
| Authentication | These lines were authenticated by their commercial provider. |
| Mycoplasma contamination | All these lines were biannually checked for mycoplasma contamination and tested negative. |
| Commonly misidentified lines (See ICLAC register) | None identiifed. |

## Animals and other organisms

Policy information about studies involving animals; ARRIVE guidelines recommended for reporting animal research

| | |
|---|---|
| Laboratory animals | Mus musculus. Female F1 hybrid mice (C3B6F1) were generated in-house by crossing C3H/HeOuJ females with C57BL/6 NCrl males (strain codes 626 and 027, respectively, Charles River Laboratories).<br><br>Drosophila melanogaster: v[1], RpII215[4] (RRID:BDSC_3663) mutant flies were obtained from the Bloomington Drosophila Stock |

Center (NIH P40OD018537). The RpII215[4] allele was backcrossed for 6 generations into the outbred white Dah wild type background (Grönke et al., 2010) generating the w1118, RpII215[4] stock, which was used for experiments. wDah, dilp2-3,5 flies (RRID:BDSC_30889) were previously generated in the lab and backcrossed for 6 generations into the outbred white Dah wild type background (Grönke et al., 2010). Female flies were used for all experiments.

C. elegans strains used: AA4274  ama-1(m322), CB1370 daf-2(e1370), N2 wild type

| Wild animals | n/a |
|---|---|
| Field-collected samples | n/a |
| Ethics oversight | The DR study was performed in accordance with the recommendations and guidelines of the Federation of the European Laboratory Animal Science Association (FELASA), with all protocols approved by the Landesamt für Natur, Umwelt und Verbraucherschutz, Nordrhein-Westfalen, Germany (reference numbers: 8.87-50.10.37.09.176 and 84-02.04.2015.A437). |

Note that full information on the approval of the study protocol must also be provided in the manuscript.

# Human research participants

Policy information about studies involving human research participants

| Population characteristics | Healthy male and female subjects between 21 and 70 years of age. |
|---|---|
| Recruitment | Participants were searched using bulletins in which healthy individuals interested in taking part in a study examining aging-related changes were asked to contact the Dept. 2 of Internal Medicine (UoC) by telephone. A trained employee ruled out relevant pre-existing diseases using a structured questionnaire. The test persons were then invited for an appointment at the University Hospital Cologne to obtain the blood samples used for sequencing in the study at hand. |
| Ethics oversight | Institutional Review Board, Medical Faculty, University of Cologne |

Note that full information on the approval of the study protocol must also be provided in the manuscript.

