## [Peer Review File · Nature]

Manuscript Title: Aging-associated changes in transcriptional elongation influence metazoan longevity

Reviewer Comments & Author Rebuttals

Reviewer Reports on the Initial Version:

Referees' comments:

Referee #1 (Remarks to the Author):

In this paper, Debès et al have shown across multiple model organisms that the rate of transcription elongation increases with age. They suggest that this increase in elongation speed is associated with a rise in splicing defects and production of circRNAs, and might contribute to the known loss of cellular RNA homeostasis seen with age. Further, they have also shown lifespan-extending interventions to have a positive effect on this phenomenon and that mutations that reduce Pol-II speed alone can promote lifespan and healthspan in worms and flies. The paper has merit as a resource and presents a novel finding shedding light on loss of transcriptional homeostasis with age. However, as is, it is over-reliant on descriptive RNA seq analysis alone, and would be greatly improved with more causal mechanistic insight. The strength of the work is the broad range of systems in which the central effect is seen. The main weakness is the overreliance on one methodology, and the very limited data assigning causal empirical data to support the hypothesis that RNA pol II speed drives aging via inducing defects in RNA processing.

Major comments:

- Perhaps the major drawback is that the authors have heavily relied on the use of descriptive RNA-Seq datasets and intron reads to visualize changes in Pol II speed. Since this is the single main finding in the paper, the conclusion needs substantiating with more direct measurements like CHIP-Seq, or more ideally Gro-Seq, NET-Seq or BruDRB-seq to look at Pol II speeds, at least in one/some model systems to validate their findings. (PMID: 25693130). The methods make a comparison between approaches using published data, but given the central premise of the paper is so reliant on one methodology it would greatly strengthen the conclusions if a similar results with age were seen using an independent approach.
- The causal data of directly reducing Pol II speed and increasing longevity are somewhat limited, and the paper would be greatly strengthened by deeper characterization of the long-lived mutant *C. elegans* and *Drosophila* lines. Critically as it stands there are no data that causally link the effect of these mutations on aging to the proposed hypothesis, namely slowed pol II and higher RNA processing fidelity. The authors attribute the longevity of genetic mutants of Pol II to reduced speed of elongation and efficient splicing (Figure 2) but alternative explanations are not sufficiently discussed/considered or explored. Is it possible that reduced Pol II speed reduces overall transcription and thereby inhibiting protein synthesis? There is evidence showing reducing protein translation increases lifespan (e.g. Hansen M, et al. Lifespan extension by conditions that inhibit translation in *Caenorhabditis elegans*. *Aging Cell*. 2007). The authors should measure overall RNA and protein synthesis in these mutants or discuss it if it has already been tested before.

- The short-lived genetic model organisms represent an opportunity to gain more insight into the mechanisms of longevity, that seems rather underutilized here given the depth of expertise and resources of the team. Longevity of the mutant *C. elegans* and *Drosophila* should be rescued to ensure specificity of the effects to the mutations. Is there a specific tissue in which RNA pol II speed has its longevity effects or is this a cell autonomous model? Does speeding up Pol II have inverse effects on aging? There are Pol II mutants that have increased speed in yeast which might be recapitulated in these systems. Malagon, F., Kireeva, M.L., Shafer, B.K., Lubkowska, L., Kashlev, M., and Strathern, J.N. (2006). Mutations in the *Saccharomyces cerevisiae* RPB1 gene conferring hypersensitivity to 6-azauracil. *Genetics* 172, 2201–2209.

- Given the wealth of known longevity modifiers in worm and fly, some attempt could be made to see if the longevity of particularly the *ama-1* mutant *C. elegans* acts dependently or independently on canonical longevity effectors well known to these groups.

- The authors observed more spliced exon junctions under conditions of accelerated Pol II and emergence of rare isoforms. Did they observe changes in the number of intron-exon junctions i.e. a readout of intron retention? The authors should present that data. Further, does changing Pol II speed have an effect on nonsense transcripts and NMD?

- The authors state “... faster Pol-II elongation resulted in an increase of circRNA formation”. This claim is based on correlation and is not directly tested- they show circRNA formation increases with age in some but not all of their data sets. Pol-II elongation mutants have reduced circRNA but they also retard the aging process and as such one might expect all aging related traits to be delayed. Therefore, if high elongation speed is causal to increased circRNA production or is a consequence of physiological age is not demonstrated. While this is not trivial to test, might it be possible to present a correlation of change in transcriptional speed on different genes with age and abundance of circRNA from back-splicing of those genes to draw a more direct comparison.

- The final data aimed toward mechanistically linking age induced changes to Pol II speed to chromatin architectural changes with age are interesting, but empirical evidence demonstrating causality between these observations is lacking. As stated: “...reduced precision in the assembly of the chromatin fiber may contribute to changes in Pol-II speed and splicing fidelity”. Transcription involves acetylation and transfer of histones behind RNA Pol II to suppress faulty transcription initiation within the gene body. Whether variability in nucleosome position is a cause or a consequence of increase in Pol II speed is not tested directly. Data toward this goal would enhance the mechanistic claim in figure 4.

Minor comments:

- The authors should elaborate more about the nature of the introns that passed the filtering steps and were used for analysis.

o Does increase of Pol II speed have anything to do with intron size? In other words, is this increase in speed reflected more on longer introns? The authors should address this since the introns

analyzed in worms (546) vs human samples (13,790) are vastly different with a lot more introns analyzed in higher organisms. Could this be attributed to abundance of long introns in humans?

o Are these introns located closer to promoters or not? (i.e. if this has anything to do with elongation speed vs leaky Pol II release from promoters?)

- In the Pol II elongation speed analysis of *ama-1* and *Rpl1215C4* mutants in worms and flies, was the analysis done with all reads or from reads from the same intronic regions that were initially identified in the young/old samples?

- Power analysis should be performed on the longevity experiments, particularly the *Drosophila* lifespans, to ensure that they have power to really detect such small differences.

- The authors rightly state the difficulty in assigning any RNA processing event as maladaptive or adaptive, and critique published methods. However, it remains only conjecture that “extremely rare isoforms are more likely erroneous than are frequent isoforms” and as such the claims around these data as being a more accurate read out of defective splicing should be toned down.

Referee #2 (Remarks to the Author):

Debes et al present a provocative study describing altered transcriptome characteristics of aged organisms relative to younger. These changes are described as consistent with altered Pol II elongation rates or other properties such as fidelity of transcription or RNA processing (increase in rate or other defects upon aging). Consistent with potential changes corresponding to determinants of lifespan, interventions that affect lifespan show altered transcriptome characteristics (mitigation of age-related effects for the most part) and genetic perturbation of Pol II corresponds to increased lifespan in two organisms, *C. elegans* and *D. melanogaster* as well as decrease in putative elongation rate and transcription/RNA processing defects in aggregate. The work is exciting and bolstered by the ability to connect putative measure of transcription with known lifespan interventions and connecting transcription intervention with lifespan. On the surface the work appears carefully done with aggregate analyses appearing to indicate what the authors suggest, while a number of distinct molecular phenotypes are examined. However, the analyses must be discussed and presented in a much more vigorous and meticulous way so it may be better understood how well the inferred elongation rates actually relate to potential elongation across genes or point to some other defects.

Issues of concern:

1. Elongation rate has been inferred indirectly (by necessity) through analysis of RNA-seq determined 5' to 3' gradient in intronic sequence levels. The reliability of this measurement on an intron and sample basis must be presented. Effect sizes are presented that appear to relate to the shift in aggregate distributions of determined elongation rates and therefore have appearance of

small confidence intervals. This may be reasonable, but given that there is no discussion on the noise of any of the individual measurements, I think this is a major shortcoming of the presentation.

2. Repeats are generally discussed as having been performed and statement is made that samples compared are handled together. It should be made clear that the sequencing strategy has not put in place any confounding variables such as comparisons sequenced in different lanes. Otherwise, it should be made clear how technical error has been estimated or dealt with.

3. Degradation of excised lariats has been stated as not contributing to potential slope for elongation rates. This analysis was based on a single sample in the SnapShot-Seq paper and potentially could be assessed directly for the libraries in the manuscript here. It seems important to rule out this potential confounding variable directly.

4. I have not directly compared the calculations used to determine elongation rate between this ms and either the cited Gray et al paper (PMID: 24586954) or this alternate approach (<https://doi.org/10.1093/bioinformatics/bty886>), but it would potentially important to determine how robust results are to method of calculation. Furthermore, in the latter work, elongation rate by intron is analyzed with respect to intron position in gene (see Figure 5 heat map). Two important things: one- the idea that premature termination should not be a issue in calculations depends on which introns are used for determination and where they are in a gene. Introns actually used should be analyzed to examine if they deviate in potentially meaningful ways from introns on average. Second, displaying rates determined by position in gene could potentially aid in interpretation. Furthermore, whether differences are localized to particular parts of genes will be apparent by generation of difference heat maps between samples being compared.

5. Throughout the manuscript, correlations are presented in language implying causation “leads to...” etc. Please adjust language to more appropriate causation-agnostic language.

Other issues

Abstract

1. “This increase in polymerase transcriptional speed was associated with extensive splicing defects..”

Associated is perhaps stronger than intended. “correlated” would be better.

Results

2. The changes in elongation rates are expressed as averages over the samples that appear very precise. However, the determination of rate by almost any reasonable measure would likely have standard deviation of 10-20% due to the nature of estimating elongation rates. Therefore effect sizes of a few dozens to ~100 nt/minute on a value that is around 3000-4000 nt/minute will maybe seem counter-intuitive. Along these lines, the reported in vitro defect for C4 fly allele would be much greater than the measured defect here. While possible that in vitro defect may be mitigated in vivo,

for other types of polymerase mutants, in vitro defects measured in vivo do somewhat correspond in magnitude. Therefore, what the aggregate effect size actual means is not quite clear.

3. p8 “for some exons, slow elongation favors weak splice sites, leading to exon inclusion, while these are skipped during fast elongation”

Because it has not been determined in the literature whether any of observed changes in processing in response to altered elongation rate are in fact direct, more conservative language on this is warranted, i.e. “leading to...during fast” are hypotheses not facts.

4. p9. Language here “Thus, our data suggests that faster Pol-II elongation resulted in an increase of circRNA formation, which lead to a global increase of exon skipping for genes hosting circular RNAs. Taken together, these findings suggest that an age-associated acceleration of transcription and splicing leads to increased splicing noise with impact on lifespan.”

“results in...lead to...leads to” all imply stronger causation than is warranted.

5. p11. “Thus, age-associated changes in chromatin structure might contribute to the changes in Pol-II elongation”

Contribute to, or alternatively reflect. Causation is not clear.

6. Figure 3. What are the multiple “Senescent vs. proliferating” data points for HUVECs?

7. For worm life span, was the censored data a significant fraction or distinct between worm genotypes/treatments?

8. Figure S7. State what the error bars are (median+IQR?).

9. Figure S13. Why do the number of genes analyzed for this analysis match the number of introns analyzed on Figure 1? Were the regions in fact the same or different?

10. Supplementary Figure 14. What is the y axis? The correlation with GRO-seq data seems to be strongly driven by a few points- how robust is this correlation in the absence of the slope outliers?

11. For the MNase analyses, were there repeats, what was the digestion level for these samples, were they digestion matched, how did the repeats correlate with each other? It is difficult to interpret these data without first understanding what is reproducible about them and whether the sample preparation is such that samples might reasonably be compared.

Referee #3 (Remarks to the Author):

Based on RNA-seq analyses in a number of model systems, the authors report a correlation between average RNAPII elongation rates and aging. This is potentially a significant result that, if published in Nature, will undoubtedly attract very wide interest. There are, however, two major issues. The first is the reliability of the assay for elongation rate that underpins the entire MS. The second is the limited evidence for causality in the observations; the concern being that these might represent multiple, but largely unconnected, effects of changes in aging cells. The magnitudes of the effects reported are also generally very small. The authors propose that this is inevitable, given that the biological impacts of the alterations are modest. This is presumably true, but it makes the functional significance of the specific observations hard to assess. The clearest data comes from Fig. 2 that convincingly shows enhanced longevity in animals with mutant RNAPII.

Overall, this is an interesting report on an important topic. However, it appears to be too preliminary for publication in Nature.

Specific points:

1) Fig. 1; The approach of “SnapShot-Seq” appears reasonable. In principal, it should provide genome-wide transcription rate data from simple RNA-seq analyses (Gray et al. PLoS One, 2014). The description of the technique in the text seems to be slightly misleading. The approach relies on very rapid, cotranscriptional splicing and degradation of intronic sequences, such that the nascent transcripts represent the predominant fraction of the total RNA population. The 5’-3’ slope of the line is then interpreted as largely reflecting the relative abundance of the nascent transcripts, and therefore the elongation rate. However, the approach crucially depend on rapid degradation of the excised intron before and after debranching, since this will also be represented in the RNAseq data. If degradation is slower than, or comparable to the RNAPII elongation rate, the intron will make a significant contribution to the RNAseq results. As one of many conceivable possibilities; if aging cells have reduced 3’ degradation of the intron (perhaps due to reduced RNA exosome activity), this will generate an increased 5’-3’ slope in the sequence data, which would be interpreted as enhanced elongation. The authors quote Gray et al (2014) for the statement that “intron gradient is not influenced by exonucleolytic degradation of excised intron lariats”. However, this is not clearly demonstrated in that paper, which reports indirect data “suggesting” this to be the case in HeLa cells. I am not proposing this as a specific explanation, rather as an indication that alternative explanations may be possible.

Given the potential importance and expected visibility of the finding on enhanced elongation, it needs some backup from an independent method to verify the changes. For example, single molecule analyses.

2) Related this, there are numerous reports of systematic differences in base composition and histone modifications between intronic and exonic sequences in systems from yeast to humans. It would therefore be important to demonstrate that the observed changes are not intron-specific.

3) Fig. 4: This seems to be something of an after-thought, presumably added to provide some

mechanistic basis for the RNA seq data. Given that HDAC inhibitors have been linked to aging, alterations in chromatin structure appear very plausible. It was, however, unclear to this referee how the changes in elongation rate can be functionally related to the “fuzzy” nucleosome positioning? Is there reason to think that alteration in nucleosomes are a cause rather than a consequence of altered transcription?

Minor points:

4) In Figs. S10 and S13: The frequency of rare splice events and errors show a tendency for increases in the aging models. However, this is far from clear cut. Is there any way show that this tendency differs statistically from random?

5) What does Fig. S14 demonstrate?

6) P8: “(S8 “ - Missing bracket

7) Fig. S10: Progeria appears to be in blue, in contrast to legend.

8) Fig. S12: A different color combination would make the data points clearer.

Author Rebuttals to Initial Comments:

Debès et al.: Aging-associated changes in transcriptional elongation influence metazoan longevity

Response to Reviewers

We are very grateful to all three reviewers for their insightful and fair comments! They have helped us a lot to significantly improve the quality of our submission.

In particular, we have now supported our claims with an independent assay to measure RNA Pol-II elongation speed, based on RNA labelling. That new data now independently supports our major finding: Pol-II speed increases with age. Further, we have significantly expanded our experiments adding evidence for a potential causal effect of chromatin structure on age-associated Pol-II speed changes. In our initial submission we already provided data showing age-associated changes in nucleosome positioning and density. We have improved the analysis of this MNase data using more advanced bioinformatic tools. Expanding on that, we have created new data using histone overexpression mutants, which show that histone overexpression reduces Pol-II speed in mammalian cells (which confirms earlier findings from yeast) and – importantly – that this overexpression delays entry into senescence. Triggered by those findings, we have created histone-overexpressing fly lines, in order to test also organismal effects of nucleosome density. Reassuringly, that data showed a lifespan-extending effect of histone 3 overexpression. To the best of our knowledge, this is the first time that such lifespan effects of nucleosome density were shown in an animal model.

Taken together this new data (1) adds much more confidence to the age-associated Pol-II speed increase and (2) provides significantly more information for a potentially causal role of chromatin organization.

Of course, we have addressed all other concerns of the reviewers (see below) and we have completely revised the text in order to structure it better, to remove possible misunderstandings and to tone down potential overstatements. In particular, we have worked on the Discussion section in that respect.

We are very excited by our new results and hope that the reviewers will agree that we have addressed all concerns. We are very much looking forward to the reviewers' feedback!

The **manuscript version with change tracking** that we are submitting along highlights changes of the text message. In the interest of readability we have refrained from highlighting grammatical changes or sentence re-arrangements that do not change the message. In addition, all figures and figure captions were changed; those changes were also not highlighted in the interest of readability.

In the following, reviewer comments are shown in **blue**, while our responses are shown in black.

Referees' comments

Referee #1 (Remarks to the Author):

In this paper, Debès et al have shown across multiple model organisms that the rate of transcription elongation increases with age. They suggest that this increase in elongation speed is associated with a rise in splicing defects and production of circRNAs, and might contribute to the known loss of cellular RNA homeostasis seen with age. Further, they have also shown lifespan-extending interventions to have a positive effect on this phenomenon and that mutations that reduce Pol-II speed alone can promote lifespan and healthspan in worms and flies. The paper has merit as a resource and presents a novel finding shedding light on loss of transcriptional homeostasis with age. However, as is, it is over-reliant on descriptive RNA seq analysis alone, and would be greatly improved with more causal mechanistic insight. The strength of the work is the broad range of systems in which the central effect is seen. The main weakness is the overreliance on one methodology, and the very limited data assigning causal empirical data to support the hypothesis that RNA pol II speed drives aging via inducing defects in RNA processing.

Major comments:

1) Perhaps the major drawback is that the authors have heavily relied on the use of descriptive RNA-Seq datasets and intron reads to visualize changes in Pol II speed. Since this is the single main finding in the paper, the conclusion needs substantiating with more direct measurements like ChIP-Seq, or more ideally Gro-Seq, NET-Seq or BruDRB-seq to look at Pol II speeds, at least in one/some model systems to validate their findings. (PMID: 25693130). The methods make a comparison between approaches using published data, but given the central premise of the paper is so reliant on one methodology it would greatly strengthen the conclusions if a similar results with age were seen using an independent approach.

We acknowledge the need for additional experimental measurements in support of the central message of our work, i.e. of the change in RNAPII translocation speed. To this end, and following the reviewer's suggestion, we opted for using 4sU-DRB-seq (Fuchs et al, *Nat Protoc*, 2015). This methodology exploits reversible inhibition of transcriptional elongation via DRB treatment (to synchronize transcription genome-wide) with the metabolic labeling of newly-synthesized transcripts following washout of the inhibitor (to discriminate between new and residual longer-lived transcripts). In addition, we combined 4sU-DRB-seq with a chemical conversion of incorporated 4sUTP into cytidines on the basis of the TUC-seq approach (Lusser et al, *Methods Mol Biol*, 2020). This allowed us to circumvent biases/variability potentially arising from 4sUTP biotinylation of its subsequent pulldown by directly measuring the number of U-to-C conversions in our sequencing data. We generated data from consecutive time points after DRB release (i.e. at 0, 5, 15, 30 and 45 min) and monitored the progression of elongation to calculate RNAPII translocation speeds. These new data are now presented in **Extended Data Figure 4** and are in full agreement with measurements inferred from nascent or bulk (ribodepleted) RNA-seq used previously and, thus, in full support of our findings that RNAPII speed increases with age.

Please note that we did not opt for using GRO-seq due to the somewhat artificial manner by which transcriptional "run-on" is induced (i.e. via the addition of sarcosyl – see comparison to nascent RNA-seq in Caudron-Herger et al, *Nucleic Acids Res*, 2015), while NET-seq was not preferred

due to the complicated and rather costly way by which libraries are constructed (Mayer et al, *Cell*, 2015). Also, both these methodologies require significantly more primary cells than the $\sim 2 \times 10^6$ that we used per condition and time point here.

2) The causal data of directly reducing Pol II speed and increasing longevity are somewhat limited, and the paper would be greatly strengthened by deeper characterization of the long-lived mutant *C. elegans* and *Drosophila* lines. Critically as it stands there are no data that causally link the effect of these mutations on aging to the proposed hypothesis, namely slowed pol II and higher RNA processing fidelity. The authors attribute the longevity of genetic mutants of Pol II to reduced speed of elongation and efficient splicing (Figure 2) but alternative explanations are not sufficiently discussed/considered or explored. Is it possible that reduced Pol II speed reduces overall transcription and thereby inhibiting protein synthesis? There is evidence showing reducing protein translation increases lifespan (e.g. Hansen M, et al. Lifespan extension by conditions that inhibit translation in *Caenorhabditis elegans*. *Aging Cell*. 2007). The authors should measure overall RNA and protein synthesis in these mutants or discuss it if it has already been tested before.

We thank the reviewer for pointing out this potential issue. Indeed, expression of components involved in transcription and translation was downregulated in the *C. elegans* slow Pol-II mutant. As pointed out, many lifespan-extending mutations in *C. elegans* are indeed associated with reduced RNA and protein biosynthesis. Hence, this observation neither confirms nor precludes the possibility that changes in protein biosynthesis also contribute to the phenotype of the pol-II speed mutation. However, we did not observe such an effect in the *Drosophila* slow Pol-II mutants. In fact, we directly measured protein synthesis in these mutants and saw that the mutation did not lead to any significant changes (see also **Extended Data Figure 17**). We therefore conclude that, at least in flies, lifespan extension cannot simply be explained by a global reduction in protein biosynthesis.

3) The short-lived genetic model organisms represent an opportunity to gain more insight into the mechanisms of longevity, that seems rather underutilized here given the depth of expertise and resources of the team. Longevity of the mutant *C. elegans* and *Drosophila* should be rescued to ensure specificity of the effects to the mutations. Is there a specific tissue in which RNA pol II speed has its longevity effects or is this a cell autonomous model? Does speeding up Pol II have inverse effects on aging? There are Pol II mutants that have increased speed in yeast which might be recapitulated in these systems. Malagon, F., Kireeva, M.L., Shafer, B.K., Lubkowska, L., Kashlev, M., and Strathern, J.N. (2006). Mutations in the *Saccharomyces cerevisiae* RPB1 gene conferring hypersensitivity to 6-azauracil. *Genetics* 172, 2201-2209.

In order to address this point, we performed additional experiments in *C. elegans*. We introduced a mutation in *C. elegans* RNAPII (E1120G) in which the corresponding residue (E1103G) in yeast accelerates polII activity (Malagon, et al.(2006).*Genetics* 172, 2201–2209). Unfortunately, these mutant worms were sterile, possibly indicating the adverse effects following excessive speed-up of transcription. Due to the sterility, we could not further profile these mutants.

However, we were able to demonstrate that Crispr engineered reversion of the slow mutation *ama-1(g/a; R739H; m322)* back to the wild type allele *ama-1(a/g; H739R; syb2315)* restored lifespan essentially back to wild-type levels, as shown in **Extended Data Figure 8b**. Crispr reversion is a more rigorous way to be certain that the phenotype arises strictly from this mutation and not linked mutations. Transgenic rescue is a less desirable approach since dose, mosaicism, and position effects can come into play.

Regarding the question on tissue specificity, we observed an age-associated increase of RNAPII speed in virtually all mammalian tissues that we tested. Thus, as far as we can tell, the speed increase is a global phenomenon. Future work would be needed to reveal which tissue(s) or cell type(s) are mostly responsible for the lifespan extension in slow RNAPII mutants, which would require generation of tissue/cell type-specific RNAPII mutants, which is beyond the scope of the current study. (Note that simply overexpressing a 'slow' allele in a cell-type specific way would not suffice. One would have to repress/replace the wild-type allele at the same time, while maintaining the stoichiometry of all Pol-II components.)

4) Given the wealth of known longevity modifiers in worm and fly, some attempt could be made to see if the longevity of particularly the *ama-1* mutant *C. elegans* acts dependently or independently on canonical longevity effectors well known to these groups.

As mentioned above, we have analyzed the slow Pol-II mutants in greater detail and indeed found downregulation of components involved in mRNA and protein productions in the case of *C. elegans* (see heatmaps below). These changes overlap with the effects of many of the known lifespan-modulating mutations in the worm. We have now systematically identified genes that were differentially expressed between speed mutants and wild-type strains. Yet, determining which of those are causally involved in the lifespan phenotype would require additional generation and characterization of strains across the different models we use, which would be undoubtedly laborious and out of scope of the current manuscript. The same holds true as regards epistasis testing of the speed mutants.

Rebuttal Letter Figure 1: Heatmap showing gene expression differences of components of RNA polymerase II, components of the NMD machinery, and ribosomal protein-coding genes between RNA-Pol-II speed mutants in *C. elegans* with *ama-1* mutation(ama1r1-r3) versus wild type(wtr1-wtr3).

5) The authors observed more spliced exon junctions under conditions of accelerated Pol II and emergence of rare isoforms. Did they observe changes in the number of intron-exon junctions i.e. a readout of intron retention? The authors should present that data. Further, does changing Pol II speed have an effect on nonsense transcripts and NMD?

Indeed, computing intron retention in the way as suggested by the reviewer leads to similar results. However, computing splicing efficiency in the way we do in the manuscript reduces possible confounding with speed change--unlike exon/intron ratio, the score we compute is not affected by alternative splicing.

Regarding a potential role of NMD, we have taken efforts to identify NMD substrates *de novo* by assuming that splicing mistakes would give rise to isoforms not annotated in any database. Despite those efforts, we did not find consistent global changes in the fraction of NMD substrates across our datasets (see figure below, where many error bars cross the zero-line). Thus, we cannot validate any direct, causal impact of RNAPII speed on the production of NMD substrates. Future work will have to analyze nuclear and cytoplasmic RNA fractions separately in order to separate changes in the production of NMD substrates from changes in (NMD-related) processing of transcripts. This way it will be possible to disentangle direct Pol-II speed effects from changes in NMD efficiency. However, such experiments would be beyond the scope of this manuscript.

Rebuttal Letter Figure 2: Expression changes of NMD substrates (i.e. transcripts with early stop codons) in the various ageing and lifespan models we tested. Note the large error bars and the complication that our data cannot distinguish the nuclear fraction from the cytoplasmic fraction.

Rebuttal Letter Figure 3: Heatmap of expression levels of NMD components in RNA-Pol-II speed mutants versus wild type in *C.elegans* and *Drosophila*.

6) The authors state "faster Pol-II elongation resulted in an increase of circRNA formation". This claim is based on correlation and is not directly tested- they show circRNA formation increases with age in some but not all of their data sets. Pol-II elongation mutants have reduced circRNA but they also retard the aging process and as such one might expect all aging related traits to be delayed. Therefore, if high elongation speed is causal to increased circRNA production or is a consequence of physiological age is not demonstrated. While this is not trivial to test, might it be possible to present a correlation of change in transcriptional speed on different genes with age and abundance of circRNA from back-splicing of those genes to draw a more direct comparison.

We thank the reviewer for pointing out this issue. We have tested such correlation and indeed did not find a correlation between speed changes (on a per gene basis) and increased formation of circRNAs (from that given gene). Thus, in full agreement with the reviewer's comment, we conclude that the reduced number of circular RNAs rather reflects an overall better homeostatic status of the cells. We have accordingly changed our conclusions in the manuscript to reflect this:

"During aging (old *versus* young) we observed either increased or unchanged average circRNA fractions (**Extended Data Fig. 14**). In contrast, reducing Pol-II speed reduced circRNA formation on average. Thus, our data suggests that faster Pol-II elongation correlates with a general increase of circRNA formation."

7) The final data aimed toward mechanistically linking age induced changes to Pol II speed to chromatin architectural changes with age are interesting, but empirical evidence demonstrating causality between these observations is lacking. As stated: ".reduced precision in the assembly of the chromatin fiber may contribute to changes in Pol-II speed and splicing fidelity". Transcription involves acetylation and transfer of histones behind RNA Pol II to suppress faulty transcription initiation within the gene body. Whether variability in nucleosome position is a cause or a consequence of increase in Pol II speed is not tested directly. Data toward this goal would enhance the mechanistic claim in figure 4.

We have now addressed this point in multiple ways. First, we generated data showing that overexpression of histone H3 in the glia of fly brains, increases chromatin compaction as would be expected, but also increases fly lifespan (Figure 5g,h). Second, we generated histone H3- and H4-overexpressing human IMR90 lines and measured elongation rates and senescence markers in late passage cells. We observed that entry into senescence was delayed in both overexpressing cell lines and that RNAPII transcriptional speed was accordingly slowed down (Figure 5a-f). This consolidates the link between the changing chromatin landscape in cellular ageing and the speed of RNAPII translocation.

Minor comments:

1) The authors should elaborate more about the nature of the introns that passed the filtering steps and were used for analysis.

a) Does increase of Pol II speed have anything to do with intron size? In other words, is this increase in speed reflected more on longer introns? The authors should address this since the introns analyzed in worms (546) vs human samples (13,790) are vastly different with a lot more introns analyzed in higher organisms. Could this be attributed to abundance of long introns in humans?

We thank the reviewer for this comment and have tested now the correlation between intron size and Pol-II speed (**Extended Data Table 1**). The change in Pol II speed is not correlated with intron size in any of the tested organisms. We agree with the reviewer that this is an important quality test, and therefore we included this table as **Extended Data Table 1** in the manuscript. Further, the reviewer is completely right: the larger number of usable introns in mammals compared to worms results from the larger total number of introns in mammals compared to worms and from the (on average) longer introns. Since we require a certain minimal length to determine Pol-II speed, we do have a bias against short introns in our analysis. Short introns simply do not provide enough data. However, since we could not detect a correlation between intron length and Pol-II speed changes we do not assume that the global trend would be very different for shorter introns.

We have made this explicit in the Results section:

“After filtering, we obtained between 546 and 14,593 introns that passed the quality criteria for reliable Pol-II speed quantification (see Methods). These different numbers of usable introns mostly result from inter-species variation in intron sizes and intron numbers and to some extent from variation in sequencing depth.”

b) Are these introns located closer to promoters or not? (i.e. if this has anything to do with elongation speed vs leaky Pol II release from promoters?)

As suggested we also tested for a possible correlation between Pol II elongation rate and the distance from the promoter. The change in elongation rate is uncorrelated with the distance of the intron from the promoter. The corresponding table has been included in the manuscript (**Extended Data Table 1**).

2) In the Pol II elongation speed analysis of *ama-1* and *RplI215C4* mutants in worms and flies, was the analysis done with all reads or from reads from the same intronic regions that were initially identified in the young/old samples?

The analysis of the mutants could not be done with the exact same set of introns as in the ageing experiment. We require high expression for every intron in every sample used in the analysis. The *ama-1* mutation and other treatments that extend lifespan induce differential expression and therefore some genes that are highly expressed in one experiment are lowly

expressed in another experiment and *vice versa*. Constraining our analysis to introns that pass our stringent filtering in every single experiment would mean working with a very limited pool of introns.

Therefore, we always use experiment-specific controls, which also minimizes batch effects. Within experiment (old versus young, mutant versus wild type) we always compare the same sets of introns. Thus, the pool of usable introns is always specifically determined for each experiment, while requiring that the respective intron was sufficiently highly expressed in all samples of the particular comparison (i.e. contrast).

3) Power analysis should be performed on the longevity experiments, particularly the *Drosophila* lifespans, to ensure that they have power to really detect such small differences.

The population sizes for the lifespan experiments are well within the range used in multiple lifespan studies by us and others (see the references below as examples). Further, the fact that we obtained statistically significant effects underlines that the population size was sufficient for the purpose of this study.

- Tain, Luke S., et al. "A proteomic atlas of insulin signalling reveals tissue-specific mechanisms of longevity assurance." *Molecular systems biology* 13.9 (2017): 939.
- Weigelt, Carina Marianne, et al. "An Insulin-Sensitive Circular RNA that Regulates Lifespan in *Drosophila*." *Molecular cell* 79.2 (2020): 268-279.

4) The authors rightly state the difficulty in assigning any RNA processing event as maladaptive or adaptive, and critique published methods. However, it remains only conjecture that "extremely rare isoforms are more likely erroneous than are frequent isoforms" and as such the claims around these data as being a more accurate read out of defective splicing should be toned down.

We intentionally wrote 'more likely', implying that neither every rare splice event is necessarily a splicing mistake, nor every frequent splicing event 'correct'. Hence, we do not make any statement about individual splicing events. In other words, we assume that rare splicing events are *enriched* for mistakes compared to frequent events. Pickrell et al. have shown more than 10 years ago that rare splice junctions are less conserved and likely represent splicing mistakes (Ref. 35). This notion has later been confirmed by Stepankiw et al. (Ref. 36). Thus, we are by far not the first ones using this concept to identify potentially erroneous splicing events. Finally, we provide additional evidence that these events are more frequently erroneous:

"Indeed, we observed that these rare exon-exon junctions often resulted from exon skipping or from the usage of cryptic splice sites (Extended Data Fig. 11a)."

Thus, we believe that in the context of existing literature and in view of our own data using the fraction of rare isoforms as a measure for global splicing fidelity is justified.

Referee #2 (Remarks to the Author):

Debes et al present a provocative study describing altered transcriptome characteristics of aged organisms relative to younger. These changes are described as consistent with altered Pol II elongation rates or other properties such as fidelity of transcription or RNA processing (increase in rate or other defects upon aging). Consistent with potential changes corresponding to determinants of lifespan, interventions that affect lifespan show altered transcriptome characteristics (mitigation of age-related effects for the most part) and genetic perturbation of Pol II corresponds to increased lifespan in two organisms, *C. elegans* and *D. melanogaster* as well as decrease in putative elongation rate and transcription/RNA processing defects in aggregate. The work is exciting and bolstered by the ability to connect putative measure of transcription with known lifespan interventions and connecting transcription intervention with lifespan. On the surface the work appears carefully done with aggregate analyses appearing to indicate what the authors suggest, while a number of distinct molecular phenotypes are examined. However, the analyses must be discussed and presented in a much more vigorous and meticulous way so it may be better understood how well the inferred elongation rates actually relate to potential elongation across genes or point to some other defects.

Issues of concern:

1. Elongation rate has been inferred indirectly (by necessity) through analysis of RNA-seq determined 5' to 3' gradient in intronic sequence levels. The reliability of this measurement on an intron and sample basis must be presented. Effect sizes are presented that appear to relate to the shift in aggregate distributions of determined elongation rates and therefore have appearance of small confidence intervals. This may be reasonable, but given that there is no discussion on the noise of any of the individual measurements, I think this is a major shortcoming of the presentation.

We thank the reviewer for pointing this out. The additional analyses we have now performed on the basis of these comments have enhanced the robustness of our findings and have strengthened our conclusions.

- 1) We have created PCA plots of the samples on the basis of the estimated Pol-II elongation rates (**Extended Data Figure 1**). Those plots show that speed changes between replicates are consistent and that the differences in elongation rate between conditions are significantly greater than the differences between replicates. When comparing samples as regards intron-specific elongation rates, they mostly cluster by treatment (senescent versus proliferating, etc.), underlining that there is a consistent trend of speed changes with age or treatment.

- 2) The average increase in Pol-II elongation speed was even more pronounced after selecting introns with consistent speed changes across all replicates (here we defined 'consistent' as always up or down with age across all replicates). Note that this result is non-trivial, because our analysis also included introns with consistent *reduction* in Pol-II speed. **Extended Data Figure 3a,b** show speed changes estimated either for all introns that passed our filters ('All') or for the subset of introns that showed consistent speed changes across all replicates ('Consistent only').
- 3) We compared the speed changes in two different human cell lines (HUVEC and IMR90). While the speed changes themselves have low correlation, introns tend to change in speed in the same direction, as shown in **Extended Data Figure 3c**.
- 4) We have confirmed the average increase in Pol-II elongation speed using an alternative RNA labelling assay (see above, comment to Reviewer 1).

Thus, even though average effect sizes (i.e. absolute changes in speed) are low, we can show that hundreds of genes are consistently affected by changes in Pol-II speed. Nevertheless, speed changes in introns of particular genes may actually be quite dramatic. For instance, in the case of the senescent IMR90 cells, we observed a great increase in *SEPTIN7* (log₂FC of slope= 3.17) or *PHACTR2* (log₂FC of slope=2.64). Note that we rarely observed such extreme cases for *reduced* speed.

Finally, we performed a number of additional tests and alternative ways of estimating Pol-II elongation speed (see below & responses to the other reviewers).

2. Repeats are generally discussed as having been performed and statement is made that samples compared are handled together. It should be made clear that the sequencing strategy has not put in place any confounding variables such as comparisons sequenced in different lanes. Otherwise, it should be made clear how technical error has been estimated or dealt with.

All comparisons of Pol-II speeds were done using samples of the same batches. As a consequence, some of the comparisons shown in Figure 1 actually use different control samples, to make sure controls are always from the same batch as the treatment.

Regarding the sequencing lanes: samples of the same batch were always sequenced on the same flow-cell and mostly by mixing across lanes. However, we are partially using published data, where this information is not available to us. The great consistency of our findings regarding Pol-II speed changes across species implies that this observation is not simply an artifact of sequencing reads on different lanes. (Plus, it is difficult to imagine how technical sequencing artifacts would consistently change the read *distribution* in introns.)

3. Degradation of excised lariats has been stated as not contributing to potential slope for elongation rates. This analysis was based on a single sample in the SnapShot-Seq paper and potentially could be assessed directly for the libraries in the manuscript here. It seems important to rule out this potential confounding variable directly.

First of all, lariat removal is known to be a very fast process acting on time scales that are not relevant for the Pol-II speed estimation (Ooi SL, Dann C 3rd, Nam K, Leahy DJ, Damha MJ, Boeke JD. RNA lariat debranching enzyme. *Methods Enzymol.* 2001;342:233-48). Furthermore, nascent RNA/ pre-mRNA degradation factors in the nucleus are Rat1(Nrd1) and Xrn1, which are 5'->3' exonucleases. Thus, even if their activity would change with age, they could at most have a global effect on the number of intronic reads, but not on the slopes that we and others observe, which results from declining read density in 5' -> 3' direction. Finally, we observe a speed increase with age also with our new DRB-4SU assay, which measures the progression of the 'transcription front'. Hence, that assay is independent of lariat removal.

4. I have not directly compared the calculations used to determine elongation rate between this ms and either the cited Gray et al paper (PMID: 24586954) or this alternate approach (<https://doi.org/10.1093/bioinformatics/bty886>), but it would potentially important to determine how robust results are to method of calculation. Furthermore, in the latter work, elongation rate by intron is analyzed with respect to intron position in gene (see Figure 5 heat map). Two important things: one- the idea that premature termination should not be a issue in calculations depends on which introns are used for determination and where they are in a gene. Introns actually used should be analyzed to examine if they deviate in potentially meaningful ways from introns on average. Second, displaying rates determined by position in gene could potentially aid in interpretation. Furthermore, whether differences are localized to particular parts of genes will be apparent by generation of difference heatmaps between samples being compared.

We do realize that RNAPII speed may change as a function of intron length and/or position. However, our conclusions consistently concern changes in speed between two conditions always comparing the same intron (e.g. the same intron in old versus young) – this way, any length or position bias would affect both conditions in the same way. In more detail, we indeed confirmed that the speed changes we observed are independent of intron length (details are provided to minor comment 1a of Reviewer 1). We also confirmed that RNAPII speed increases independently of the position of the given intron in the gene (see also above: minor comment 1b of Reviewer 1). Finally, our new DRB-4SU data confirms speed increase using a fully independent way of method for assessing translocation speeds (refer to major comment 1 of Reviewer 1 and major comment 1 of Reviewer 3).

5. Throughout the manuscript, correlations are presented in language implying causation "leads to." etc. Please adjust language to more appropriate causation-agnostic language.

This point is well taken; text has now been adjusted accordingly.

Other issues

Abstract

1. "This increase in polymerase transcriptional speed was associated with extensive splicing defects.."

Associated is perhaps stronger than intended. "correlated" would be better.

The manuscript text has been changed accordingly.

Results

2. The changes in elongation rates are expressed as averages over the samples that appear very precise. However, the determination of rate by almost any reasonable measure would likely have standard deviation of 10-20% due to the nature of estimating elongation rates. Therefore effect sizes of a few dozens to ~100 nt/minute on a value that is around 3000-4000 nt/minute will maybe seem counter-intuitive. Along these lines, the reported in vitro defect for C4 fly allele would be much greater than the measured defect here. While possible that in vitro defect may be mitigated in vivo, for other types of polymerase mutants, in vitro defects measured in vivo do somewhat correspond in magnitude. Therefore, what the aggregate effect size actual means is not quite clear.

Indeed, effect sizes are on average very small. Although some genes showed larger speed changes, most genes had very small relative changes. However, that is expected as large global changes of elongation speeds would most likely have devastating consequences for the cell (and possibly the whole organism) and are thus not observable. Regarding the in vitro data: those data were obtained from 'naked' DNA without nucleosomes and let alone any realistic chromatin structure (Coulter, Douglas E., and Arno L. Greenleaf. "A mutation in the largest subunit of RNA polymerase II alters RNA chain elongation in vitro." *Journal of Biological Chemistry* 260.24 (1985): 13190-13198.). It is therefore difficult to compare those rates with in vivo rates.

3. p8 "for some exons, slow elongation favors weak splice sites, leading to exon inclusion, while these are skipped during fast elongation"

Because it has not been determined in the literature whether any of observed changes in processing in response to altered elongation rate are in fact direct, more conservative language on this is warranted, i.e. "leading to.during fast" are hypotheses not facts.

The manuscript text has been changed accordingly.

4. p9. Language here "Thus, our data suggests that faster Pol-II elongation resulted in an increase of circRNA formation, which lead to a global increase of exon skipping for genes hosting circular RNAs. Taken together, these findings suggest that an age-associated acceleration of transcription and splicing leads to increased splicing noise with impact on lifespan."

"results in.lead to.leads to" all imply stronger causation than is warranted.

Text has been adjusted accordingly:

"Thus, our data suggests that faster Pol-II elongation correlates with a general increase of circRNA formation."

We have deleted the last sentence of that paragraph ("Taken together, these ...") and instead left it to the Discussion section, where we wrote:

"We observed consistent changes in splicing and transcript quality that correlated with Pol-II elongation speed changes ..."

5. p11. "Thus, age-associated changes in chromatin structure might contribute to the changes in Pol-II elongation"

Contribute to, or alternatively reflect. Causation is not clear.

Text has been adjusted accordingly.

6. Figure 3. What are the multiple "Senescent vs. proliferating" data points for HUVECs?

Sorry, the extra data points have been removed.

7. For worm life span, was the censored data a significant fraction or distinct between worm genotypes/treatments?

Actually, no worms were censored, but only objects falsely identified as worms. We have changed the wording in the Methods accordingly:

"Objects falsely identified as worms were censored."

8. Figure S7. State what the error bars are (median+IQR?).

Yes, error bars are median \pm 95% confidence interval. (Has been added to the figure caption.)

9. Figure S13. Why do the number of genes analyzed for this analysis match the number of introns analyzed on Figure 1? Were the regions in fact the same or different?

Yes, the regions are the same. Analysis was focused on the same introns analyzed in Figure 1.

10. Supplementary Figure 14. What is the y axis? The correlation with GRO-seq data seems to be strongly driven by a few points- how robust is this correlation in the absence of the slope outliers?

We have changed the way we calculate elongation rate changes. Instead of converting to base pairs per minute and then calculating the difference between the samples, we directly measure the fold change of the slopes. Thus, we have removed Supplementary Figure 14.

11. For the MNase analyses, were there repeats, what was the digestion level for these samples, were they digestion matched, how did the repeats correlate with each other? It is difficult to interpret these data without first understanding what is reproducible about them and whether the sample preparation is such that samples might reasonably be compared.

Two independent replicates were performed for each condition according to Diermeier et al, Genome Biol. (2014). Electrophoresis profiles for three of them are shown below (the 4th was run on a different gel on the day). As is evidence by the equiloading profiles, the 3-min MNase treatment resulted in comparable “ladder-like” profiles in both “young” and senescent IMR90, with the latter also being indicative of the overall reduced histone production in senescent cells. Mononucleosomal bands were cut out, DNA was isolated, and sequenced. Please note that each replicate comes from an independent donor/isolate, i.e. “repl1” comes from IMR90 I-10 and “repl2” from IMR90 I-79 (both obtained from the Coriell repository), which were digested side-by-side using the same amount of MNase from the exact same enzyme batch (see Methods). Finally, please also see PCA plots for the nucleosomal distances and sharpness scores derived from the NGS replicates.

Rebuttal Letter Figure 4: Western blot of three out of four replicates, showing comparable mononucleosomal quantities.

Rebuttal Letter Figure 5: PCA plots of (a) the nucleosomal distances in the introns and (b) the peak sharpness score in the introns, from the proliferating and senescent replicate and pooled samples.

Referee #3 (Remarks to the Author):

Based on RNA-seq analyses in a number of model systems, the authors report a correlation between average RNAPII elongation rates and aging. This is potentially a significant result that, if published in Nature, will undoubtedly attract very wide interest. There are, however, two major issues. The first is the reliability of the assay for elongation rate that underpins the entire MS. The second is the limited evidence for causality in the observations; the concern being that these might represent multiple, but largely unconnected, effects of changes in aging cells. The magnitudes of the effects reported are also generally very small. The authors propose that this is inevitable, given that the biological impacts of the alterations are modest. This is presumably true, but it makes the functional significance of the specific observations hard to assess. The clearest data comes from Fig. 2 that convincingly shows enhanced longevity in animals with mutant RNAPII.

Overall, this is an interesting report on an important topic. However, it appears to be too preliminary for publication in Nature.

Specific points:

1) Fig. 1; The approach of "SnapShot-Seq" appears reasonable. In principal, it should provide genome-wide transcription rate data from simple RNA-seq analyses (Gray et al. PLoS One, 2014). The description of the technique in the text seems to be slightly misleading. The approach relies on very rapid, cotranscriptional splicing and degradation of intronic sequences, such that the nascent transcripts represent the predominant fraction of the total RNA population. The 5'-3' slope of the line is then interpreted as largely reflecting the relative abundance of the nascent transcripts, and therefore the elongation rate. However, the approach crucially depend on rapid degradation of the excised intron before and after debranching, since this will also be represented in the RNAseq data. If degradation is slower than, or comparable to the RNAPII elongation rate, the intron will make a significant contribution to the RNAseq results. As one of many conceivable possibilities; if aging cells have reduced 3' degradation of the intron (perhaps due to reduced RNA exosome activity), this will generate an increased 5'-3' slope in the sequence data, which would be interpreted as enhanced elongation. The authors quote Gray et al (2014) for the statement that "intron gradient is not influenced by exonucleolytic degradation of excised intron lariats". However, this is not clearly demonstrated in that paper, which reports indirect data "suggesting" this to be the case in HeLa cells. I am not proposing this as a specific explanation, rather as an indication that alternative explanations may be possible.

Given the potential importance and expected visibility of the finding on enhanced elongation, it needs some backup from an independent method to verify the changes. For example, single molecule analyses.

We appreciate the need for orthogonal validation of our data on RNAPII translocation speed (as also pointed out by Reviewer 1). To this end, we opted for using 4sU-DRB-seq (Fuchs et al, *Nat Protoc*, 2015). This methodology exploits reversible inhibition of RNAPII elongation via DRB

treatment (to synchronize transcription genome-wide) with the metabolic labeling of newly-synthesized transcripts upon washout of the inhibitor (to discriminate between new and residual transcripts). In addition, we combined 4sU-DRB-seq with a chemical conversion of incorporated 4sUTP into cytidines on the basis of the TUC-seq approach (Lusser et al, *Methods Mol Biol*, 2020). This allowed us to circumvent biases/variability potentially arising from 4sUTP biotinylation of its subsequent pulldown by directly measuring the number of U-to-C conversions in our sequencing data. We generated data from consecutive time points after DRB release (i.e. at 0, 5, 15, 30 and 45 min) and monitored the progression of elongation to calculate RNAPII translocation speeds. In this approach, measurements do not rely on the slope of the intronic signal, but rather on the progressing “front” of RNAPII elongation, and thus infer translocation speeds in a manner orthogonal to our previous data. In the end, this new data is in full agreement with measurements inferred from nascent or bulk (ribodepleted) RNA-seq used previously, and thus in full support of our findings as regards age-related speed changes. Please note that we did not resort to single molecule analyses (i.e. to imaging-based experiments) for two reasons. First, these are highly complex approaches difficult to perform in primary cells and requiring expertise that are beyond those available to us. Second, such an approach would be of considerably lower throughput to the ones we used here, thus running the risk of only sampling a small number of gene-specific effects.

2) Related this, there are numerous reports of systematic differences in base composition and histone modifications between intronic and exonic sequences in systems from yeast to humans. It would therefore be important to demonstrate that the observed changes are not intron-specific.

Certainly, the reviewer is correct that important differences between introns and exons exist in this respect. For example, elongation speed is slowed down at exon-intron junctions, presumably to aid the splicing process. The slope-based assay that we are using cannot determine the elongation speed inside exons basically by design. However, we are not aware of any single existing assay that is capable of measuring Pol-II elongation speed within specific exons, simply due to the fact that exons are generally very short compared to introns. Our speed measurements using the DRB-4SU assay - which covers both exons and introns - came to the same conclusion. However, since the vast majority of genic sequence is intronic that change might also only be due to speed changes in introns. As a consequence, we cannot make any statements about Pol-II speed in exons. In order to make this more explicit, we have changed the text in the manuscript accordingly.

E.g. in the Abstract:

“The average transcriptional elongation speed (Pol-II speed) in introns increased with age in all five species.”

... and in the Discussion:

“We have found a consistent increase in average intronic Pol-II elongation speed with age ...”

3) Fig. 4: This seems to be something of an after-thought, presumably added to provide some mechanistic basis for the RNA seq data. Given that HDAC inhibitors have been linked to aging, alterations in chromatin structure appear very plausible. It was, however, unclear to this referee how the changes in elongation rate can be functionally related to the "fuzzy" nucleosome positioning? Is there reason to think that alteration in nucleosomes are a cause rather than a consequence of altered transcription?

Chromatin is composed of well-positioned 'phased' nucleosomes and 'fuzzy' nucleosomes in-between (Jiang & Pough 2009 PMID: 19204718). In cells where individual phased nucleosomes are lost this could also lead to the de-phasing of neighboring nucleosomes. Thus, increasing fuzziness may indicate a general loss of chromatin integrity and specifically more sparse positioning of nucleosomes. (Which is consistent with the increasing average distance between nucleosomes that we observed.) We have replaced the fuzziness score by a more advanced peak sharpness measure that accounts for peak height and peak width (Flores & Orzco 2011). We have added substantial new data to the revised version of our manuscript further corroborating a potential causal role of nucleosome packaging and chromatin density on Pol-II speed and ageing phenotypes. Increase in chromatin compaction in IMR90 cell lines by overexpressing H3 and H4 histones slows down the average elongation speed of RNA Pol II. See also our reply above to Reviewer 1 (comment 7) for more details on this topic.

Minor points:

4) In Figs. S10 and S13: The frequency of rare splice events and errors show a tendency for increases in the aging models. However, this is far from clear cut. Is there any way show that this tendency differs statistically from random?

All changes of mismatch levels are statistically significant (see **Extended Data Figure 13**) .

5) What does Fig. S14 demonstrate?

This figure has been removed. Since now all speed changes are quantified using log-fold changes we do not need to calibrate the speed estimates anymore.

6) P8: "(S8 " - Missing bracket

Text has been corrected accordingly.

7) Fig. S10: Progeria appears to be in blue, in contrast to legend.

This issue has been fixed.

8) Fig. S12: A different color combination would make the data points clearer.

This supplementary figure has been removed.

Reviewer Reports on the First Revision:

Referees' comments:

Referee #1 (Remarks to the Author):

In the manuscript "Aging-associated changes in transcriptional elongation influence metazoan longevity", Debès et al. have shown across different species that Pol II speeds and elongation rates in introns increase with age. This age-associated increase correlates with increase in production of rare splice isoform of genes that is likely due to mis-splicing as well as increased production of circular RNAs. Furthermore, the authors have shown that age also resulted in a decrease in nucleosome density and reduced the precision of nucleosome positioning, which contributed to the increased elongation speeds. The authors use genetic mutants of reduced Pol II speed as well as overexpression of histone proteins to manipulate Pol II speeds, thereby impacting lifespan in worms and flies.

The authors have made substantial efforts to address the reviewer comments experimentally and, in some cases, performed additional analyses to support their claims. Particularly noteworthy is the introduction of 4SU-DRB labelling to calculate average Pol II elongation speeds in young and senescent IMR90 cells which independently supports the authors' claims. While additional experiments might be added to improve the depth of data, the manuscript in its current form sufficiently supports the claims made and can be considered for publication to Nature. However, some parts need attention and minor revision.

1. Data figures for daf-2 mutants in worms and IRS1 null mice showing that long-lived mutants have reduced Pol II speeds should be called out in the text (p. 7).

2. Violin plots showing the RNA Pol-II elongation rate from 4SU-DRB labelling suggest that while the average elongation rate increased in senescent vs proliferating cells (Ext. Data Fig. 4), the elongation rate spreads show a wide variance in senescent cells implying that some genes showed slower Pol II speeds as compared to proliferating cells. Perhaps the authors would like to comment on this observation in the main text since this is the only method that allows one to visualize the spread in elongation speeds across individual genes.

3. The authors analyzed circRNA formation at the gene level and writes "We have tested such correlation and indeed did not find a correlation between speed changes (on a per gene basis) and increased formation of cirRNAs (from that given gene)." This should be introduced in the text for readers.

4. The revision has shown the effect of H3 overexpression in flies using UAS-Gal4 system. Overexpressing H3 and H4 in worms and measuring their lifespan would be trivial in comparison and would greatly support the claims as a broad mechanism that would make the paper of impact for this journal.

Referee #2 (Remarks to the Author):

Debes et al. present a revised manuscript examining “Aging-associated changes in transcriptional elongation influence metazoan longevity”. The manuscript is greatly improved from previous and an orthologous approach to examine elongation rate in one of the analyzed systems, human cell line(s) supports the findings from the more widely employed sequencing assay, which necessarily is the only current approach that could have been applied.

There are some comments that might improve or clarify the presentation.

An important issue that is under-discussed and potentially minimized inappropriately is the correlation between measured elongation rate and expression (in the intron slope assay). It is not clear how the correlation was examined, but it should be made much more clear what the correlation actually is. Within a sample, does elongation rate correlate with expression, and in which direction? To what extent are the introns that show changes in speed the introns that show changes in rate? If the analysis is done on expression-matched introns, are the differences still apparent? I think they absolutely will be, but it is important to understand if this is the case.

Other points

1. PCA plots are quite welcome and very useful. I recommend that they be paired with heatmap and correlation analysis among samples.
2. MNase reproducibility and metrics inferred before would be a histogram of fragment length distributions to indicate how well-matched digestions were across samples (and some are not over or underdigested), and potentially some correlation analysis for a particular region as example for how well repeats indicate reproducibility. The PCA is also welcome to include in the manuscript and not just rebuttal.
3. If Ext. Data Fig. 2 were instead x-y scatter plots, the variance in individual introns would be apparent; this gives a much better idea of how the results actually are (for example when histograms are the same, do we also observe a tight correlation in the actually determined elongation rates for specific introns)?
4. The TUC-seq methodology is described as generating T-C changes in RNA, and the analysis should be stranded, so why are there both A-G and T-C changes apparent over time? For Ext. Data Fig. 4, only IMR90 cells are shown. Presumably this analysis should also be paired because introns are matched between the two cell states. The Methods state that HUVEC cells were also done, and both HUVEC and IMR90 were done in biological replicates. These data do not appear to be shown in the figure. This is important because for some assays, HUVEC and IMR90 are divergent in phenotype (the RNA splicing assays), which also argues against splicing defects being a necessary consequence of purported elongation changes.
5. p. 7 first para: This para should cite Fig. 1.

6. The negative results on rate changes in specific genes strike me as better for a supplemental discussion. The discussion on potential gene expression changes is more important, but it seems less optimal to focus on expression changes of specific genes that show elongation changes rather than expression changes regardless of elongation change, and expressly histones should be checked for expression changes and reversion upon treatment, given where the paper ends up. Though perhaps it is worth noting that in the senescent cells, minimally, SUPT6H and SUPT16H reduction would be expected to have global effects on chromatin structure in transcribed regions; as the major nucleosome analysis is in these systems, there could also be system-specific changes to gene expression that are meaningful (and potentially not necessarily drivers across all systems).

7. There is a typo in legend for Ext. Data Fig. 8, where CRISPR edited *ama-1* is referred to as m322 and not syb2315. This is a beautiful experiment; how many independent edited lines were there and were they in fact tested? Regardless, this experiment should be cited in main text as it is a compelling control.

8. p. 9, “An optimal elongation rate...”: Please consider that elongation rate has also been described as affecting alternative polyadenylation, 3'-end formation at histone genes, and in RNA secondary structure leading to altered RNA processing, Cf. Saldi, 2018. *Genes Dev.*, 10.1101/gad.314948.118; Saldi, 2021. *Mol. Cell*, 10.1016/j.molcel.2021.01.040 and others regarding APA.

9. p. 9, “We observed more spliced transcripts...”: This has also been observed for a fast Pol II mutant in *A. thaliana*: Leng et al., 10.15252/embr.201949315.

10. Fig. 3a: Yellow dot for progeria sample is small.

11. p. 10, “Further, we noticed that genes with increased Pol-II elongation speed had on average larger fractions of rare isoforms than genes with reduced speed (Extended Data Fig. 11b).”: Does this relate to the correlation with expression- analysis vs expression level should be done to ask if there is confounding variable in detecting rare isoforms (more reads, greater potential for detection).

12. p. 10, “As indicated above, no aging-associated process was on average significantly more affected by the age-associated Pol-II speed increase than other cellular processes.”: This sentence is not very clear.

13. Fig. 3b examples: Because HUVEC and IMR90 show quite disparate effects in RNA processing phenotypes, I don't know how powerful these examples are (see ED12, 3a).

14. Fig. 4c,e: I think it would be important to ask if expression differences can also lead to the observed changes. If there are elongation-dependent chromatin defects, they might be expected to also correlate with level of expression. It seems that it would also be useful to examine genes in different elongation rate change bins, and in different expression bins, and then match rate change bins by expression to see if nucleosome changes are greater in one dimension than in the other.

In general, even though effects are statistically significant (most strongly for occupancy it seems), the fuzziness and the spacing analyses are less than compelling. Minimally, analysis in a and c/e

should be done in expression aware fashion to ask about confounding variables (regardless, the differences in occupancy appear to be real, though it must be clearly explained how such global effects in occupancy were determined). This is not clear from methods; is this simply peak height? If so, height is measured how (peak to trough)? Absolute occupancy is difficult to determine.

15. p. 15, "Thus, reduced precision in the assembly of the chromatin fiber can be correlated with changes in Pol-II speed (42,46,50).": I do not think these are the correct references for this statement. Please check over citations to make sure they are correct. Please also see Qiu et al. (Genome Biol. 2020 10.1186/s13059-020-02040-0) for Pol II mutants altering nucleosome spacing.

16. Fig. 5a: Histone protein expression is measured in one system; what about histone RNA levels across the examined systems? Given some of the authors' here recent findings that there is a TORC1-histone axis functioning in lifespan, histone expression across treatments would be interesting to discuss.

17. Fig. 5f: Why does it appear there is a severe dox effect on the GFP-only (compare +/- dox)?

18. p, 30 "using substract and merge commands": I assume "subtract" is meant.

19. Ext. Data Table 1: What is the measure of correlation shown? r ? r^2 ? Pearson/Spearman? The correlation with expression is perhaps not "weak". A similar level correlation for elongation speed compared to GRO-seq data is considered "highly significant" elsewhere in the manuscript.

20. In general, make sure n of genes/introns/features is always given for any of the analyses.

21. Methods regarding mismatch detection are insufficient to clearly understand aspects of the analysis. Potentially, the spectrum of mismatches should be reported (it is expected that there could be some bias for actual mismatches vs different spectrum of sequencing errors, and potentially could be differentiated from sequencing errors depending on if error spectrum known for platform or through use of sequence over adapters or other non reverse-transcribed features). Other aspects of approach (were base quality filters in sequencing used to remove some positions?), any coverage level filter (one out of how many reads at a position?).

Referee #3 (Remarks to the Author):

The major problem in the original version of the MS, raised by all referees, is in the calculation of elongation rates. This is the key result of the MS and has been supported in the revised MS. However, this notable and unexpected finding needs to be very robustly established, which has not yet been achieved.

1: Ext. Data Fig. 4: The verification of the quantification method should be done more carefully using a gene-based analysis. The boxplot shown is consistent with the major claim of the MS, however, this is not really a validation of the method. 4SU-DRB labelling allows measurement of transcription elongation rate [kb/min] whereas RNA-seq based calculation results in the measure of [slope]. To validate the method, the [slope] and [kb/min] should correlate on gene-by-gene basis. It would be valuable to present these data as a scatterplot, to convincingly demonstrate that the [slope] measurement is reliable. This validation should be a part of Fig. 1.

An additional question is whether nucleotide analog incorporation potentially affects elongation rates?

2: P9, Ext. Data Fig. 10: The authors report the surprising finding that increased elongation rates enhance splicing efficiency. This appears to go against the popular “window of opportunity” model for cotranscriptional splicing. Relevant papers are cited in the Introduction, but not in this section. The changes in the proportion of unspliced reads could, as proposed, reflect alterations in the rate of splicing. Alternatively, it could reflect altered mRNA turnover. This could be addressed by analyses of the 4sU labeling data that the authors have collected.

3: Fig. 5: The new findings about role of H3 overexpression and its effect on the longevity are rather confusing than strengthening the MS. The rebuttal states “...overexpression of histone H3 in the glia of fly brains, increases chromatin compaction as would be expected ...”. Actually, this is not what might have been expected. The mechanistic explanation for how overexpression of a single histone gene would increase total levels of nucleosomes is not clear, unless H3 levels are normally strictly limiting for nucleosome assembly. If the authors have evidence for this it would be a useful addition to the MS. Otherwise, can the authors provide evidence, eg a western blot, showing that overexpression of single histone increases level of all histones?

Author Rebuttals to First Revision:

Point-by-point Response to Reviewer Comments

Referee #1 (Remarks to the Author):

In the manuscript “Aging-associated changes in transcriptional elongation influence metazoan longevity”, Debès et al. have shown across different species that Pol II speeds and elongation rates in introns increase with age. This age-associated increase correlates with increase in production of rare splice isoform of genes that is likely due to mis-splicing as well as increased production of circular RNAs. Furthermore, the authors have shown that age also resulted in a decrease in nucleosome density and reduced the precision of nucleosome positioning, which contributed to the increased elongation speeds. The authors use genetic mutants of reduced Pol II speed as well as overexpression of histone proteins to manipulate Pol II speeds, thereby impacting lifespan in worms and flies.

The authors have made substantial efforts to address the reviewer comments experimentally and, in some cases, performed additional analyses to support their claims. Particularly noteworthy is the introduction of 4SU-DRB labelling to calculate average Pol II elongation

speeds in young and senescent IMR90 cells which independently supports the authors' claims. While additional experiments might be added to improve the depth of data, the manuscript in its current form sufficiently supports the claims made and can be considered for publication to Nature. However, some parts need attention and minor revision.

We thank the reviewer for the overall positive evaluation of our manuscript and thank for the constructive criticism.

1. Data figures for *daf-2* mutants in worms and IRS1 null mice showing that long-lived mutants have reduced Pol II speeds should be called out in the text (p. 7).

We have changed the text to explicitly call out the speed reduction in the interventions as follows: “In all comparisons, except IRS1-null mice and livers from 26 months old DR mice, lifespan-extending interventions resulted in a significant reduction of Pol-II speed.”

2. Violin plots showing the RNA Pol-II elongation rate from 4SU-DRB labelling suggest that while the average elongation rate increased in senescent vs proliferating cells (Ext. Data Fig. 4), the elongation rate spreads show a wide variance in senescent cells implying that some genes showed slower Pol II speeds as compared to proliferating cells. Perhaps the authors would like to comment on this observation in the main text since this is the only method that allows one to visualize the spread in elongation speeds across individual genes.

We thank the reviewer for raising this point. Indeed, when we used the slope-based speed measures we also observed that many individual genes (introns) showed reduced elongation speed upon ageing. Thus, a consistent observation using both assays is that while the majority of genes exhibited increased speed, some genes also had reduced speeds with age. We have changed the manuscript to make this explicit (page 5): “Note that, although many individual genes showed a decrease in elongation speed with aging in both assays, the majority exhibited increased speed.”

Please additionally note that when we limit our analysis to the consistently changing introns (Fig. 1 and Extended Data Figure 3), the fraction of introns with an increase in speed is greater.

3. The authors analyzed circRNA formation at the gene level and writes “We have tested such correlation and indeed did not find a correlation between speed changes (on a per gene basis) and increased formation of cirRNAs (from that given gene).” This should be introduced in the text for readers.

We have added an extra column in Supplementary Table 2 showing the correlation between change in circRNA formation and change in elongation rate. We are also explicitly mentioning this observation now in the text of the manuscript (p.13):

“Nevertheless, our data does not provide evidence that increased circRNA levels directly result

from increased Pol-II speed, despite it being a consequence of the overall reduced quality in RNA production.”

4. The revision has shown the effect of H3 overexpression in flies using UAS-Gal4 system. Overexpressing H3 and H4 in worms and measuring their lifespan would be trivial in comparison and would greatly support the claims as a broad mechanism that would make the paper of impact for this journal.

We agree with the reviewer that such experiments would strengthen the manuscript. However, conducting such experiments is less trivial than it might seem at first glance. In flies we overexpressed H3 specifically in glial cells, which was motivated by earlier data from our own lab (Partridge lab). In *C. elegans* we could only globally overexpress H3 in the whole worm, which might have developmental effects. Also, it would be difficult to control the amount of overexpression such that it would not lead to other detrimental side effects. For example, overexpressing H3 and H4 in human cells had different physiological consequences (Fig. 5f). Hence, there seems to be a narrow window of optimal histone expression levels. Fine tuning such a genetic system would at least take a lot of time and substantially delay the publication of this work. Further we note that our findings are already backed up by earlier work, including work in *C. elegans* (see Ref. 54, Sural et al. 2020). Since we already have data for human cell lines and flies we would prefer to publish the data as is.

Referee #2 (Remarks to the Author):

Debes et al. present a revised manuscript examining “Aging-associated changes in transcriptional elongation influence metazoan longevity”. The manuscript is greatly improved from previous and an orthologous approach to examine elongation rate in one of the analyzed systems, human cell line(s) supports the findings from the more widely employed sequencing assay, which necessarily is the only current approach that could have been applied.

There are some comments that might improve or clarify the presentation.

We are thankful for the appreciation of our efforts and for the constructive feedback that has helped to further improve the presentation of our results.

An important issue that is under-discussed and potentially minimized inappropriately is the correlation between measured elongation rate and expression (in the intron slope assay). It is not clear how the correlation was examined, but it should be made much more clear what the correlation actually is. Within a sample, does elongation rate correlate with expression, and in which direction? To what extent are the introns that show changes in speed the introns that show changes in rate? If the analysis is done on expression-matched introns, are the

differences still apparent? I think they absolutely will be, but it is important to understand if this is the case.

We only compute speed *changes* by comparing slopes of the same intron between conditions, not elongation rates per sample, because for multiple reasons the slopes are not easily comparable within samples (i.e. one should be careful with comparing slopes of different introns; see also our reply to Reviewer 3 comment 1 below). Therefore, we only compare slopes of the same introns between samples and cannot compute the correlation of slopes with absolute expression within a given sample. The speed changes have however in general a consistent negative correlation with expression change, as it can be seen in Supplementary Table 1.

In order to evaluate the impact of this relationship on our conclusions, we split genes into three groups (upregulated, downregulated or unchanged). We generally see increased average elongation speed irrespective of the direction of the expression change, even though the negative correlation exists.

Response Letter Figure 1: Log2FC of average elongation rate in HUVECs, fruit flies (brain; old *versus* young) and worms (whole body; slow Pol-II mutant *versus* wild type), binned by the change in the expression. These data are representative for all species. Genes were ranked in order of expression change. Overexpressed bin contains the top 30% of the genes, underexpressed bin contains the bottom 30% and the rest are in the bin called “No Change”.

If we only use introns that don't change expression, our results remain consistent, as shown in the following figure.

Response Letter Figure 2: Log2 fold change of average Pol-II elongation rates in worm, fruit fly, mouse, rat, human blood, and human cell culture. Error bars show median variation $\pm 95\%$ confidence interval (Wilcoxon signed rank test). Only introns in genes with negligible changes in expression (absolute $\log_2\text{FC} < 0.3$) were used for this analysis.

Other points

1. PCA plots are quite welcome and very useful. I recommend that they be paired with heatmap and correlation analysis among samples.

We agree with the reviewer that heatmaps are a useful visualization tool. However, we already have a lot more extended data figures than recommended in the journal's submission rules. We are therefore including the figures in this response letter, which (according to Nature's current publication standards) will be published along with the final paper.

Response Letter Figure 3: Pairwise correlations of slopes of intronic read distributions. (a) *C. elegans* wt 21 d vs 1 d; (b) *C. elegans* ama-1(m322) 14 d vs wt 14 d, (c) *D. melanogaster* wt heads 50 d vs 10 d, (d) *D. melanogaster* RplI2154 heads 50 d vs wt 50 d, (e) *M. musculus* kidney: 24 mo vs 3 mo, (f) *R. norvegicus* old vs. young, (g) *H. sapiens* HUVEC and (h) *H. sapiens* IMR90: senescent vs. proliferating.

Response Letter Figure 4: Slopes of intronic read distribution (slopes of all introns that pass the filtering criteria). (a) *C. elegans* wt 21 d vs 1 d; (b) *C. elegans* ama-1 (m322) 14 d vs wt 14 d, (c) *D. melanogaster* wt heads 50 d vs 10 d, (d) *D. melanogaster* RplI2154 heads 50 d vs wt 50 d, (e) *M. musculus* kidney: 24 mo vs 3 mo), (f) *R. norvegicus* old vs. young, (g) *H. sapiens* HUVEC and (h) *H. sapiens* IMR90: senescent vs. proliferating.

2. MNase reproducibility and metrics inferred before would be a histogram of fragment length distributions to indicate how well-matched digestions were across samples (and some are not over or underdigested), and potentially some correlation analysis for a particular region as example for how well repeats indicate reproducibility. The PCA is also welcome to include in the manuscript and not just rebuttal.

The fragment length distributions are very similar across all samples and the coverage of the different MNase-seq replicates are highly correlated with each other. We have added extra panels in Figure 4 to show the PCA analysis.

Response Letter Figure 5: Fragment length distribution of the MNase-seq IMR90 samples.

Response Letter Figure 6: Correlation heatmap of the fragments of the MNase-seq IMR90 samples (bin size=1000 bp).

3. If Ext. Data Fig. 2 were instead x-y scatter plots, the variance in individual introns would be apparent; this gives a much better idea of how the results actually are (for example when histograms are the same, do we also observe a tight correlation in the actually determined elongation rates for specific introns)?

We thank the reviewer for the comment. We are now showing scatter plots in Ext. Data Figure 2.

4. The TUC-seq methodology is described as generating T-C changes in RNA, and the analysis should be stranded, so why are there both A-G and T-C changes apparent over time? For Ext. Data Fig. 4, only IMR90 cells are shown. Presumably this analysis should also be paired because introns are matched between the two cell states. The Methods state that HUVEC cells were also done, and both HUVEC and IMR90 were done in biological replicates. These data do not appear to be shown in the figure. This is important because for some assays, HUVEC and IMR90 are divergent in phenotype (the RNA splicing assays), which also argues against splicing defects being a necessary consequence of purported elongation changes.

As for the A-G changes, both sequencing and subsequent analysis were strand-specific. However, in paired-end sequencing the second read is always on the opposite strand of the first read. Thus, a T-C mismatch on the plus strand will be interpreted as a T-C mismatch in a first read of a given read pair, while the same mismatch would be interpreted as an A-G mismatch if it is covered by a second read of another read pair.” All changes mapped are bona fide T-C conversions in the RNA transcribed from the proper strand given the gene directionality.

We have removed the mentioning of HUVEC cells from the methods. The 4sU sequencing of HUVECs was only done for calibration purposes; we do not have data from senescent HUVEC cells.

5. p. 7 first para: This para should cite Fig. 1.

We have added the missing citation.

6. The negative results on rate changes in specific genes strike me as better for a supplemental discussion. The discussion on potential gene expression changes is more important, but it seems less optimal to focus on expression changes of specific genes that show elongation changes rather than expression changes regardless of elongation change, and expressly histones should be checked for expression changes and reversion upon treatment, given where the paper ends up. Though perhaps it is worth noting that in the senescent cells, minimally, SUPT6H and SUPT16H reduction would be expected to have global effects on chromatin structure in transcribed regions; as the major nucleosome analysis is in these systems, there could also be system-specific changes to gene expression that are meaningful (and potentially not necessarily drivers across all systems).

We thank the reviewer for this suggestion. Indeed, in senescent cells, both HUVEC and IMR90, H3 protein levels were reduced compared to proliferating cells, which - as indicated by the reviewer - makes a lot of sense in view of the rest of our work. We are now showing this data in a new panel Fig. 5a. The mechanisms that cause RNA-Pol II increase may indeed be different depending on the system under study.

Neither SUPT6H or SUPT16H mRNA levels are differentially expressed with senescence in HUVEC or IMR90 cells.

7. There is a typo in legend for Ext. Data Fig. 8, where CRISPR edited ama-1 is referred to as m322 and not syb2315. This is a beautiful experiment; how many independent edited lines were there and were they in fact tested? Regardless, this experiment should be cited in main text as it is a compelling control.

The typo has been corrected and we have added a citation to the External Data Figure in the main text.

8. p. 9, “An optimal elongation rate...”: Please consider that elongation rate has also been described as affecting alternative polyadenylation, 3'-end formation at histone genes, and in RNA secondary structure leading to altered RNA processing, Cf. Saldi, 2018. *Genes Dev.*, 10.1101/gad.314948.118; Saldi, 2021. *Mol. Cell*, 10.1016/j.molcel.2021.01.040 and others regarding APA.

We agree with the reviewer that the Saldi, 2021 reference is pertinent and have included it in the manuscript.

9. p. 9, “We observed more spliced transcripts...”: This has also been observed for a fast Pol II mutant in *A. thaliana*: Leng et al., 10.15252/embr.201949315.

We thank the reviewer for hinting at this publication and have added this citation as well.

10. Fig. 3a: Yellow dot for progeria sample is small.

The plot has been redone since the progeria sample was removed.

11. p. 10, “Further, we noticed that genes with increased Pol-II elongation speed had on average larger fractions of rare isoforms than genes with reduced speed (Extended Data Fig. 11b).”: Does this relate to the correlation with expression- analysis vs expression level should be done to ask if there is confounding variable in detecting rare isoforms (more reads, greater potential for detection).

We repeated the rare splicing analysis with a more sophisticated approach and the correlation of elongation speed and rare isoforms does not exist anymore. We have thus removed Extended Figure 11b and made the appropriate alterations to the text.

12. p. 10, “As indicated above, no aging-associated process was on average significantly more affected by the age-associated Pol-II speed increase than other cellular processes.”: This sentence is not very clear.

The sentence has been rewritten: “Genes exhibiting accelerated Pol-II elongation were not enriched for specific cellular processes, indicating that speed increase is probably not a deterministically cell-regulated response, but rather a spontaneous age-associated defect.”**(Discussion)**

13. Fig. 3b examples: Because HUVEC and IMR90 show quite disparate effects in RNA processing phenotypes, I don't know how powerful these examples are (see ED12, 3a).

These examples were dropped, as the splicing analysis changed.

14. Fig. 4c,e: I think it would be important to ask if expression differences can also lead to the observed changes. If there are elongation-dependent chromatin defects, they might be

expected to also correlate with level of expression. It seems that it would also be useful to examine genes in different elongation rate change bins, and in different expression bins, and then match rate change bins by expression to see if nucleosome changes are greater in one dimension than in the other.

In general, even though effects are statistically significant (most strongly for occupancy it seems), the fuzziness and the spacing analyses are less than compelling. Minimally, analysis in a and c/e should be done in expression aware fashion to ask about confounding variables (regardless, the differences in occupancy appear to be real, though it must be clearly explained how such global effects in occupancy were determined). This is not clear from methods; is this simply peak height? If so, height is measured how (peak to trough)? Absolute occupancy is difficult to determine.

We have changed the text of the methods to explain in detail how the score is calculated. Since occupancy was indeed not the clearest term, we refer to the measurement now as nucleosomal density. We removed Fig 4c,e in the new version of the manuscript. In addition we have repeated the analysis of nucleosome density by grouping genes either by absolute expression (Response Letter Figure 8) or by expression change (Response Letter Figure 9). The conclusions regarding changes in nucleosome density are not affected by the stratification of genes based on expression.

Response Letter Figure 8: Average differences in nucleosome density between exons and introns and between proliferating and senescent cells, binned by average gene expression.

Response Letter Figure 9: Average differences in nucleosome density between exons and introns and between proliferating and senescent cells, binned by gene expression change (overexpressed: $\log_2FC > 0.5$, underexpressed: $\log_2FC < 0.5$).

15. p. 15, “Thus, reduced precision in the assembly of the chromatin fiber can be correlated with changes in Pol-II speed (42,46,50).”: I do not think these are the correct references for this statement. Please check over citations to make sure they are correct. Please also see Qiu et al. (Genome Biol. 2020 10.1186/s13059-020-02040-0) for Pol II mutants altering nucleosome spacing.

The citations have been fixed.

16. Fig. 5a: Histone protein expression is measured in one system; what about histone RNA levels across the examined systems? Given some of the authors' here recent findings that there is a TORC1-histone axis functioning in lifespan, histone expression across treatments would be interesting to discuss.

It is very hard to accurately estimate expression levels of histones from RNA-seq data. Histone genes are short and transcribed together with multiple variants of similar sequences. In order to accurately measure histone expression across all the examined systems, multiple new experiments would have to be performed. In particular, Western blots of protein levels (as we did for Fig. 5a) would be required, which is practically impossible for all species and all histone proteins.

17. Fig. 5f: Why does it appear there is a severe dox effect on the GFP-only (compare +/- dox)?

The plots are showing relative growth within a set of samples (all normalized to start at 1), so this is not a severe effect. It is only severe compared to the boost that the overexpression of either H3 or H4 produces.

18. p, 30 “using subtract and merge commands”: I assume “subtract” is meant.

The typo has been corrected.

19. Ext. Data Table 1: What is the measure of correlation shown? r ? r^2 ? Pearson/Spearman? The correlation with expression is perhaps not “weak”. A similar level correlation for elongation speed compared to GRO-seq data is considered “highly significant” elsewhere in the manuscript.

The correlation used was Pearson’s correlation (r). We agree with the reviewer that the correlation with expression needed more analysis (please refer to our comment at the beginning of our response to Reviewer 2)

20. In general, make sure n of genes/introns/features is always given for any of the analyses.

Number of features is now given for all the analyses in the main text.

21. Methods regarding mismatch detection are insufficient to clearly understand aspects of the analysis. Potentially, the spectrum of mismatches should be reported (it is expected that there could be some bias for actual mismatches vs different spectrum of sequencing errors, and potentially could be differentiated from sequencing errors depending on if error spectrum known for platform or through use of sequence over adapters or other non reverse-transcribed features). Other aspects of approach (were base quality filters in sequencing used to remove some positions?), any coverage level filter (one out of how many reads at a position?).

We have added the relevant information in the methods section.

As for the sequencing errors, it has been established that sequencing can introduce significant numbers of mismatches and that these errors are biased. However, since we are always comparing old versus young samples (both coming from the same sequencing batch), we are correcting for sequencing errors, which are expected to be the same within a batch.

Referee #3 (Remarks to the Author):

The major problem in the original version of the MS, raised by all referees, is in the calculation of elongation rates. This is the key result of the MS and has been supported in the revised MS. However, this notable and unexpected finding needs to be very robustly established, which has not yet been achieved.

We thank the Reviewer for pointing out the improvements that we had achieved in the previous version of our MS and for the constructive criticism. The new analyses that we

have performed to address the remaining concerns of this and the other two reviewers have together substantially improved the robustness of our main finding, namely that RNAPII elongation speed increases with age.

1: Ext. Data Fig. 4: The verification of the quantification method should be done more carefully using a gene-based analysis. The boxplot shown is consistent with the major claim of the MS, however, this is not really a validation of the method. 4SU-DRB labelling allows measurement of transcription elongation rate [kb/min] whereas RNA-seq based calculation results in the measure of [slope]. To validate the method, the [slope] and [kb/min] should correlate on gene-by-gene basis. It would be valuable to present these data as a scatterplot, to convincingly demonstrate that the [slope] measurement is reliable. This validation should be a part of Fig. 1.

We agree with the reviewer that ideally the two measures of RNAPII elongation speed should be strongly correlated. However, we do not expect a perfect correlation of the slope measurement with the 4sU estimation, because the slopes are affected by intron/gene specific confounding factors such as intron length. Therefore, in our main analysis (Figs. 1c & 2a) we compare slopes of the same intron between conditions and we only report fold changes of slopes, but not absolute slopes. (See also our reply to Reviewer 2 above.) On the other hand, the 4sU assay enables measuring absolute elongation rates in [bp/min]. Despite those limitations we have performed such a direct comparison (see the new Figure 1d). In order to be able to compare the two speed measures for a sufficient number of genes we had to develop a new approach for deriving speed estimates from our time course 4sU data. The approach presented in the previous version of this manuscript was limited to genes with a minimum length of 100 kb, which resulted in too few genes overlapping with slope-based estimates. Therefore, we devised a new method that is slightly less precise but applicable to shorter genes, which increased the number of genes with speed estimates from both assays to $n = 217$ (intronic read distribution of total RNA-seq versus 4sU-based time course data). Directly comparing those estimates on a gene-by-gene basis (as requested by the reviewer) resulted in a highly significant correlation ($p = 2.5e-06$, Figure 1d). Because our initial approach for deriving speed estimates from 4sU labelling data is more precise, we are still using it for the data in Figure 1e. Both approaches lead to the same conclusion: elongation speed increases with age.

An additional question is whether nucleotide analog incorporation potentially affects elongation rates?

The existing literature on this topic does not provide any evidence that transcriptional elongation is affected by the analog. This is including the initial 4SUDRB-seq work (Fuchs, et al. "4sUDRB-seq: measuring genomewide transcriptional elongation rates and initiation frequencies within cells." *Genome biology* 15.5 (2014): 1-11.), the SLAM-seq method (Herzog, et al. "Thiol-linked alkylation of RNA to assess expression dynamics." *Nature methods* 14.12 (2017): 1198-1204.), TUC-seq (Lusser, et al. "Thiouridine-to-cytidine conversion sequencing (TUC-Seq) to measure mRNA transcription and degradation

rates." *The Eukaryotic RNA Exosome*. Humana, New York, NY, 2020. 191-211.) and TT-seq (Schwalb, et al. "TT-seq maps the human transient transcriptome." *Science* 352.6290 (2016): 1225-1228.). Melvin and co-authors (*Eur J Biochem*, 1978) reported after using 4sU concentrations 20 times greater than the one we are using in our work, "... even after long exposure to high concentrations of 4-thiouridine, the syntheses of RNA and protein is not affected". More recent work by Altieri and Hertel, *PLoS ONE* (2021) has shown that 4sU incorporation does not affect splicing or RNA half-lives as long as the 4sU incorporation rate stays below 30%.

Even if there was an effect, a 4sU nucleotide is only installed in at most 5% of the U-sites, which means that a delay could occur at a maximum of ~1.5% of all nucleotide positions. Even if this were significantly slower per 4sU incorporation, the effect on the measured elongation rate would be negligible.

2: P9, Ext. Data Fig. 10: The authors report the surprising finding that increased elongation rates enhance splicing efficiency. This appears to go against the popular "window of opportunity" model for cotranscriptional splicing. Relevant papers are cited in the Introduction, but not in this section. The changes in the proportion of unspliced reads could, as proposed, reflect alterations in the rate of splicing. Alternatively, it could reflect altered mRNA turnover. This could be addressed by analyses of the 4sU labeling data that the authors have collected.

We thank the reviewer for pointing this out and for hinting at the incomplete referencing of this section. The average increase in elongation speed is very small. Thus, it does not prevent splicing altogether. The fact that Pol-II reaches the end of introns earlier shortens the time-to-splicing (see Ref. 29), which explains our observation. Our findings are consistent with earlier work and we are now properly referencing these earlier findings (Refs. 28-31).

We agree with the reviewer that the 4sU labeling data could provide some insights, but its coverage is too shallow to allow for an analysis of splicing efficiency. Since never more than 4% of the reads were labeled, we could not reliably distinguish new from old individual splicing events. Our speed measure, on the other hand, is using the *distribution* of labeled reads in the entire gene body, hence using much more information.

Although our interpretation is consistent with the existing literature we cannot formally rule out that there is some kind of effect due to altered RNA turnover on the splicing efficiency measure. Therefore, we are now mentioning this caveat in the discussion: "However, we cannot exclude the possibility that this increase resulted from changes in RNA half-lives." (p. 18)

3: Fig. 5: The new findings about role of H3 overexpression and its effect on the longevity are rather confusing than strengthening the MS. The rebuttal states "...overexpression of histone H3 in the glia of fly brains, increases chromatin compaction as would be expected ...". Actually, this is not what might have been expected. The mechanistic explanation for how overexpression

of a single histone gene would increase total levels of nucleosomes is not clear, unless H3 levels are normally strictly limiting for nucleosome assembly. If the authors have evidence for this it would be a useful addition to the MS. Otherwise, can the authors provide evidence, eg a western blot, showing that overexpression of single histone increases the level of all histones?

We thank the reviewer for noticing this point, because our original wording was indeed imprecise. Early work from Feser et al. (Ref. 47) in yeast already showed that histone levels decline with age, and that overexpressing H3 can compensate for this loss and extend lifespan. Later work in *C. elegans* (Ref. 54) and *D. melanogaster* (Ref. 56) showed that overexpressing individual histone components increases lifespan also in animals. In response to the reviewer's comment we are now showing in Figure 5a that H3 levels decline also in human cells upon entry into replicative senescence, which is at least partially compensated through overexpressing this protein. Overexpression of any component of a protein complex can increase the efficiency of its assembly (within limits), because the probability that all required components are available at the assembly location increases. For example, we had shown that overexpressing the proteasomal subunit RPN6 alone was sufficient to increase the assembly of functional proteasomes (Tain et al. 2017, <https://pubmed.ncbi.nlm.nih.gov/28916541/>). Further, H3 and H4 are distinct from other nucleosome components, because they are deposited first on the DNA before other histone components get recruited to complete the assembly of a new nucleosome (Krude 1995, [https://doi.org/10.1016/S0960-9822\(95\)00245-4](https://doi.org/10.1016/S0960-9822(95)00245-4); Liu et al. 2017, <https://doi.org/10.1126/science.aah4712>). It is therefore plausible that overexpression of these particular components (H3, H4) will increase the assembly of nucleosomes.

All that the overexpression of H3 is meant to achieve is to compensate for the age-related loss of histone proteins. The correct interpretation of Figure 5g is that H3 overexpression replenishes the pool of mono-nucleosomes without (over-)compacting chromatin.

Therefore we have changed the text in the manuscript in this section as follows:

“H3 overexpression led to significantly increased numbers of mono-nucleosomes in aged (60 day-old) compared to the wild-type fly heads (Fig. 5g), thus possibly compensating for age-associated loss of histone proteins. Further, H3 overexpression in glial cells increased fruit fly lifespan (Fig. 5h).”

Reviewer Reports on the Second Revision:

Referees' comments:

Referee #1 (Remarks to the Author):

This new revision further extends the depth of data to support the claims of the paper. I was perhaps the most supportive reviewer at the previous round and I continue to feel that the depth of the work and the toned down and defined conclusions make this of interest to the readership of Nature.

Although I still feel expression of even 'glial' limited H3/H4 in *C. elegans* and titration of levels is fairly trivial and standard practice - ie I do not agree with the rebuttals statement that it isn't - I sympathize with their desire to publish their data as it. Don't we all!

But overall all this is not a deal breaker. I also have looked at the response to the other reviewers and in my assessment this manuscript has reached the bar set by all.

Referee #2 (Remarks to the Author):

I feel the manuscript has improved with the second round of revisions and the much more compelling analysis for elongation rate. Comments are minor:

Lifespan effects from just expression H3 in *Drosophila* glial cells. Is there precedence for lifespan effects in just neuronal or brain-related tissues? This is a very nice result but it is quite striking and some context on how striking it is could be important.

“flatbad” should be “flatbed” in Methods p.23

“(Quiagen” should be Qiagen p.27

ED F1 still has progeria sample?

Referee #3 (Remarks to the Author):

The authors have strengthened the MS and addressed the key questions raised in both previous rounds of review. I would recommend acceptance of the revised MS.